# PinkyCaMP: an mScarlet-based calcium sensor with enhanced brightness, photostability and multiplexing capabilities

Ryan Fink [1,2,18], Shosei Imai [3,18], Nala Gockel [4], German Lauer[5], Kim Renken[2], Jonas Wietek [6,7], Paul J. Lamothe-Molina [8], Falko Fuhrmann[4], Manuel Mittag[4], Tim Ziebarth [5], Annika Canziani [8], Martin Kubitschke[2], Vivien Kistmacher[9], Anny Kretschmer[10], Eva Sebastian [11], Jana Ottens [2], Dietmar Schmitz[6,7,10,12,13,14], Takuya Terai [3], Jan Gründemann [11,15], Sami I. Hassan[9], Tommaso Patriarchi [8,16], Andreas Reiner [5], Martin Fuhrmann [4], Robert E. Campbell [3,17] & Olivia Andrea Masseck [1,2] ✉

Genetically encoded calcium (Ca²⁺) indicators (GECIs) are essential tools for monitoring neuronal activity, but the performance of red fluorescent GECIs has remained limited. In particular, many red indicators are relatively dim, produce low signal-to-noise ratios and can undergo unwanted photoswitching when exposed to blue light, restricting their use in all-optical experiments that combine imaging with optogenetics or multicolor imaging. Here we show the development of PinkyCaMP, a Ca²⁺ sensor based on the bright red fluorescent protein mScarlet. PinkyCaMP exhibits markedly improved brightness, photostability and signal-to-noise ratio compared to existing red GECIs, while remaining fully compatible with blue-light-based optogenetic and dual-color imaging approaches. PinkyCaMP is well-tolerated by neurons, showing no detectable toxicity or aggregation, both in vitro and in vivo. PinkyCaMP enables a broad spectrum of imaging modalities, including single-photon methods, such as fiber photometry, widefield imaging and miniature microscopy imaging, as well as two-photon imaging in awake mice.

Genetically encoded calcium (Ca²⁺) indicators (GECIs) are widely used for in vivo imaging of neuronal populations[1]. Since the development of the first green fluorescent protein (GFP)-based GECIs[2,3], substantial progress has been made in improving signal-to-noise ratios (SNRs) and kinetics, with advancements seen in GCaMP3, GCaMP5, G-GECO1 and GCaMP6 (refs. 4–6). These improvements have culminated in the recent development of jGCaMP8, which offers ultrafast kinetics and heightened sensitivity[7–9]. In contrast to green fluorescent GECIs, red-shifted sensors are generally preferable due to properties such as deeper tissue penetration and reduced phototoxicity of longer-wavelength light, as well as better spectral separation from blue-light-activated opsins that

enables multiplexed experiments with minimal optical crosstalk. In addition, the availability of high-performance red GECIs would allow dual-color imaging approaches, permitting the simultaneous recording of two distinct neuronal populations or of neuronal and glial activity; however, despite the advancements in green fluorescent biosensors, red-shifted GECIs still face limitations, such as lower brightness, reduced photostability and photoswitching, as well as challenges like lysosomal accumulation, which have persisted since they were first reported[6]. Commonly used red GECIs, such as jRCaMP1a,b[10], R-GECO1 (ref. 6), jRGECO1a[10], XCaMP-R[11] and RCaMP3 (ref. 12), each have their own limitations.

**Fig. 1 | Development and characterization of PinkyCaMP. a**, Overview of the domain structure of PinkyCaMP. The two gate post residues[45] in cpmScarlet are shown in orange (S28) and purple (W144). **b**, $\Delta F/F$ rank plot representing all crude proteins tested under the directed evolution screening conditions. $\Delta F/F$ values measured under these conditions are different from the values measured with purified proteins. **c**, Modeled structure of PinkyCaMP with the positions of mutations indicated. RS20, cpmScarlet, CaM and linker residues are colored cyan, red, magenta and yellow, respectively. The mutated residues are highlighted green. The model was prepared using AlphaFold3[46]. **d**, Excitation (emission at 620 nm) and emission (excitation at 540 nm) spectra of PinkyCaMP in the presence (39 µM) and absence of $Ca^{2+}$. **e**, Absorbance spectra of PinkyCaMP in the presence (39 µM) and absence of $Ca^{2+}$. **f**, $Ca^{2+}$ titration curve of PinkyCaMP, $n = 3$ replicates (mean ± s.d.). **g**, Baseline brightness in HEK cells expressing

jRCaMP1a (50 ± 5 a.u.; $n = 47$ total cells), jRGECO1a (294 ± 25 a.u.; $n = 50$ total cells), RCaMP3 (292 ± 35 a.u.; $n = 89$ total cells), and PinkyCaMP (701 ± 71 a.u.; $n = 93$ total cells), all with three replicate measurements. One-way ANOVA with Tukey's post hoc test, ****$P \le 0.001$, *$P \le 0.05$ (mean ± s.e.m.). **h**, Photoswitching of RCaMP3 ($n = 33$ cells) and PinkyCaMP ($n = 20$ cells) was assessed in HEK cells by imaging at 5 Hz with constant 560 nm excitation light and periodic 10-ms pulses of 470 nm light at 1 mW mm⁻² every 10 s. Inset is an enlarged version of each first stimulation (mean ± s.e.m.). Three replicate measurements were performed for each sensor. **i**, Averaged and normalized photostability traces of PinkyCaMP ($n = 72$ cells), RCaMP3 ($n = 56$ cells) and jRGECO1a ($n = 76$ total cells), from three replicate measurements (mean ± s.e.m.). Inset of photostability normalized against the unbound extinction coefficient for each respective sensor (mean ± s.e.m.).

The red fluorescent GECIs reported to date are derived from three distinct lineages of naturally occurring red fluorescent proteins (RFPs): R-GECO1 (ref. 6) and its progeny are based on the *Discosoma* sp.-derived[13] mApple[14]; jRCaMP1a,b[10] are based on *Entacmaea quadricolor* eqFP611-derived[15] mRuby[16]; and K-GECO1 (ref. 17) and FR-GECO1 (ref. 18) are based on *E. quadricolor* eqFP578-derived[19] mKate[20]. While jRGECO1a, evolved from R-GECO1, is the brightest commonly used red GECI, it is still more than three times dimmer than GCaMP6s[10] and although RCaMP3, the successor to jRGECO1a, is brighter than other red GECIs, it is still dim compared to green GECIs such as GCaMP6,7,8 (refs. 10,12). jRGECO1a, which is based on circularly permuted mApple

(cpmApple), exhibits photoswitching under blue light. This complicates its use in combination with other tools, as blue-light excitation can artificially increase its fluorescence without corresponding to actual $Ca^{2+}$ changes. The same issue applies to the recently developed RCaMP3, which, despite being the brightest red GECI reported to date, is unsuitable for all-optical applications due to persistent photoswitching under blue light illumination. Additionally, red-shifted $Ca^{2+}$ sensors based on R-GECO (mApple-based), such as jRGECO1a, R-CaMP2 (ref. 21), XCaMP-R[11] and RCaMP3 (ref. 12), inherit not only the photoswitching, but also all suffer from lysosomal accumulation[22], limiting their use in combined imaging and optogenetic experiments.

Developing new red-shifted GECIs is therefore crucial to overcome these limitations. One promising candidate fluorescent protein for the development of red-shifted GECIs is mScarlet, known for its enhanced brightness and minimal photoswitching behavior[23]. First published in 2017, mScarlet quickly gained popularity as a red fluorescent marker protein[23]. This protein consists of 232 amino acids, has a molecular weight of 26.4 kDa and is comparable in size to other RFPs[14,24]; however, it differs substantially in important structural properties such as its chromophore orientation[23]. mScarlet is one of the brightest known RFPs only outperformed recently by mScarlet3 (ref. [25]). mScarlet has a quantum yield of about 70%, while other monomeric RFPs used in GECIs, such as mApple or mRuby, have lower quantum yields than mScarlet[10,14,23]. All these attributes make mScarlet an excellent candidate for use in optogenetic tools and biosensors; however, possibly due to its limited structural similarity to other RFPs, no GECIs utilizing mScarlet have been developed yet.

Here we introduce PinkyCaMP, an mScarlet-based Ca$^{2+}$ indicator, offering improved SNR, brightness, photostability, no photoswitching and an exceptional change in absolute fluorescence upon Ca$^{2+}$ binding. We demonstrate its compatibility with blue-light optogenetics and simultaneous green fluorescence-based neuromodulator and Ca$^{2+}$ imaging through various in vitro and in vivo experiments. Additionally, we validate PinkyCaMP in multicolor two-photon imaging, highlighting its potential for advanced imaging applications.

## Results

### Rational engineering of an mScarlet-based GECI

To develop an mScarlet-based GECI, we took inspiration from the design of previously reported GFP and RFP-based GECIs. As a first step, we screened a large library of circularly permuted mScarlet (cpmScarlet) variants with different lengths and compositions of linkers connecting a calmodulin (CaM)-binding peptide (RS20 derived from R-GECO1)[6] to the N terminus and CaM (also derived from R-GECO1) to the C terminus (Fig. 1a and Extended Data Fig. 1a). Two promising prototypes were identified and designated as PinkyCaMP0.1a (brighter) and 0.1b (more responsive). To further improve their performance, the two prototypes were subjected to directed evolution (Fig. 1b and Extended Data Fig. 1b,c). After 12 rounds of library creation and screening in *Escherichia coli*, which included assaying approximately 6,000 protein variants for brightness and response to Ca$^{2+}$, we had three promising variants designated as PinkyCaMP0.9a (brighter), PinkyCaMP0.9b (more responsive) and PinkyCaMP0.9c (a balance of brightness and responsiveness) (Fig. 1c and Supplementary Fig. 1).

To characterize the biophysical and photophysical properties of PinkyCaMP0.9a,b,c, we first expressed and purified each of the three proteins. Based on this characterization, as well as preliminary cell-based imaging studies (Supplementary Fig. 2), PinkyCaMP0.9c was selected as the best variant for its balance of brightness and responsiveness and renamed as PinkyCaMP (Fig. 1d–f). PinkyCaMP exhibits excitation and emission peaks of 568 nm and 600 nm, respectively, an absorbance peak of 567 nm, and a high $\Delta F/F$ of 15.1 (Fig. 1d,e and Extended Data Table 1). PinkyCaMP has a pKa of 6.83 and 4.24 with and without Ca$^{2+}$, respectively (Supplementary Fig. 3 and Extended Data Table 1), and an apparent dissociation constant ($K_d$) of 54 nM. These results indicate that, relative to other red GECIs, PinkyCaMP has the desirable characteristics of being less sensitive to cytoplasmic pH changes and having higher affinity for Ca$^{2+}$ (Fig. 1f, Supplementary Fig. 3a,d and Extended Data Table 1.) Upon binding to Ca$^{2+}$, the quantum yield increases from 0.03 to 0.48, and the extinction coefficient decreases from 71,000 M$^{-1}$cm$^{-1}$ to 60,000 M$^{-1}$cm$^{-1}$ (Extended Data Table 1). The intrinsic brightness of PinkyCaMP in the Ca$^{2+}$-bound state matches that of jRCaMP1a which is the brightest, yet poorly responsive, red GECI.

### Basic characterization of PinkyCaMP in cultured cells

To assess the brightness of PinkyCaMP in mammalian cells, we expressed PinkyCaMP, jRCaMP1a, jRGECO1a and RCaMP3 in human embryonic kidney (HEK)293T cells. As demonstrated for other GECIs, the inclusion of the bacterial expression plasmid-derived leader sequence (RSET) enhances GECI expression in mammalian cells[12], and the inclusion of a nuclear export sequence (NES) prevents the mixing of slower or biphasic Ca$^{2+}$ kinetics in the nucleus. Thus these two elements were incorporated into the final version of PinkyCaMP for expression and characterization in cultured cells, ex vivo and in vivo. In HEK293T cells, PinkyCaMP showed a mean ± s.e.m. baseline fluorescence of 701 ± 71 arbitrary units (a.u.) and is more than twice as bright as RCaMP3 (292 ± 35 a.u.; $P < 0.0001$, one-way analysis of variance (ANOVA)) and jRGECO1a (294 ± 25 a.u.; $P < 0.0001$, one-way ANOVA) and 14 times brighter than jRCaMP1a (50 ± 4 a.u.; $P < 0.0001$, one-way ANOVA; Fig. 1g).

Given its mScarlet backbone, PinkyCaMP was hypothesized to have no photoswitching under blue light, which is an essential feature for multiplexed imaging in combination with other optogenetic tools. Of note, the original description of RCaMP3 (ref. [12]) did not include data on photoswitching despite its jRGECO1a derived origins. To test this, we stimulated HEK293T cells expressing either sensor with 470 nm light (10-ms pulses, 1 mW mm$^{-2}$). PinkyCaMP displayed no increase in $\Delta F/F$, confirming an absence of photoswitching behavior. In contrast, RCaMP3 exhibited pronounced photoswitching, with $\Delta F/F$ increasing up to 0.43 ± 0.04 (mean ± s.e.m.) during stimulation, followed by a subsequent decay in fluorescence (Fig. 1h).

**Fig. 2 | Characterization of PinkyCaMP in cultured neurons and spectral multiplexing. a**, Tight-seal cell-attached recording (black top trace, right) of a PinkyCaMP-expressing neuron (schematic, left) and simultaneously measured PinkyCaMP signal (pink bottom trace). The number of spikes is indicated above the trace. **b**, Spike count per bout across all samples measured; 87 events in $n = 6$ samples. **c**, PinkyCaMP fluorescence change per detected spike; 87 events in $n = 6$ samples. **d**, Schematic illustration of the primary neuronal culture field stimulation (left) and maximum intensity projection of stimulated neurons (right), on out of 14 samples. The arrow indicates the neuron which calcium transients are shown in **e**. **e**, Calcium transients upon different field stimuli as indicated from a single cell (left) and all neurons in the field of view (FOV). **f**, Calcium transients upon different field stimuli as indicated across all samples. $n = 185$ neurons from 14 samples. **g**, Peak fluorescence change per field stimuli extracted from calcium transients shown in **f**. **h**, SNR per field stimuli extracted from calcium transients shown in **f**. **i**, Transient half-rise time per field stimuli extracted from calcium transients shown in **f**. **j**, Transient half-decay time per field stimuli extracted from calcium transients shown in **f**. **k**, Schematic of expressed proteins and neuronal localization (top) and construct design (bottom). **l**, Single neuron Ca$^{2+}$-imaging (top) with a single pulse field stimulation and electrophysiological current traces (bottom) elicited with different blue light applications. **m**, Emission (solid, pink) and excitation (broken, pink) spectra of Ca$^{2+}$-saturated PinkyCaMP together with the action spectrum recorded from PinkyCaMP-P2A-stCoCHR. $n = 6$ cells. Stimulation light properties (blue and orange shaded areas) are shown. **n**, Representative cell-attached measurement (left) and quantification of the spike probability (right). $n = 5$ cells. **o**, Representative whole-cell voltage-clamp measurement (left). * indicates action potentials. Right: quantification of the generated photocurrent under imaging conditions in comparison to low intensity illumination used for action spectroscopy. Statistics, $P = 0.0312$ $n = 6$ cells. **p**, Ca$^{2+}$-imaging with different stimuli (from left to right): 438 nm LED (blue), field stimulation (black), and no stimulus (gray). The number of stimuli changes from top to bottom: one stimulus, five stimuli, and five stimuli with TTX treatment. The maximum fluorescence change is quantified in the right-most panels. Statistics, one stimulus, $P = 0.028$ and five consecutive stimuli, $P < 0.0001$, $n = 59$ measured in the same cells. Statistics, five consecutive stimuli under 1 µM TTX treatment. $n = 30$. All data are shown as mean ± s.e.m., and all statistical comparisons were performed as Wilcoxon matched-pairs signed-rank tests.

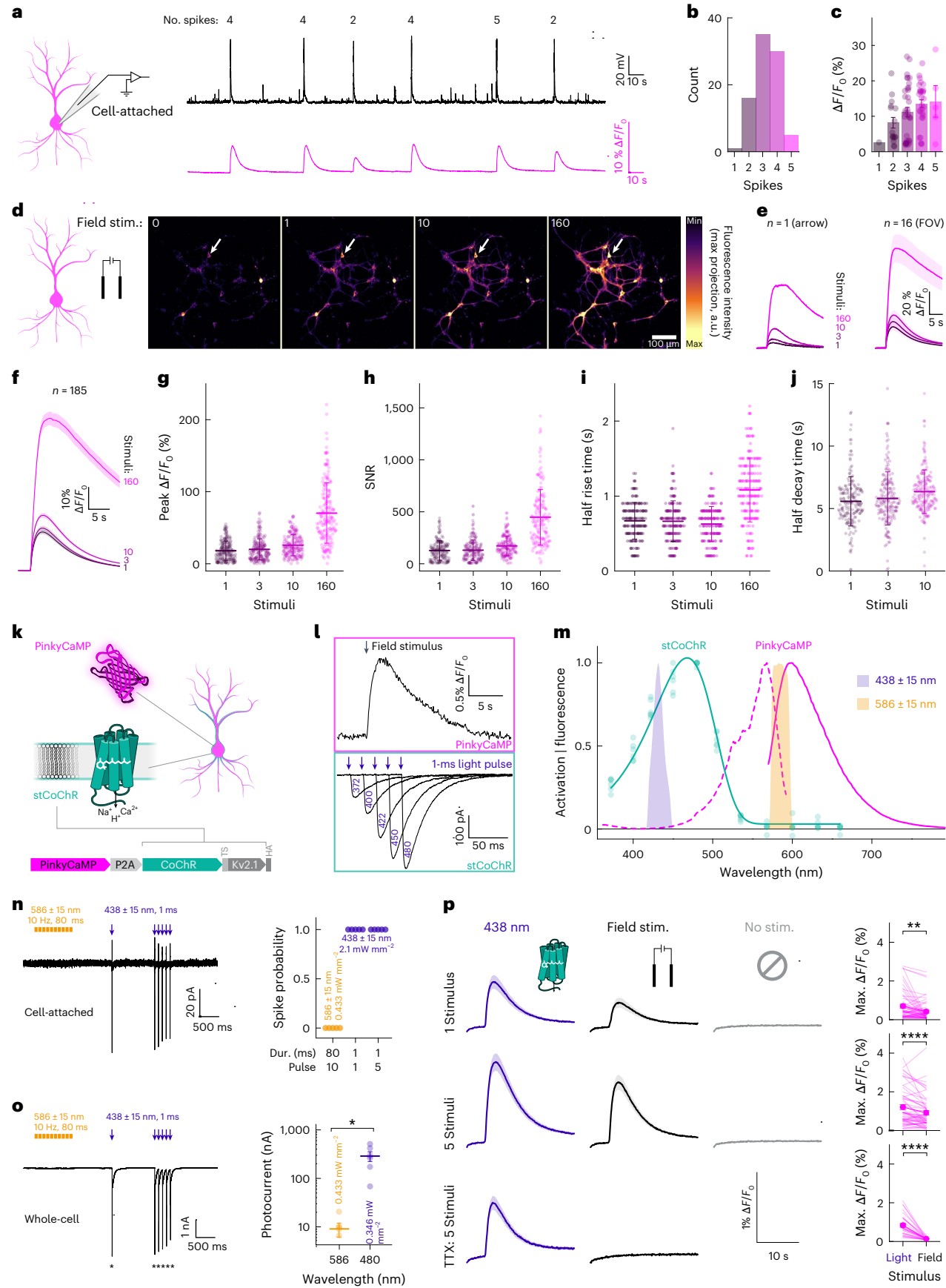

Last, we assessed the photostability of PinkyCaMP in comparison to other GECIs. HEK293T cells expressing the sensors were exposed to continuous 560 nm light (1 mW mm$^{-2}$), and fluorescent decay was recorded and normalized to the peak fluorescence. After more than 13 min of constant light exposure, PinkyCaMP retained nearly 40% of its initial fluorescence, outperforming RCaMP3 (23%) and jRGECO1a (28%) (Fig. 1i). PinkyCaMP's photobleaching half time (410.8 s) is more than double that of RCaMP3 (197.0 s) and much longer than jRGECO1a (222.3 s), highlighting its superior photostability under continuous illumination. As PinkyCaMP absorbs many times more light than RCaMP3 and jRGECO1a, normalizing the time with the unbound extinction coefficient reports the photostability of each sensor for how much light each one absorbed at any point in time (Fig. 1i). PinkyCaMP continued to be the most photostable, with a half photon absorption time of $2.92 \times 10^7$ s M$^{-1}$ cm$^{-1}$, whereas RCaMP3 and jRGECO1a had $6.50 \times 10^6$ s M$^{-1}$ cm$^{-1}$ and $1.37 \times 10^5$ s M$^{-1}$ cm$^{-1}$, respectively. Photobleaching purified protein confirmed the higher photostability of PinkyCaMP to the other sensors, both regardless of extinction coefficient normalization and Ca$^{2+}$ binding state (Supplementary Fig. 2h–k).

### In vitro characterization of PinkyCaMP

Next, we expressed PinkyCaMP in mouse hippocampal cultures. Upon treatment with 4 µM gabazine, the cultures displayed spontaneous bouts of spiking activity, which we recorded simultaneously using cell-attached measurements and calcium imaging in single neurons (Fig. 2a). These network-driven bouts typically consisted of 2–5 spikes (Fig. 2b) and produced an average fluorescence change of $12 \pm 1\%$ $\Delta F/F_0$ per bout (mean ± s.e.m.) (Fig. 2c). To achieve tighter control over neuronal activity, we next monitored PinkyCaMP Ca$^{2+}$ transients during field stimulation (Fig. 2d). The Ca$^{2+}$ transient amplitudes scaled with the number of stimuli (Fig. 2d,e), faithfully reporting neuronal activity at the single-cell level (Fig. 2e). Across all samples, a single field stimulus evoked an average calcium response of $18 \pm 1\%$ $\Delta F/F_0$ (Fig. 2g, representing an upper bound for the response to a single action potential (AP) as the number of underlying APs has not been verified) falling within the range reported for other red calcium indicators and GCaMP6 variants (Extended Data Table 2). Notably, PinkyCaMP exhibited a superior SNR, reaching $129 \pm 7$ mean ± s.e.m.) for a single field stimulus and $172 \pm 7$ for ten field stimuli (Fig. 2h and Extended Data Table 2). PinkyCaMP showed relatively slow kinetics, with a half-rise time of $670 \pm 18$ ms and a half-decay time of $5.6 \pm 0.1$ s (Fig. 2i,j).

Given PinkyCaMP's useful biophysical properties, foremost the non-detectable blue light-mediated photoswitching, we aimed to combine PinkyCaMP with a channelrhodopsin (ChR) in a single bicistronic construct (Fig. 2k) to enable all-optical manipulation and readout of neuronal activity. PinkyCaMP was combined with the trafficking-enhanced and soma-targeted variant of the ChR CoChR[26,27] (stCoChR = CoChR-TS-Kv2.1, Fig. 2k) to allow for blue-light-mediated photocurrent generation, while in parallel enabling imaging of Ca$^{2+}$ activity using PinkyCaMP with orange light (Fig. 2l). Low intensity

activation of CoChR with 1 ms light pulses (Fig. 2l) showed a maximum activity at $468 \pm 2$ nm mean ± s.e.m.) (Fig. 2m). In the following, we used narrow bandpass filtered blue light ($438 \pm 15$ nm) to activate stCoChR and orange light ($586 \pm 15$ nm) for Ca$^{2+}$-imaging via PinkyCaMP (Fig. 2m) to enable multiplexing.

Next, we assessed the electrophysiological impact caused by different light applications. We first measured AP spiking activity in a cell-attached configuration to maintain an unperturbed intracellular environment. While imaging light of 586 nm did not cause any spiking, blue light at 438 nm caused reliable spiking when applied as a single pulse or at 10 Hz (Fig. 2n). Following cell-attached measurements, we measured light-mediated photocurrents in a whole-cell voltage-clamp. Similarly to cell-attached recordings, blue light application caused AP firing (Fig. 2o). The application of orange light used for imaging at 0.433 mW mm$^{-2}$ caused an inward current of $9 \pm 3$ pA (mean ± s.e.m.), which was much smaller than currents measured with 480 nm excitation at the much lower power of 0.346 mW mm$^{-2}$ ($288 \pm 65$ pA; Fig. 2o).

In the next step we combined orange light PinkyCaMP imaging with stCoChR photoexcitation and additionally compared electric field stimuli on the same PinkyCaMP-2A-stCoChR expressing neurons (Fig. 2p). A single 1-ms stimulus (either field stimulation or light application) caused a reliably detectable Ca$^{2+}$ signal, whereas light-evoked stimuli showed higher Ca$^{2+}$ response on average (Fig. 2p). The same neurons showed an increased Ca$^{2+}$ response upon five delivered stimuli (at 10 Hz), while the tendency for a lower response (five field stimuli versus five light stimuli) persisted (Fig. 2p); however, we consistently observed an up-ramping of the PinkyCaMP fluorescence signal during imaging, most noticeable at the start of the imaging session (Fig. 2j–l,p). We anticipated that this small signal increase could be caused by the small photocurrent generated by stCoChR (Fig. 2o) with or without activation of voltage-gated calcium channels paired with the high sensitivity of PinkyCaMP ($K_d$ = 54 nM). Indeed, when we blocked AP generation with TTX, no field stimulation-induced Ca$^{2+}$ spikes were detectable anymore while light application still caused a substantial Ca$^{2+}$ signal (Fig. 2o).

To compare PinkyCaMP to other red fluorescent sensors and GCaMP6f under standardized conditions, we used mouse cortical slice cultures as an in situ model. In all cases, rAAV-mediated neuronal GECI expression resulted in readily identifiable Ca$^{2+}$ transients, as shown for GCaMP6f, jRGECO1a, jRCaMP1a, RCaMP3 and PinkyCaMP (Fig. 3a and Supplementary Fig. 4). These Ca$^{2+}$ transients reflect spontaneous bursts of synchronized network activity, which are typical for slices of that age[28,29]. PinkyCaMP showed the highest baseline brightness ($F_0$; Fig. 3b) and large signal changes ($\Delta F/F_0$) (Fig. 3c). In combination, these two properties give PinkyCaMP the highest absolute signal strengths of all tested red fluorescent GECIs (Fig. 3d), on par with GCaMP6f. PinkyCaMP showed relatively long decay times (Fig. 3e; $\tau$ = 4.9 ± 1.0 s, mean ± s.d.), similar to jRCaMP1a, and a high photostability (Fig. 3f), whereas jRGECO1a and GCaMP6f showed substantial bleaching under these conditions. The stability of PinkyCaMP was also confirmed in 60-min long-term recordings

**Fig. 3 | Comparison of PinkyCaMP and other GECIs in organotypic cultures and acute brain slices. a,** Spontaneous, synchronous calcium transients recorded with GCaMP6f, jRGECO1a, jRCaMP1a, RCaMP3 and PinkyCaMP in cortical slice cultures from mouse (days in vitro (DIV) 13–21 AAV-mediated transduction at DIV 1). **b–g,** Fluorescence brightness ($F_0$) (**b**), relative signal change ($\Delta F/F_0$) (**c**), absolute signal strength ($F_0$aver × $\Delta F/F_0$) (**d**), decay time constants (**e**), photostability (**f**) and assessment of blue-light photoswitching for the indicated variants (**g**). For 60-min recordings and other details see Supplementary Fig. 4. Black boxes indicate 25–75% percentiles and medians, the yellow lines means and the numbers of analyzed slices (**b,f,g**) or the number of observed events (**c–e**; 20 from each slice) are given. Statistical differences (**b–g**) were obtained by Dunn's multiple comparisons after Kruskal–Wallis ANOVAs ($P < 0.05$) and are indicated with *$P \leq 0.05$, **$P \leq 0.01$ and ***$P \leq 0.001$. **h,** Representative confocal image of CA2 neurons in a 300-µm thick acute mouse

brain slice expressing PinkyCaMP; scale bar, 50 µm; $n$ = 23 slices from three mice. **i,** Average $\Delta F/F$ traces of six stimulation pulses at 5 Hz stimulation for PinkyCaMP under low excitation light intensity (left; $n$ = 167 cells). **j,** Average $\Delta F/F$ traces of six simulation pulses for PinkyCaMP and RCaMP3 at 5 Hz stimulation with high excitation light intensity (PinkyCaMP $n$ = 79 cells, RCaMP3 $n$ = 55 cells). **k,** Representative confocal image of RCaMP3 in CA2 neurons; scale bar, 50 µm; $n$ = 20 slices from three mice. **l,** Baseline brightness of cells recorded with 10 Hz stimulation; PinkyCaMP, $106.250 \pm 1.908$; RCaMP3, $52.160 \pm 0.171$. **m,** Maximal $\Delta F/F$ values for 10-Hz stimulation; PinkyCaMP, $0.0557 \pm 0.0055$; RCaMP3, $0.0051 \pm 0.0007$. **n,** SNR at 10 Hz field stimulation; RCaMP3, $33.711 \pm 4.767$; PinkyCaMP, $63.527 \pm 7.880$. Values presented as mean ± s.e.m. (**l–n**). PinkyCaMP $n$ = 60 cells, and RCaMP3 $n$ = 50 cells. Two-tailed Mann–Whitney $U$-test, ****$P \leq 0.0001$.

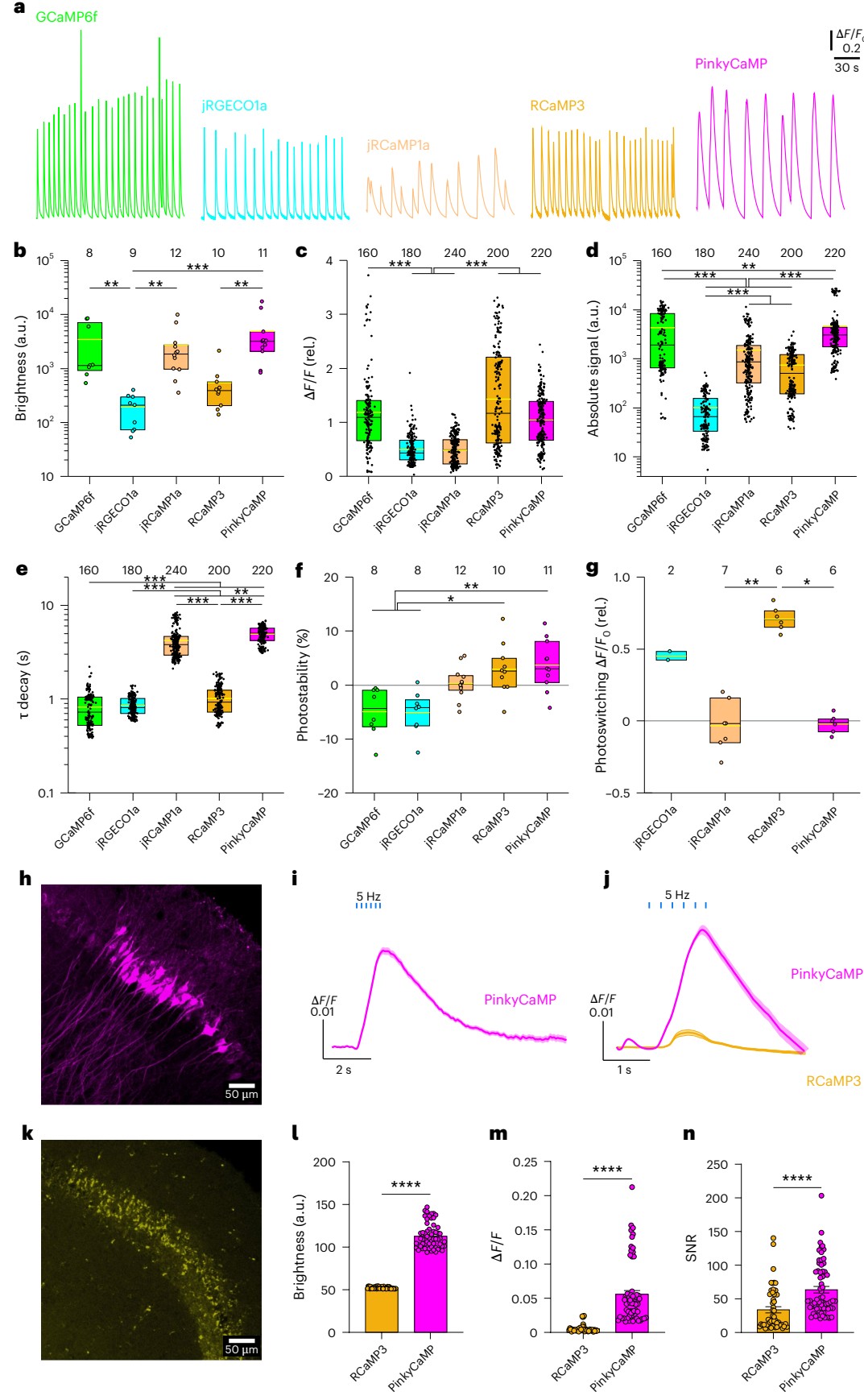

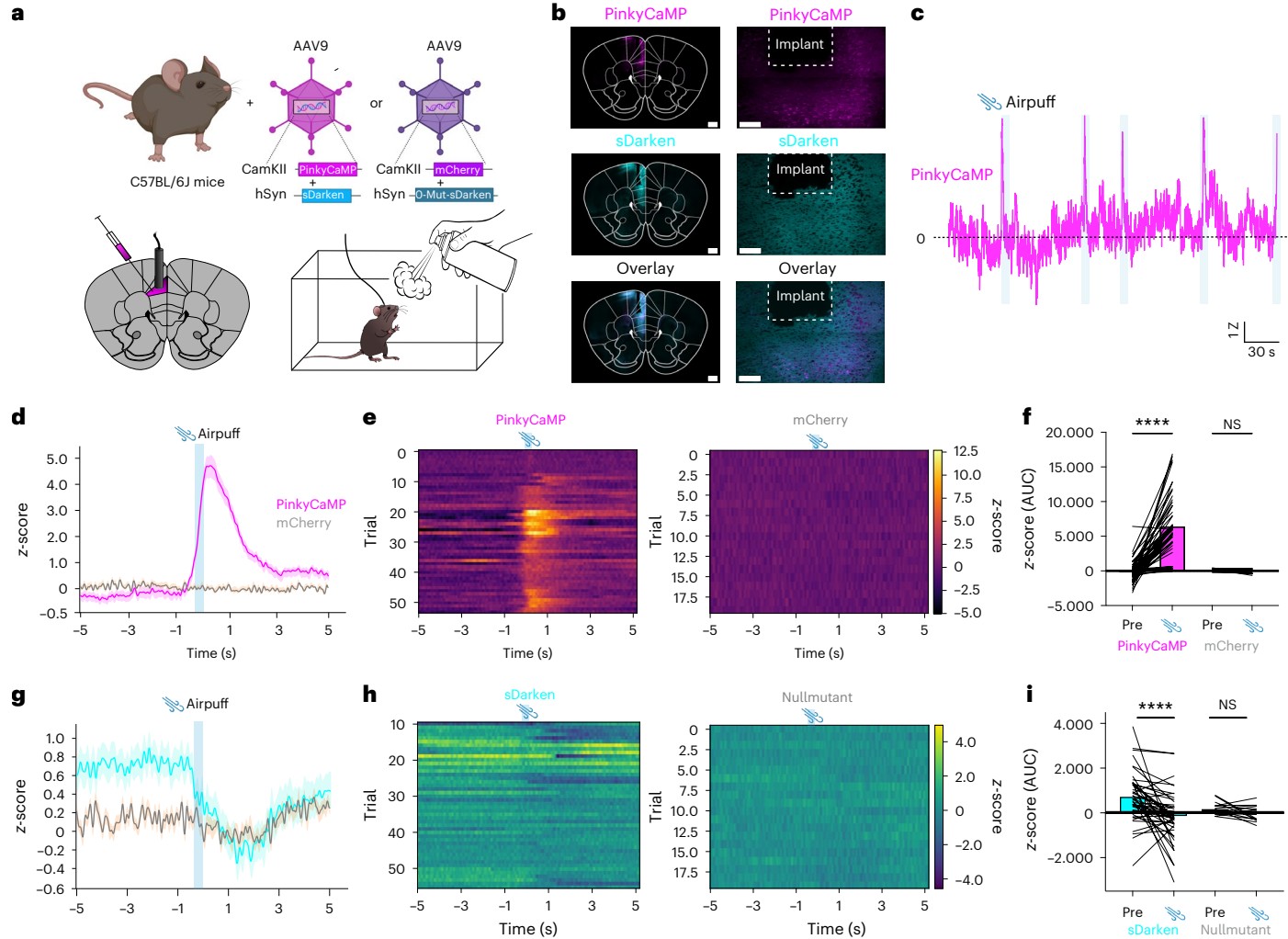

**Fig. 4 | Simultaneous measurement of neuronal activity with PinkyCaMP and serotonin with sDarken. a**, Schematic drawing of AAV injection into the prelimbic area (PrL) of the prefrontal cortex. Experimental setup for airpuff (bottom). **b**, Histology example of PinkyCaMP expression and fiber placement. Scale bars, 500 μm; magnification inset scale bars, 100 μm. **c**, Example traces of PinkyCaMP fluorescence in freely moving mice during an aversive airpuff in their home cage (airpuff). **d**, Averaged PinkyCaMP activity aligned to an aversive airpuff $n = 7$ mice (PinkyCaMP), $n = 4$ mice (control) (mean ± s.e.m.). **e**, Single trial heatmap of PinkyCaMP (54 trials from seven mice) and control (20 trials from four mice). **f**, AUC of PinkyCaMP and mCherry signal before and during the

airpuff. Ordinary one-way ANOVA, PinkyCaMP: mean pre 136 ± 211; mean during: 6,503 ± 579; mCherry: mean pre 124 ± 44; mean during: −47 ± 64, ****$P ≤ 0.0001$. NS, not significant. **g**, Averaged sDarken activity aligned to an aversive airpuff $n = 7$ mice (sDarken), $n = 4$ mice (null-mutant) (mean ± s.e.m.). **h**, Single trial heatmap of sDarken (54 trials from seven mice) and null-mutant (20 trials from four mice). **i**, AUC of sDarken and null-mutant signal before and during the airpuff. Ordinary one-way ANOVA, sDarken: mean pre 749 ± 163; mean during: −170 ± 165; null-mutant: mean pre 169 ± 59; mean during: −27 ± 67, ****$P ≤ 0.0001$. Panel **a** created in BioRender; Renken, K. https://biorender.com/py39z9s (2026).

(Supplementary Fig. 4a–d). Last, we compared how the different red fluorescent GECIs were affected by additional blue-light (470 nm) illumination (Supplementary Fig. 4f–j and Fig. 3g). The PinkyCaMP signal remained stable after blue-light illumination, whereas jRGECO1a and RCaMP3 showed increased signals due to photoswitching with signal amplitudes being similar to synchronous events.

In addition, we performed a more detailed comparison between PinkyCaMP and RCaMP3 using widefield microscopy in acute hippocampal slices (Supplementary Fig. 5a). PinkyCaMP showed strong cytosolic and dendritic expression (Fig. 3h) resulting in low-light recording capability (Fig. 3i,j), with $\Delta F/F$ increasing substantially by field stimulation (Supplementary Fig. 5b,c). While neuropil fluorescence contributed to $\Delta F/F$ ratios, PinkyCaMP's absolute fluorescent signal, photostability and signal-to-noise ratio (SNR) remained high across trials. In contrast, RCaMP3 exhibited severe organellar, presumably lysosomal accumulation (Fig. 3k) and required much higher light intensities (51 times higher than PinkyCaMP) for signal detection;

therefore both sensors were measured with the higher light intensity (Fig. 3l,m and Supplementary Fig. 5d–f). While PinkyCaMP demonstrated photobleaching from the high light intensity, it proved to be much more photostable than RCaMP3 as RCaMP3 failed to produce detectable signals after only a few repeated measurements. In contrast, cells expressing PinkyCaMP produced strong signals for dozens of replicate measurements. Under the higher light intensity conditions, PinkyCaMP maintained strong baseline fluorescence (Fig. 3l). Aside from substantially brighter fluorescence, PinkyCaMP demonstrated a higher $\Delta F/F$ at 10 Hz stimulation than RCaMP3 (Fig. 3m), and a superior SNR of 63.5 ± 7.9 (mean ± s.e.m.) compared to RCaMP3's 33.7 ± 4.8 (Fig. 3n).

## In vivo multiplexed and optogenetic fiber photometry
Next, we assessed the in vivo performance of PinkyCaMP. We chose to express PinkyCaMP in pyramidal neurons of the prefrontal cortex due to their well-established role in innate avoidance and decision-making behaviors[30]. We transduced the medial prefrontal cortex (mPFC), that

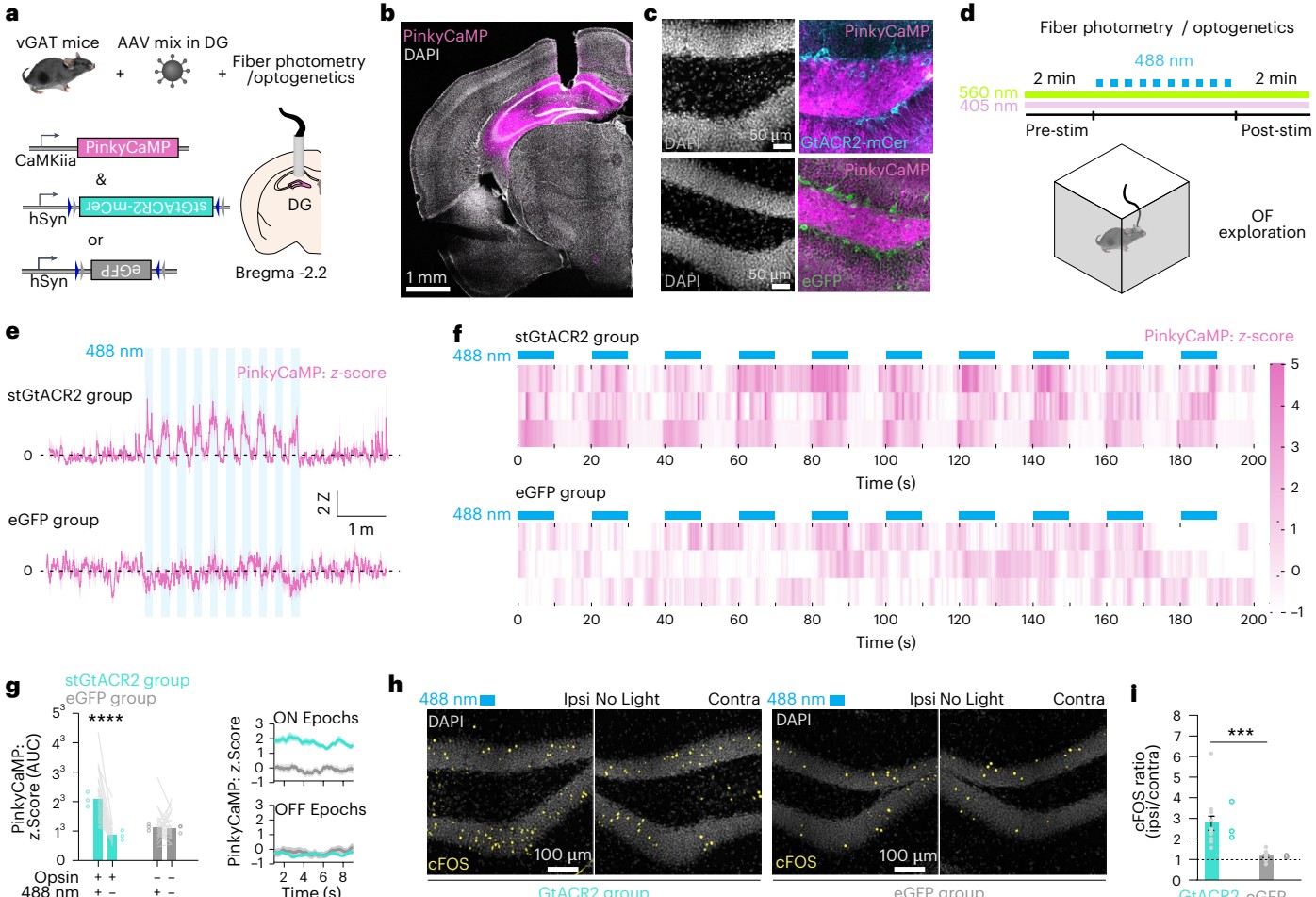

**Fig. 5 | Combining PinkyCaMP with blue-sensitive optogenetics.**
**a**, Experimental design. Fiber photometry recording of GC activity with PinkyCaMP. To drive GC activity, stGtACR2–mCerulean was expressed on vGAT$^+$ neurons. eGFP was used as control. **b**, Exemplary immunofluorescent confocal images. Fiber location and PinkyCaMP expression targeting DG. **c**, Magnification of the hilar and GC layer regions of DG. 4,6-diamidino-2-phenylindole (DAPI) (left), AAV-transduced neurons (right). vGAT expressing stGtACR2 (top) and eGFP (bottom). The experiment was repeated three times per group for **b** and **c**. **d**, PinkyCaMP fiber photometry during OF exploration. PinkyCaMP transients were measured using a 560 nm LED excitation light and a quasi isoemissive wavelength (405 nm) was used to control for motion artifacts. To test for positive photoswitching of PinkyCaMP while simultaneously driving GC activity, the 488 nm light was switched ON/OFF while mice explored the OF (10×, 10 s ON/10 s OFF). **e**, Average PinkyCaMP traces (z-score) during OF exploration in stGtACR2 group (n = 3 mice, top) versus eGFP group (n = 3 mice, bottom). Blue shading indicates the periods when the 488 nm light was on. **f**, Heatmaps of PinkyCaMP

signals (z-score) during 488 nm light optogenetic-driven disinhibition. Blue squares represent the 488 nm light ON epochs. Each row represents an individual mouse. stGtACR2 group (top), eGFP group (bottom). **g**, AUC of PinkyCaMP z-score during 488 nm light ON versus OFF epochs: n = 30 epochs (lines) from three mice (open circles) per group. The 488 nm light substantially increases PinkyCaMP fluorescence due to GC disinhibition (stGtACR2 group) and not due to positive photoswitching (eGFP group). Two-way ANOVA with Šídák multiple comparison test, z-score (AUC) during ON versus OFF epochs stGtACR2 group: ****$P = 2.689 \times 10^{-14}$; eGFP group, $P = 0.8956$. Data are presented as mean ± s.e.m. **h**, Exemplary immunofluorescent confocal images comparing cFOS expression in GCs after OF exploration and optogenetic GC disinhibition between ipsilateral and contralateral to 488 nm light irradiation. **i**, Quantification of cFOS$^+$ GCs and shown as a ratio between the 488 nm irradiated (ipsilateral) versus contralateral side. The 488 nm light increased cFOS expression in the stGtACR2 group, whereas no increase was seen in the eGFP group (unpaired, two-tailed t-test, ***$P = 0.0002$). Data are presented as mean ± s.e.m.

is, prelimbic area (PrL), with either rAAV2/9.CamKII–PinkyCaMP or rAAV2/9.CamKII–mCherry and performed fiber photometry during an aversive airpuff stimulus (Extended Data Fig. 2). PinkyCaMP reported robust Ca$^{2+}$ transients in response to the airpuff, clearly visible in individual traces. In contrast, control mice expressing only mCherry showed no detectable transients (Extended Data Fig. 2d,e). Fluorescence changes were observed in the PinkyCaMP group before and during the airpuff (Extended Data Fig. 2e; $P < 0.0001$, ordinary one-way ANOVA), whereas no airpuff-related changes were detected in the control group ($P > 0.999$, ordinary one-way ANOVA). Subsequently we determined whether PinkyCaMP could also indicate an approach-avoidance conflict in an elevated zero maze (EZM). As before, either PinkyCaMP or mCherry control was virally expressed in the mPFC (Extended Data Fig. 3). Consistent with previous reports, we observed

that Ca$^{2+}$ activity was lowest in the closed arms and increased when the animals transitioned to the open arms. PinkyCaMP-expressing mice showed an increase in fluorescence when transitioning from the closed to the open arm (Extended Data Fig. 3) ($P < 0.01$, ordinary one-way ANOVA), and a decrease when moving back to the closed arm ($P < 0.01$, ordinary one-way ANOVA).

To demonstrate the capability of PinkyCaMP in dual-color fiber photometry, we aimed to simultaneously image the innate avoidance response observed in our previous experiments, along with serotonin release in response to an aversive air puff. We coexpressed rAAV2/9.CamKII–PinkyCaMP together with a serotonin biosensor rAAV2/9.hSyn-sDarken[31] or rAAV2/9.CamKII-mCherry with rAAV2/9. hSyn-nullmutant-sDarken, in the PrL (Fig. 4a,b). As expected, aversive airpuffs elicited robust Ca$^{2+}$ transients in pyramidal neurons, visible

even in individual traces (Fig. 4c). We observed consistent responses to the air puff (Fig. 4d,e), with a substantial increase in PinkyCaMP fluorescence during the stimulus ($P < 0.001$, ordinary one-way ANOVA), whereas mCherry showed no fluorescence changes ($P = 0.9973$, ordinary one-way ANOVA) (Fig. 4f). Additionally, we detected serotonin release, as indicated by a decrease in sDarken fluorescence, coinciding with the increase in neuronal activity (Fig. 4g,h). sDarken fluorescence was reduced during and shortly after the air puff ($P = 0.0001$, ordinary one-way ANOVA), whereas the null-mutant of sDarken exhibited no fluorescence changes before or during the air puff ($P = 0.9119$, ordinary one-way ANOVA) (Fig. 4i).

One caveat of red-emitting GECIs is that the 488 nm light-driven positive photoswitching can be interpreted as an increase in the GECI's signal[28,32]. That is especially concerning in experiments in which the 488 nm light is directed to the same location where the red-emitting GECI is expressed. To test this, we designed an experiment to drive neuronal activity in principal cells via optogenetic disinhibition. Given the strong inhibitory input that granule cells (GCs) receive from the dentate gyrus (DG)[29], as well as the large proportion of GC that is silent[33–35], we expected a strong increase in GC activity upon silencing of GABAergic interneurons. To this end, we injected the left DG of vGAT mice with rAAV2/9.CamKII–PinkyCaMP together with the soma-targeted chloride-conducting opsin GtACR2 (rAAV/DJ.dlox-GtAC R2-ST-mCerulean[36], where 'dlox' indicates a DIO (double-floxed inverted open reading frame) configuration using two different lox sites and an open reading frame in reverse orientation to the promoter) or rAAV2/9.DIO–eGFP as control (Fig. 5a).

This viral mix approach successfully and orthogonally transduced GCs with PinkyCaMP and vGAT cells with the opsin or the control eGFP (Fig. 5b,c). We recorded GC PinkyCaMP transients during open field (OF) exploration with fiber photometry using 560 nm light and 405 nm light as the isoemissive wavelength control. Additionally, a 488 nm laser was switched ON/OFF during the exploration (10 s ON, 10 s OFF, 10×) through the same fiber (Fig. 5c). PinkyCaMP signals were not affected in the eGFP group (**two-way-ANOVA, 488 nm light × Opsin interaction, $P = 0.0015$) (Fig. 5e,f). While in the stGtACR2 group PinkyCaMP signals during the 488 nm ON epochs were notably higher than during the OFF epochs, which was not the case for the eGFP group (Šídák multiple comparison test, $z$-score (area under the curve; AUC) during ON versus OFF epochs stGtACR2 group: $P = 2.689 × 10^{-14}$; eGFP group, $P = 0.8956$) (Fig. 5g). Of note, the lack of increase in the red fluorescent signal during the ON epochs in the eGFP group confirms that indeed, 488 nm light does not lead to positive photoswitching of PinkyCaMP. Finally, we observed a threefold increase in cFOS expression in the ipsilateral DG in the stGtACR2 group, whereas no difference in the eGFP group was observed (***unpaired, two-tailed $t$-test, ratio cFOS (ipsi/contra); $P = 0.0002$) (Fig. 5h,i). The cFOS increase suggests that the observed PinkyCaMP signals report optogenetically driven disinhibition of GC activity. Collectively, these results showcase how PinkyCaMP can be used in single fiber photometry experiments combined with blue-light sensitive opsins without photoswitching artifacts.

## Two-photon and miniscope imaging

Red-shifted $Ca^{2+}$ indicators might be useful to simultaneously carry out $Ca^{2+}$ imaging in different neuronal populations or cell types. Moreover, they can be used in combination with other fluorescent reporters in the green fluorescent range when no red fluorescent alternatives exist. Here, we tested the properties of PinkyCaMP in hippocampal CA1 neurons in awake head-fixed and freely moving mice. We performed stereotactic injections of AAV9–CaMKII–PinkyCaMP or AAV2/1–CaM-KII–PinkyCaMP into dorsal CA1 of the hippocampus. Hippocampal windows were implanted 1 week later and imaging started 6 weeks after injection (Fig. 6a). The PinkyCaMP two-photon excitation curve showed a peak at ~1,100 nm and a secondary peak at below 750 nm (Fig. 6b). As the tuning range of the laser on the awake imaging setup was limited, we recorded $Ca^{2+}$ transients in awake head-fixed mice at 1,040 nm excitation wavelength (Fig. 6c and Supplementary Video 1). PinkyCaMP was very photostable at 1,040 nm excitation (Extended Data Fig. 4a). $Ca^{2+}$ transients were measured in PinkyCaMP-expressing CA1 neurons and were analyzed offline with CaImAn[37] (Fig. 6c–e). We analyzed $Ca^{2+}$ transient parameters and measured a mean $Ca^{2+}$ transient amplitude of $41 ± 0.6\%$ (mean ± s.e.m.) (Fig. 6f). CA1 neurons displayed a transient rate of $3.2 ± 0.2$ min$^{-1}$ (mean ± s.e.m.) (Fig. 6g). We selected 129 representative PinkyCaMP transients and calculated an average rise time of $260 ± 10$ ms (mean ± s.e.m.) (Fig. 6h,i). PinkyCaMP transients were recorded repetitively at 60 and 90 days after injection underscoring the stability of expression and possibility of long-term measurements (Extended Data Fig. 4b–d). Across all recordings and animals ($n = 11$ mice), no $Ca^{2+}$ microwaves were observed, which have previously been reported for GCaMP6 and GCaMP7 (ref. 38).

To assess PinkyCaMP's compatibility with miniscope imaging, we first tested it with a one-photon miniature microscope (Extended Data Fig. 5 and Supplementary Video 2). The one-photon miniscope experiment was performed analogous to the head-fixed two-photon imaging experiment (Fig. 6) with a gradient-index (GRIN) lens placed in the hippocampal window and the one-photon miniscope positioned above. $Ca^{2+}$ transients could be readily detected (Supplementary Video 2). In addition to single-photon miniscope recordings, we imaged PinkyCaMP-expressing CA1 neurons with a miniature two-photon microscope (PhenoSys-Mini2P) equipped with a 1,064 nm excitation laser (Fig. 6j,k). The Mini2P microscope was mounted and aligned to the hippocampal window and mice were voluntarily exploring an OF arena during PinkyCaMP recording (Supplementary Video 3). $Ca^{2+}$ transients were measured in freely moving mice ($n = 3$) (Fig. 6l,m and Extended Data Fig. 6) and individual neuronal activity was allocated to the position of the mouse in the arena (Fig. 6n). These data demonstrate measurements of $Ca^{2+}$ transients with PinkyCaMP in CA1 neurons of awake head-fixed and freely moving mice.

The availability of red and green fluorescent genetically encoded $Ca^{2+}$-indicators enable simultaneous recording of $Ca^{2+}$ transients in separate neuronal populations. To test how suitable PinkyCaMP would be for such measurements, we injected Gad2-Cre mice that express Cre-recombinase in GABAergic interneurons with

---

**Fig. 6 | Awake head-fixed and freely moving recording of PinkyCaMP in the hippocampus. a**, Schematic representation of the experimental timeline for in vivo two-photon imaging. **b**, Two-photon excitation curve of PinkyCaMP at three different constant excitation powers (15, 30 and 50 mW), showing mean gray values of PinkyCaMP (left) and signal-to-background ratio (SBR; right). **c**, Example FOV of PinkyCaMP expression in CA1 pyramidal layer (average intensity projected, 30,3 Hz recording; scale bars, 50 μm) (top). Same FOV, CaImAn-detected active cells are labeled (white ROIs) (bottom). **d**, $Ca^{2+}$ traces of detected cells in **c** of one animal, $n = 115$ cells. **e**, Selected magnified $Ca^{2+}$ traces of five cells from **c** (bottom). **f**, Histogram showing the distribution of transients and their respective amplitude in % of $ΔF/F$ of 10 Hz recordings. All transients ($n = 2,009$) were filtered >20% $ΔF/F$ of $n = 252$ cells and $n = 3$ animals (mean = $41 ± 0.6\%$ $ΔF/F$). **g**, Frequency distribution of cells according to their $Ca^{2+}$ transient frequency per minute of 10 Hz recordings (mean = $3.2 ± 0.2$ min$^{-1}$), filtered for >20% $ΔF/F$, $n = 2,009$ transients, $n = 252$ cells, $n = 3$ animals. **h**, Representative PinkyCaMP transients in black ($n = 129$, >60% $ΔF/F$) and their average plotted in pink. **i**, Frequency distribution of PinkyCaMP transients according to their rise time in milliseconds (mean = $260 ± 10$ (ms), 10–90% interval of onset until peak, filtered for >60% $ΔF/F$, $n = 129$ transients, $n = 3$ animals). **j**, Scheme of freely moving Mini2P imaging during OF exploration. **k**, Average intensity projection of the imaging FOV (left) and CaImAn-detected numbered ROIs (right). Scale bar, 100 μm. **l**, $Ca^{2+}$ traces of 18 labeled neurons in **k** (top) and corresponding $x$ and $y$ positions and velocity during 5 min of OF exploration (bottom). **m**, Exemplary individual $Ca^{2+}$ transients >5 % $ΔF/F$ of cell no. 8. **n**, Trajectory for 5 min of OF exploration. Pink dots indicate $Ca^{2+}$ transient positions of cell no. 8. Panel **a** created in BioRender; Renken K. https://biorender.com/py39z9s (2026).

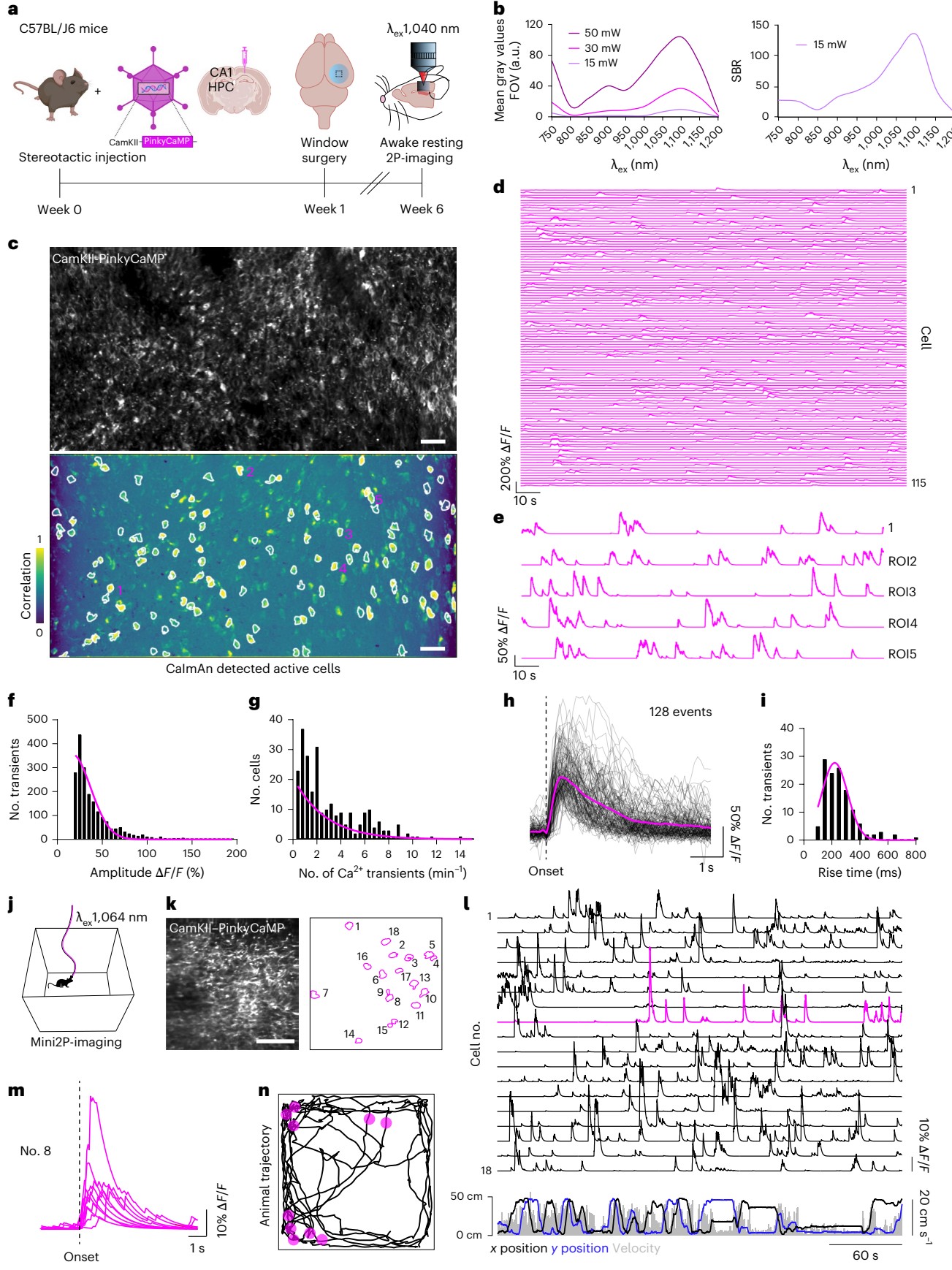

two adeno-associated viruses (AAVs), one encoding a loxP-flanked Syn1-GCaMP8s (*AAV–loxP–Syn–GCaMP8s*) and another encoding CaM-KII–PinkyCaMP (*AAV–CaMKII–PinkyCaMP*) into CA1 of the dorsal hippocampus. Thereby, we achieved specific expression of GCaMP8s in GAD2-positive GABAergic interneurons and expression of PinkyCaMP in a wide range of CA1 neurons, such as pyramidal neurons in pyramidal layer (Extended Data Fig. 7a–d and Supplementary Video 4). To excite both GCaMP8s and PinkyCaMP, we used 980 nm excitation wavelength as a compromise. GCaMP8s recordings from GAD2-positive neurons showed $Ca^{2+}$ transients with long phases of high fluorescence indicative of high activity of inhibitory interneurons in dorsal CA1. PinkyCaMP recordings showed distributed $Ca^{2+}$ transients with fluorescence peak distribution indicative of excitatory pyramidal neurons (Extended Data Fig. 7e–g). The two recording channels for PinkyCaMP and GCaMP8s were well separated with hardly any crosstalk of Pinky-CaMP into the GCaMP8s channel and only minor crosstalk of GCaMP8s signal into the PinkyCaMP channel that might be due to neuropil signal in the manually selected ROIs. We also simultaneously recorded Pinky-CaMP and CcaMP8s in the same CA1 neurons (Extended Data Fig. 8a–f and Supplementary Video 5). Both $Ca^{2+}$ indicators were expressed and recorded in the same cells. Some neurons showed larger Pinky-CaMP than GCaMP8s transients, some neurons with equally large transients and some with larger GCaMP8s than PinkyCaMP transients (Extended Data Fig. 8e,f). These data show that PinkyCaMP can be used for two-color recordings in combination with GCaMP8s primarily in separate cell subsets. This will be highly advantageous for simultaneous two-color $Ca^{2+}$-measurements in different neuronal populations or in combination with other cell types (for example glia).

## Discussion

In conclusion, we developed PinkyCaMP, an mScarlet-based $Ca^{2+}$ sensor. It bridges the performance gap between red-shifted $Ca^{2+}$ sensors and green GECIs such as GCaMP variants. PinkyCaMP is the brightest and most photostable red GECI so far, free from photoswitching, making it ideal for multicolor imaging and optogenetic compatibility. Compared to established red GECIs such as jRGECO1a and jRCaMP1a[10], it is an order of magnitude brighter with a superior SNR, enabling effective low-light imaging and consistent long-term signal detection. Its photostability ensures reliable signal tracking, and its lack of photoswitching makes it an excellent choice for multiplexing, such as dual-color imaging or optogenetic experiments; however, we found that orange light excitation of PinkyCaMP can still cause residual activation of sensitive blue-light opsins, such as CoChR[26,27]. This could be circumvented by using more red-shifted excitation (>590 nm) for PinkyCaMP or further blue-shifted channelrhodopsins for multiplexing conditions[39–41]. As a red GECI, PinkyCaMP operates at wavelengths that typically result in lower tissue scattering, phototoxicity and autofluorescence, which will be in general advantageous for deep-tissue imaging[8,32,42–44]. In contrast to other R-GECO-based GECIs[6,10], PinkyCaMP exhibited minimal lysosomal accumulation and no observable $Ca^{2+}$ microwaves, highlighting its suitability for long-term expression without compromising neuronal health. Due to its high $Ca^{2+}$ affinity, PinkyCaMP exhibits relatively slow kinetics, limiting its ability to resolve fast spiking activity and reducing spike timing precision. This may constrain its use in experiments which require high temporal resolution, such as encoding and decoding analyses. Notably, the relatively slow decay kinetics may lead to temporal integration of signals, potentially masking high-frequency spiking activity. This could result in an underestimation of spike rates or misinterpretation of population dynamics, especially in circuits where precise spike timing carries critical information. However, its high sensitivity makes it well-suited for imaging sparsely active neuronal populations, where reliable detection of individual events is essential. Our results demonstrate that, despite its slow kinetics, PinkyCaMP enables robust detection of activity in pyramidal neurons and DG GCs across various in vivo settings, such as fiber photometry, awake head-fixed two-photon microscopy and freely moving

one-photon, two-photon and miniscope. The slow kinetics are likely a direct consequence of its high $Ca^{2+}$ affinity, which also increases the risk of calcium buffering when overexpressed. This can likely be mitigated by careful titration of viral load and control of expression levels; however, even after long-term expression of 90 days in the hippocampus, Pinky-CaMP recording was possible. Future engineering efforts may focus on accelerating its kinetics to broaden its applicability to fast-firing neurons. As this is the first generation of an mScarlet-based calcium sensor, there is still room for improvement (primarily in its kinetics) before it reaches the temporal resolution of state-of-the-art sensors such as GCaMP8; however, in terms of brightness, photostability and SNR, PinkyCaMP already reaches the performance level of the latest GCaMP variants. The next challenge will be to improve PinkyCaMP's kinetics to match other GECIs. A feasible approach for this could be achieved by exchange the RS20 peptide with either the ENSOP or ckkap peptide, as it has already been carried out for jGCaMP8 (ref. 9). Despite these limitations, Pinky-CaMP enables easy imaging alongside other green GECIs or fluorescent biosensors in multicolor experiments and facilitates multiplexing with optogenetic tools. Simultaneous $Ca^{2+}$ imaging with GCaMP variants and PinkyCaMP in separate neuronal populations is feasible at 980 nm excitation wavelength and can be even optimized by applying a dual excitation wavelength approach (for example 920 nm and 1,100 nm). This advancement broadens the scope for analyzing interactions within neuronal subpopulations, such as pyramidal cells and interneurons, but also in different cell types for example glia and in combination with green fluorescent indicators for which no red fluorescent alternatives are available. Therefore, PinkyCaMP will be highly relevant to address various fundamental biological questions.

## Online content

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

[1]Neuromodulatory Circuits, Institute of Zoology, University of Cologne, Cologne, Germany. [2]Synthetic Biology, University of Bremen, Bremen, Germany. [3]Department of Chemistry, Graduate School of Science, The University of Tokyo, Tokyo, Japan. [4]Neuroimmunology and Imaging Group, German Center for Neurodegenerative Diseases (DZNE), Bonn, Germany. [5]Cellular Neurobiology, Department of Biology and Biotechnology, Ruhr University Bochum, Bochum, Germany. [6]Charité-Universitätsmedizin Berlin, corporate member of Freie Universität Berlin and Humboldt-Universität zu Berlin, Neuroscience Research Center, Berlin, Germany. [7]Charité-Universitätsmedizin Berlin, corporate member of Freie Universität Berlin and Humboldt-Universität zu Berlin, Institute of Cell and Neurobiology, Berlin, Germany. [8]Institute of Pharmacology and Toxicology, University of Zürich, Zurich, Switzerland. [9]System Neurobiology, University of Bremen, Bremen, Germany. [10]Network Dysfunction, German Center for Neurodegenerative Diseases (DZNE), Bonn, Germany. [11]Neural Circuit Computation, German Center for Neurodegenerative Diseases (DZNE), Bonn, Germany. [12]Charité-Universitätsmedizin Berlin, corporate member of Freie Universität Berlin and Humboldt-Universität Berlin, Einstein Center for Neuroscience, Berlin, Germany. [13]Charité-Universitätsmedizin Berlin, corporate member of Freie Universität Berlin and Humboldt-Universität Berlin, NeuroCure Cluster of Excellence, Berlin, Germany. [14]Humboldt-Universität zu Berlin, Bernstein Center for Computational Neuroscience, Berlin, Germany. [15]University of Bonn, Medical Faculty, Bonn, Germany. [16]Neuroscience Center Zürich, University and ETH Zürich, Zürich, Switzerland. [17]CERVO Brain Research Center and Department of Biochemistry, Microbiology, and Bioinformatics, Université Laval, Quebec, Quebec, Canada. [18]These authors contributed equally: Ryan Fink, Shosei Imai. ✉e-mail: omasseck@uni-koeln.de

## Methods

### Animals

All experiments involving animals were carried out according to the guidelines stated in directive 2010/63/EU of the European Parliament on the protection of animals used for scientific purposes. All procedures involving animals were conducted in accordance with the guidelines of the responsible authorities and adhered to the 3R Principles. Experiments at the University of Bremen were approved by Senator für Gesundheit, Frauen und Verbraucherschutz of the Freie Hansestadt Bremen. Experiments carried out at the Ruhr University Bochum and at the German Center for Neurodegenerative Diseases, Bonn were approved by the Landesamt für Natur, Umwelt und Verbraucherschutz. Experiments carried out at the Charité – Universitätzmedizin Berlin were approved by local authorities and the animal welfare committee of the Charité. Experiments performed at the University of Zurich were performed in compliance with the guidelines of the European Community Council Directive and the Animal Welfare Ordinance (TSchV 455.1) of the Swiss Federal Food Safety and Veterinary Office, and were approved by the Zürich Cantonal Veterinary Office.

For spectral multiplexing experiments with CoChR hippocampal neuronal cultures were prepared from P0 to P1 mice (C57BL/6NHsd; Envigo, 044) of either sex. For comparison with other red GECIs cortico-hippocampal slices were prepared from 7–9-day-old CB57BL/6n mice of both sexes. Mice for postmortem tissue removal were obtained from the Animal Facility of the Faculty of Biology and Biotechnology at Ruhr University Bochum, where they were bred and housed under standard conditions with food and water ad libitum under a 12-h light–dark cycle. For the dual-color fiber photometry experiments and acute slice preparations male and female wild-type C57BL/6J mice (The Jackson Laboratory) were used. Animals were raised under standard 12-h light–dark cycles and housed in groups in individually ventilated cages (Zoonlab) under controlled conditions (22 °C ± 2 °C, 50 ± 5% humidity). Food and water were available ad libitum. After surgical implantation, mice were individually housed. Experiments took place during the dark phase, aligning with the animals' primary activity period. For optogenetic disinhibition of GCs and PinkyCaMP fiber photometry recording, surgeries were performed on adult isoflurane-anesthetized vGAT mice (B6J.129S6(FVB)-Slc32a1tm2(cre)Lowl/MwarJ). For chronic in vivo two-photon Ca²⁺ imaging and two-photon miniscope imaging, C57BL/6J mice and Gad2-Cre mice (Gad2^tm2(cre)Zjh/J, 010802, Jackson) were used.

### Molecular biology, protein purification, DNA constructs and availability of reagents

Phusion high-fidelity DNA polymerase (Thermo Fisher Scientific) was used for routine PCR amplification. Restriction endonucleases, rapid DNA ligation kits and GeneJET miniprep kits were purchased from Thermo Fisher Scientific. PCR products and products of restriction digests were purified using agarose gel electrophoresis and the Gene-JET gel extraction kit (Thermo Fisher Scientific). DNA sequences were analyzed by DNA sequence service of Fasmac Co.

To identify the initial prototypes, a portion of a tandem copy of the mScarlet gene (PDB 5LK4, chain A), fused by a linker encoding GGTGGS, was amplified by PCR using three forward and three reverse primers. These primers were encoded for linkers of different lengths and had one NNK codon (Extended Data Fig. 1a). The product was digested by restriction enzymes BglII and EcoRI. Starting from a plasmid encoding R-GECO1 in the pBAD vector, an EcoRI site was introduced into the RS20 to FP linker and a BglII site was introduced into the FP to CaM linker. This plasmid was digested with EcoRI and BglII and the large fragment was ligated with the PCR product to create a library of PinkyCaMP prototypes. The ligation product was used to transform E. coli strain DH10B (Thermo Fisher Scientific) and 400 brightly red fluorescent colonies were picked from approximately 10,000 colonies inspected. Colonies illuminated by yellow light were screened visually using red-tinted goggles. The bacteria was grown in 1 ml of LB

supplemented with 100 µg ml⁻¹ ampicillin and 0.02% L-arabinose at 37 °C overnight, then transferred to room temperature and incubated overnight. Proteins were extracted with B-PER (bacterial protein extraction reagent; Thermo Fisher Scientific) and the fluorescence brightness and the Ca²⁺-dependent response was assayed using a plate reader equipped with monochromators (Tecan). For the directed evolution of PinkyCaMP variants, libraries were generated by the overlap-extension method[47]. These libraries were screened as described above, and the most promising variants selected in each round were used as the templates for the next library creation. To construct the RCaMP3 plasmid, we first amplified the RS20 and cpmApple regions from a plasmid encoding R-GECO1.2, as well as the CaM region from jRGECO1a, with overlapping sequences. Subsequently, these two DNA fragments were combined by overlap-extension PCR. The full-length PCR product was then ligated into the pBAD vector using XhoI and HindIII restriction sites. Finally, the E217D mutation was introduced to generate RCaMP3. The genes encoding PinkyCaMP variants and RCaMP3 in the pBAD vector, which includes a 5' poly-histidine tag, were expressed in E. coli. Cell pellets were lysed with a cell disruptor (Branson), and proteins were purified by Ni-NTA affinity chromatography (Agarose Bead Technologies).

The PinkyCaMP–CoChR bicistronic construct (pAAV–Cam-KII(0.4)–PinkyCaMP–GSG–P2A–CoChR–ts–Kv2.1–HA–WPRE) was created by Gibson assembly[48] using PinkyCaMP, pAAV–hSyn–GCaMP6m–p2a–ChRmine-Kv2.1–WPRE (Addgene, #131004)[49] and pAAV–EF1a–DIO–CoChR-Kv2.1–P2A–mScarlet–WPRE[26] as donors followed by restriction enzyme based sub-cloning (BamHI and HindIII) into a pAAV_CamKII(0.4) vector (Addgene, #198508)[50]. The Cre-dependent stGtACR–mCerulean AAV is available via the Viral Vector Facility of the University of Zurich. pGP–CMV–NES–jRCaMP1a (Addgene, plasmid #61562), pGP–CMV–NES–jRGECO1a (Addgene, plasmid #61563), pAAV–CaMKIIa–mCherry (Addgene, plasmid #114469) and pAAV–Rep/Cap pAAV2/9n (Addgene, plasmid #112865) are all available from Addgene. All PinkyCaMP (Addgene, #232858) and sDarken constructs are available via Addgene (https://www.addgene.org/Olivia_Masseck/; #232861, #232860, #32859, #232858, #232857 and #232856) and via Viral Vector facility of the University of Zurich (v1126, v1197), or can be requested directly from the corresponding author. The PinkyCaMP sequence has been deposited in GenBank under accession number PZ111304.

### rAAV vector production

rAAV2/1.CamKII(0.4)–PinkyCaMP–WPRE and rAAV2/9.CamKII(0.4)–PinkyCaMP–GSG–P2A–CoChR–ts–Kv2.1–HA–WPRE were produced at the Charité Viral Core Facility. All other AAVs were produced in house using the AAV helper-free vector system (Takara Bio). HEK cells (293tsA1609neo, Sigma Aldrich) were co-transfected with endotoxin-free plasmids: pHelper (Takara Bio), pAAV-Rep/Cap (pAAV2/9n, Addgene, plasmid #112865) and pAAV ITR expression vectors containing the gene of interest.

In 500 µl DMEM (Roti cell DMEM high-glucose, sterile, with glutamine, without pyruvate), 12 µg pHelper, 10 µg rep/cap plasmid and 6 µg of the pAAV expression vector were mixed with 120 µl polyethyleneimine (branched, MW ~25,000, Aldrich Chemistry), incubated at room temperature for 15 min and dropwise added to dishes (TC-Dish 150 Standard, Sarstedt) containing HEK cells (70–80% confluency). The medium was replaced the following day with fresh medium.

Three days post-transfection, supernatants were centrifuged (1,500g, 15 min) and AAVanced™ concentration reagent (System Bioscience) was added (4:1 ratio) to the cleared supernatants. After mixing, the solutions were incubated at 4 °C for at least 12 h. The mixtures were then centrifuged (4 °C, 1,500g, 30 min), and the supernatants discarded. The pellets were resuspended in 500 µl DMEM + 10% FBS (Gibco), centrifuged (room temperature, 1,500g, 3 min), and the supernatants removed. The pellets were resuspended in cold, sterile 1× PBS.

For virus titer determination, 1 µl of the virus sample was incubated for 1 h at 37 °C with 120 µl DNase-digest buffer (10 mM Tris-Cl, 10 mM MgCl$_2$, 50 U ml$^{-1}$ DNase I; Roche Diagnostics). The reaction was stopped by adding 5 µl EDTA solution (0.5 M) and incubating at 70 °C for 10 min. The solution was then incubated at 50 °C overnight with 120 µl proteinase solution (1 M NaCl, 1% N-lauroylsarcosine, 100 µg ml$^{-1}$ proteinase K; Roche Diagnostics). The following day, the solution was incubated for 10 min at 90 °C, then 754 µl H$_2$O was added. The sample was diluted (1:200) and virus titers were determined by PCR using a standard curve of diluted plasmid samples.

### Measurement of biophysical properties in purified protein

Following purification the buffer was exchanged to 30 mM MOPS (pH 7.2) with 100 mM KCl. To perform pH titrations, protein solutions were diluted into buffers that contained 30 mM trisodium citrate, 30 mM sodium borate, 30 mM MOPS, 100 mM KCl, 10 mM EGTA and either no Ca or 10 mM CaCl$_2$, and that had been adjusted to pH values ranging from 3 to 12. The protein solutions were mixed with buffers containing 30 mM MOPS (pH 7.2), 100 mM KCl and various concentrations of free Ca$^{2+}$ concentration[51]. Absorbance spectra were measured using the Shimadzu UV1800 spectrometer and molar extinction coefficients were determined as previously reported[52]. Fluorescence quantum yields were measured using a Hamamatsu Photonics absolute quantum yield spectrometer (C9920-02G) using an excitation wavelength of 560 nm.

Excitation and emission spectra were measured with protein solutions consisting of TBS with 10% glycerol (pH 8) containing either 10 mM EGTA or CaCl$_2$ in a plate reader (Infinite 200 Pro, Tecan). Photobleaching time-lapses of purified proteins were performed with 25 nM for each sensor and either 10 mM CaCl$_2$ or EDTA in a closed 0.2-ml Eppendorf PCR tube resting in a water bath using an upright LNscope microscope (Luigs and Neumann) equipped with a ×40 water objective (LUMPLFLN40xW, Olympus), a CMOS camera (ORCA-spark, Hamamatsu), and 560 nm illumination at 57 mW mm$^{-2}$. Irradiance was measured with a calibrated S170C slide power sensor (Thorlabs).

### Characterization of PinkyCaMP in HEK cell culture

HEK293T cells (tsA201 cells, ATCC) were cultured and transfected following the protocol described in ref. 31. Before imaging, cells were washed with TBS supplemented with 2 mM CaCl$_2$. Imaging and time-lapse acquisition was performed with the microscope described above in the purified protein photobleaching, with a ×40 water immersion objective. To assess in vitro brightness, single images were captured with 16 mW mm$^{-2}$. To evaluate photoswitching behavior, transfected HEK293T cells were imaged under the same conditions as described above at a frequency of 5 Hz. Continuous 560 nm illumination was applied throughout the recording. After a 30-s baseline period, a 470 nm laser (1 mW mm$^{-2}$, 10-ms pulses) was applied every 10 s for a total of six pulses. Photostability was assessed in transfected HEK293T cells under continuous illumination with 560 nm light at 1 mW mm$^{-2}$. Cells were tested for *Mycoplasma* contamination on a regular basis.

### Primary dissociated hippocampal neuronal culture and gene delivery

P0 to P1 mice (C57BL/6NHsd; Envigo, 044) of either sex were utilized. Hippocampi were dissected and cells were dissociated by papain digestion followed by manual trituration. Neurons were seeded on glial feeder cells at a density of $1.6 \times 10^4$ cells per cm$^2$ in 24-well plates and maintained in Neurobasal-A supplemented with 2% B27 and 0.2% penicillin–streptomycin (Invitrogen). Neurons were transduced with rAAV ($2.58 \times 10^{13}$ viral genomes (vg) per well) at DIV 1–3 and were recorded between DIVs 14 and 21. Experiments using rAAV2/1.CamKII(0.4)–PinkyCaMP–WPRE (PinkyCaMP-only) or rAAV2/9.CamKII(0.4)–PinkyCaMP–GSG–P2A–CoChR–ts–Kv2.1–HA–WPRE (PinkyCaMP-stCoChR) were performed on an Olympus BX51 upright microscope equipped with a

LUMPlanFL/IR ×40/0.80 W objective (cell-attached and spectral multiplexing experiments) or an UMPlanFL N 20 × 0.5 W (field stimulation with PinkyCaMP-only experiments) objective.

For cell-attached recordings and simultaneous Ca$^{2+}$ imaging of spontaneous neuronal activity, PinkyCaMP-only expressing neurons were constantly perfused (0.5 ml min$^{-1}$) with extracellular solution containing 140 mM NaCl, 2.4 mM KCl, 10 mM HEPES, 10 mM D-glucose, 2 mM CaCl$_2$, 4 mM MgCl$_2$ (pH adjusted to 7.3 with NaOH, 300 mOsm) and 4 µM gabazine (SR-95531). Cell-attached measurements were performed in a tight- or loose-seal configuration with microelectrodes pulled from quartz glass capillaries (3–6 mΩ), filled with above listed solution without synaptic blockers. The number of spikes per bout were manually counted.

All other samples were constantly perfused (0.5 ml min$^{-1}$) with extracellular solution containing 140 mM NaCl, 2.4 mM KCl, 10 mM HEPES, 10 mM D-glucose, 2 mM CaCl$_2$, 4 mM MgCl$_2$ (pH adjusted to 7.3 with NaOH, 300 mOsm) 2 µM NBQX and 4 µM gabazine (SR-95531). Cell-attached measurements followed by whole-cell recordings were performed microelectrodes pulled from quartz glass capillaries (3–6 mΩ), filled with 136 mM KCl, 17.8 mM HEPES, 1 mM EGTA, 0.6 mM MgCl$_2$, 4 mM MgATP, 0.3 mM Na$_2$GTP, 12 mM Na$_2$-phosphocreatine and 50 U ml$^{-1}$ phosphocreatine kinase (300 mOsm), with pH adjusted to 7.3 with KOH. A Multiclamp 700B (Molecular Devices) and Digidata 1550B digitizer (both Molecular Devices) were used to control and acquire electrophysiological recordings as well as light engine LEDs, bipolar field stimulation and camera exposure timing. Electrophysiological data were acquired at 10 kHz and filtered at 3 kHz.

In PinkyCaMP-stCoChR experiments, bipolar 1 ms, 66 mA field stimuli were delivered through a Warner Instruments SIU-102 Stimulus Isolator to custom made electrodes (1-mm platinum wire) in the bath chamber. PinkyCaMP-only expressing neurons were stimulated with a ISO-STIM-II stimulus isolator (NPI Electronic) with bipolar 1-ms 45 V stimuli at 83 Hz.

For action spectra recordings, light from the Lumencor SpectraX23 light engine was filtered with narrow bandpass filters mounted on a FW212C filter wheel (Thorlabs) and delivered to the sample plane using a FM03R cold mirror (Thorlabs) in the epifluorescence beam path. The following filters were used (center wavelength ± 10 nm of 372 nm; Edmund Optics cat no. 12147), 400 nm (cat no. 65071), 422 nm (cat no. 34496), 450 nm (cat no. 65079), 480 nm (cat no. 65084), 505 nm (cat no. 34505), 535 nm (cat no. 65095), 568 nm (cat no. 65099), 600 nm (cat no. 65102), 632 nm (cat no. 65105) and 660 nm (cat no. 86086). The light intensity for each wavelength was calibrated to the same photon flux corresponding to 0.325 mW mm$^{-2}$ at 505 nm. Short light pulses were applied (1 ms) and the wavelength was either changed from UV to red light or vice versa or measurements were taken in both directions per cell. In the latter case, photocurrents were averaged per cell.

In multiplexing experiments with PinkyCaMP-stCoChR, imaging was performed using a Semrock FF605-Di02 dichroic and a 620 LP ET longpass filter (Chroma), while the excitation light from the green-yellow LED of a Lumencor SpectraX23 was filtered with a 586 ± 15 nm bandpass filter (FF02-586/15 Semrock). For stCoChR activation the blue LED of the SpectraX23 filtered with a 438 ± 15 nm bandpass filter (438/29× Lumencor) was used. Regular calcium imaging experiments on PinkyCaMP-only expressing neurons was performed using a Semrock FF593-Di03 dichroic and a Semrock FF01-593/LP longpass filter, while the excitation light from the green-yellow LED of a Lumencor SpectraX23 was filtered with a 575 ± 25 nm bandpass filter (18–4,750 lumencore). The 575 nm light intensities were 0.91 mW mm$^{-2}$ (×40 objective) and 0.99 mW mm$^{-2}$ (×20 objective), measured in the sample plane.

Imaging was performed with a Hamamatsu ORCA-Fire digital CMOS camera (C16240-20UP) at 4 × 4 binning with 80-ms exposure at 10 Hz with 16 bit. Blue light application for stCoChR activation was performed with a duration of 1 ms between the imaging frames.

Light intensities were measured with a calibrated S170C power sensor (Thorlabs). Electrophysiological data were recorded using Clampex v.11.4, while imaging data were acquired using MicroManager (v.2.0)[50]. Illumination spectra were recorded with an M-spectrometer (Thunder Optics). Electrophysiological data were acquired using Clampex.

## Organotypic slice cultures from mouse cortex and transduction

Slice cultures from 7–9-day-old CB57BL/6n mice of both sexes were prepared and maintained according to published protocols[53,54]. After separating the hemispheres, a parasagittal 45° cut was performed from the top of cerebral cortex to the center of the thalamus, the tissue placed in ice-cold Ringer's solution (125 mM NaCl, 2.5 mM KCl, 1.25 mM NaH$_2$PO$_4$, 26 mM NaHCO$_3$, 2 mM CaCl$_2$, 1 mM MgCl$_2$ and 20 mM D-glucose, saturated with 95% O$_2$ and 5% CO$_2$) and cut in 250-µm thick slices using a vibratome (Leica VT1200, 1-mm amplitude, 0.9 mm s$^{-1}$, 15° angle). The slices were temporarily stored in 34 °C Ringer's solution, until both hemispheres were cut. Under sterile conditions, the cortico-hippocampal slices were washed five times with HBSS (Sigma, H9394) and 2–3 slices were placed on one membrane insert (Millicell PICM0RG50, hydrophilized PTFE, pore size 0.4 µm). Slices were supplied with organotypic slice culture medium consisting of MEM (Sigma, M7278), 20% heat-inactivated horse serum (GIBCO/Life Technologies, 26050088) and additionally 1 mM L-glutamine, 0.001 mg ml$^{-1}$ insulin, 14.5 mM NaCl, 2 mM MgSO$_4$, 1.44 mM CaCl$_2$, 0.00012% ascorbic acid and 13 mM D-glucose and cultured at 37 °C and 5% CO$_2$. Transduction was performed at DIV 1 by adding 1 µl AAV suspension (up to fivefold dilution in PBS; Sigma, 806552) to the center of the cortex. A full medium exchange was performed every 2 days. Ca$^{2+}$ imaging experiments were performed at DIV 13–22. Slices were placed in a custom recording chamber (1.5 ml volume) and superfused with Ringer's solution at 1 ml min$^{-1}$ at 24 °C using a peristaltic pump (Minipuls 3, Gilson) and an in-line heater. The chamber was placed under an upright microscope (Axioscope, Zeiss) fitted with a ×10/0.3 water immersion objective (W N-Achroplan, Zeiss). Epifluorescence excitation for all red fluorescent indicators was provided by a collimated 554 nm LED (MINTL5, Thorlabs) using a 560/40 nm excitation filter and a 585 nm dichroic mirror, while fluorescence was collected with a 630/75 nm emission filter (ET-TxRed filter set, Chroma). The light intensity in the focal plane was 0.1 mW mm$^{-2}$ or 0.3 mW mm$^{-2}$. GCaMP6f measurements were performed with a collimated 470 nm LED (M470L4, Thorlabs) using a 470/40 nm excitation filter, a 495 nm dichroic mirror (T495LPXR, AHF) and a 525/39 nm emission filter (BrightLine HC, Semrock). Here the intensity in the focal plane was 0.4 mW mm$^{-2}$ or 0.7 mW mm$^{-2}$. Images were acquired with a sCMOS camera (Orca-Flash 4.0 LT C11440, Hamamatsu) at 10 frames per second (fps) with 16-bit and 512 × 512 pixels (4 × 4 binning) using MicroManager v.2.047. The slices were allowed to equilibrate for 10–20 min before imaging. Standard recordings lasted 10 min, long-term recordings lasted 60 min. Photoswitching recordings (Supplementary Fig. 4) lasted 16 min with three 50-s intervals of additional high-intensity blue-light illumination in the focal plane (4.2 mW mm$^{-2}$) followed by 200 s without blue light.

## In vitro electrophysiology in hippocampal slices

Hippocampal slices were prepared from 16–18-week-old male and female wild-type C57BL/6J mice (The Jackson Laboratory), 3 weeks after intracranial viral injection of either pAAV9–CaMKIIa–PinkyCaMP or control pAAV9–CaMKIIa–RCaMP3. Acute hippocampal slices were prepared as previously described[55]. In brief, sucrose-substituted artificial cerebrospinal fluid (ACSF) (dissection ACSF) and standard ACSF were prepared with filtered (0.22 µm) purified water. ACSF contained 22.5 mM glucose, 125 mM NaCl, 25 mM NaHCO$_3$, 2.5 mM KCl, 1.25 mM NaH$_2$PO$_4$, 3 mM sodium pyruvate, 1 mM ascorbic acid, 2 mM CaCl$_2$ and 1 mM MgCl$_2$. Dissection solution used for slice preparation contained 195 mM sucrose, 10 mM glucose, 25 mM NaHCO$_3$, 2.5 mM KCl, 1.25 mM NaH$_2$PO$_4$, 2 mM sodium pyruvate, 0.5 mM CaCl$_2$ and 7 mM MgCl$_2$. Dissection and standard ACSF were prepared freshly before each experiment and the osmolarity checked to range between 315 and 325 mOsm. Dissection ACSF was chilled on ice and bubbled with carbogen gas (95% O$_2$/5% CO$_2$, resulting in a pH of 7.3) for at least 30 min before slice preparation. A recovery beaker was prepared with a 50:50 mixture of dissection and standard ACSF and warmed to 33 °C. Mice were deeply anesthetized and transcardially perfused with ice-cold carbogenated dissection ACSF for approximately 30–45 s. After decapitation, brains were quickly removed and transferred to ice-cold dissection ACSF in which the hippocampi were dissected free, placed into an agar block and secured to a vibratome slicing platform with cyanoacrylate adhesive. Hippocampal slices were cut at 400 µm, parallel to the transverse plane. Slices were collected from the dorsal and intermediate hippocampus and transferred to the warm, continually carbogenated recovery beaker and allowed to recover for 30 min, after which the beaker was allowed to come to room temperature and left for an additional 90 min before start of the experiment.

Acute hippocampal brain slices were imaged in ACSF bubbled with carbogen gas (95% O$_2$, 5% CO$_2$) at 33 °C and recorded at 50 Hz. Field stimulation was delivered using a glass patch pipette filled with 1 M NaCl, with its tip placed in the stratum oriens of CA2. Stimulation consisted of six 1-ms pulses of 300 µA, applied at varying frequencies using an isolated current stimulator (DS3, Digitimer). PinkyCaMP fluorescence was recorded at low light intensity (0.23 mW mm$^{-2}$) during 2 Hz and 5 Hz field stimulation.

## Surgeries

For stereotactic surgeries, viral injections and fiber-optic cannula implantation for dual-color fiber photometry mice were initially anesthetized with 5.0% isoflurane in oxygen (1 l min$^{-1}$) and positioned in a stereotactic apparatus (Stoelting Co.) with integrated heating. Anesthesia was maintained with 1.5–2.0% isoflurane, monitoring respiration and toe pinch reflex. Carprofen (5 mg kg$^{-1}$) was administered subcutaneously for analgesia and ophthalmic ointment (Bepanthen) was applied to prevent corneal dehydration. After scalp shaving, the area was sterilized with ethanol and iodine, and lidocaine spray was applied. A midline incision was made, and the lambda-bregma distance measured. Holes were drilled above target regions. rAAV9-CaMKII–PinkyCaMP (8.94 × 10$^{10}$ GC per ml) and rAAV9-hSyn-sDarken (5.15 × 10$^{14}$ GC per ml) (or rAAV9-CaMKII-mCherry/AAV9-hSyn-0Mut-sDarken (1.8 × 10$^{13}$ GC per ml) as control) were injected unilaterally into the right mPFC, that is, PrL (prelimbic area of the prefrontal cortex) at three depths (+1.70 mm anterior–posterior (AP), +0.30 mm mediolateral (ML) and −1.85 mm, −1.75 mm and −1.65 mm dorsoventral (DV)). For ex vivo electrophysiology, rAAV9–CaMKII–PinkyCaMP, rAAV9–CaMKII–RCaMP3 were bilaterally injected into hippocampal CA1 (+2.10 mm AP, ±1.30 mm/±1.60 mm ML and −1.85 mm, −1.75 mm and −1.65 mm DV). Following injections, surgical sites were sutured (SMI, 191050). The skull surface was prepared with 37.5% phosphoric acid (Kerr, OptiBond FL kit) for up to 25 s to enhance implant adhesion. Ceramic fiber-optic cannulas (1.25 mm ferrule Ø, 400 µm fiber core Ø, NA: 0.5, 2.5 mm length) were implanted in the right mPFC PrL (+1.70 mm AP, +0.30 mm ML and −1.70 mm DV). The skull was coated with primer and adhesive (Kerr, OptiBondTM FL kit), UV-cured (850 W cm$^{-2}$) and optical fibers were fixed with UV-cured dental cement (Geiz Dental GC 2278). After surgery, iodine ointment was applied to the skin, and mice recovered in clean, heated cages. Implanted animals were housed individually and experiments began 4 weeks after surgery to ensure viral expression.

For combining PinkyCaMP with blue-sensitive optogenetics the left dorsal DG (−2.2 AP, −1.37 ML and −1.9 DV) was injected with a 1:1 virus mix of AAV9–CaMKii–PinkyCaMP–VariantC (9.8 × 10$^{12}$ vg per ml) and AAV-DJ.hSyn.chl.dlox.stGtACR2_mCerulean(rev). dlox-WPRE-hGHbp(A) (8.0 × 10$^{12}$ vg per ml) or AAV9.hSyn.dlox. eGFP(rev).dlox-WPRE-hGHbp(A) (8.2 × 10$^{12}$ vg per ml). A single 400-nl

injection at a rate of 5 nl s$^{-1}$ was performed using a glass capillary and controlled by a microinjector (Nanoject III, Drummond Scientific). A self-made 400-µm core fiber with a 2.5-mm metal ferrule cannula was implanted 0.1–0.2 mm above the injection site on the same surgery day. The ferrule was cemented (Tetric EvoFlow, A1) onto the skull. The mice had at least 3 weeks of recovery before the photometry recordings.

For chronic in vivo two-photon Ca$^{2+}$ imaging mice were anesthetized with an intraperitoneal (i.p.) injection of a mixture of ketamine (0.13 mg g$^{-1}$) and xylazine (0.01 mg g$^{-1}$) for a stereotactic injection of 1 µl of AAV9–CaMKII–PinkyCaMP (8.94 × 10$^{10}$ GC per ml) into right dorsal hippocampus (+2 mm AP, 2.3 mm lateral and −1.4 mm ventral, relative to Bregma) at 0.1 µl min$^{-1}$, using an UltraMicroPump, 34 G cannula and Hamilton syringe (World Precision Instruments). Stereotactic coordinates were taken from ref. 56. Analgesia was carried out with tramadol in the drinking water (0.1 mg ml$^{-1}$) for 3 consecutive days. Hippocampal window surgery followed 1 week after AAV injection and was performed as described previously[53]. For head fixation during in vivo imaging a headpost (Luigs and Neumann) was cemented adjacent to the hippocampal window. Analgesia was carried out with tramadol in the drinking water (0.1 mg ml$^{-1}$) for 3 consecutive days. In vivo imaging was performed after 5 weeks of recovery. Meanwhile mice were habituated to rest head-fixed on a rotating disk in the microscope setup.

### Fiber photometry recordings for dual-color imaging during an aversive airpuff and in the elevated zero maze

In vivo Ca$^{2+}$ signals (PinkyCaMP) and serotonin dynamics (sDarken) were recorded using an RX10x LUX-I/O Processor and Synapse software (TDT). An integrated LED driver controlled three LEDs for excitation: 560 nm (Lx560, TDT) for PinkyCaMP and mCherry, 465 nm (Lx465, TDT) for sDarken and 0Mut-sDarken and 405 nm (LX405, TDT) for the isosbestic control signal. Each wavelength was set to a light intensity of 25–30 µW and modulated at unique frequencies—530 Hz for 560 nm and 330 Hz for 465 nm and 210 Hz for 405 nm. The LEDs were connected to a six-port Fluorescence Mini Cube (Doric Lenses), with output delivered via a fiber-optic patch cord (NA 0.48, 600 µm; Thorlabs) through a rotary joint (RJ1 1 × 1, Thorlabs) to a subject cable secured to the implanted ceramic fiber-optic cannula using an interconnect (ADAL2, Thorlabs). Emitted fluorescence was collected through the same fibers, separated by the filter cube's dichroic mirrors, and detected by integrated photosensors at a sampling frequency of 1,017.25 Hz. Event time stamps for airpuff applications were within the TDT system.

The airpuff, a robust anxiogenic stimulus, elicits a startle response in mice. Testing was performed in each mouse's home cage under bright lighting conditions. Before testing, all cage enrichments (for example, house, nesting material and paper roll) were temporarily removed and promptly returned afterward. Mice were allowed 5 min to habituate to the light, experimental room and adjusted cage setup. A total of eight airpuffs were administered at intervals of at least 1 min using compressed air directed toward the mouse. Airpuff time stamps were recorded and analyzed offline.

The EZM had a white plastic floor with black, infrared-permeable walls surrounding the two closed areas. It featured a 55-cm inner diameter, 45-cm height and 17-cm wall height. The test leverages mice's natural exploratory drive in novel spaces and their aversion to open, elevated areas. The experiment was conducted under bright lighting with additional infrared LED illumination. Mice were initially placed at the entrance of a designated closed area (closed area A) and allowed to explore the maze for 15 min or until they made at least eight transitions between open and closed areas. If a mouse remained in closed area A for the entire session, it received a 5-min break in its home cage, with the subject cable attached. For the second session, mice were placed at the entrance of the opposite closed area (closed area B) and allowed to explore for 15 min or until eight transitions were completed. This protocol effectively encouraged exploratory behavior. Transitions between open and closed areas were tracked offline.

### Photometry recording + optogenetics disinhibition of GCs

Fiber photometry recordings in the dorsal DG were performed using an iFMC6_IE(400-410)_E1(460-490)_F1(500-540)_E2(555-570)_F2(580-680)_S photometry system (Doric Lenses) controlled by the Doric Neuroscience Studio v.6.1.2.0 software. A low-autofluorescence patch cord (400 µm, 0.57 N.A., Doric Lenses) was attached to the metallic ferrule on mouse's head and used to excite PinkyCaMP with 560 nm (30 µW at the patch cord tip – 1 mW mm$^{-2}$ irradiance) while collecting fluorescence emission measured by a photodiode detector (Newport). We used 405 nm as an isoemissive control fluorescence signal. Signals were sinusoidally modulated at 208 Hz and 333 Hz (405 nm and 560 nm, respectively) via lock-in amplification, then demodulated online and low-pass filtered at 12 Hz. Mice were connected to the patch cord 5 min before the OF exploration in a new cage. Mice were placed in the center of the OF arena (50 × 50 cm) for 7.5 min, while PinkyCaMP transients were recorded. A 488 nm light (laser diode, 480 µW at the patch cord tip – 3.8 mW mm$^{-2}$) was alternated (10 × 10 s ON/10 s OFF) after 2 min from the start of the PinkyCaMP photometry recording in the OF. After optogenetic silencing, mice explored the OF arena for a further 2 min. A tailored MATLAB code was used to extract, process and analyze PinkyCaMP signals.

### In vivo two-photon Ca$^{2+}$ imaging

Two-photon imaging was performed using an upright Thorlabs Bergamo II galvo-resonant scanning microscope, equipped with a Ti:sapphire excitation laser (Chameleon Ultra II, Coherent) and a ×10 0.5 NA objective (TL10X-2P, Thorlabs). The laser was operated at 1,040 nm for PinkyCaMP fluorescence excitation. PinkyCaMP fluorescence emission was isolated using a bandpass filter (607/70) and detected using a GaAsP PMT (Hamamatsu). ThorImage software (Thorlabs) was used for microscope control and image acquisition. Image series (1,024 × 512 pixels, ~830 × 415 µm FOV) were acquired at 30.3 Hz.

Two-photon imaging was also performed using an upright Zeiss 7 multiphoton microscope, equipped with an Insight ×3 tunable laser (Spectra-Physics) and a ×10 0.5 NA objective (TL10X-2P, Thorlabs). The laser was operated at 1,040 nm for PinkyCaMP fluorescence excitation, emission was isolated using a bandpass filter (617/73) and detected using a non-descanned detector. Zen blue software was used for microscope control and image acquisition. Image series (512 × 200 pixels, ~460 × 189 µm FOV) were acquired at 10 Hz. PinkyCaMP emission profile imaging was acquired at 6.2 Hz with an imaging FOV of 512 × 168 pixels and ~706 × 231 µm.

### Dual-color two-photon Ca$^{2+}$ imaging

PinkyCaMP and GCaMP8s viruses were diluted to achieve equal virus titer (2 × 10$^{12}$ vg per ml) and 1 µl of Syn.PinkyCaMP × CamKII.GCaMP8s mixture. Cell type specific CA$^{2+}$ imaging: equal virus titer (2 × 10$^{12}$ vg per ml) and 1 µl of CamKII.PinkyCaMP × Syn.FLEX.GCaMP8s mixture. Dual-color two-photon imaging of mice expressing GCaMP8s and PinkyCaMP was performed using a custom built Thorlabs microscope equipped with an 8 kHz resonant scanner, a Coherent Chameleon Ultra II Laser and a ×16 0.8 NA objective (N16XLWD–PF, Nikon). For excitation of both, GCaMP8s and PinkyCaMP, the laser was operated at 980 nm. Red (PinkyCaMP) and green (GCaMP8s) channels were separated by an emission cube set (green, 525/50 bandpass; red, 655/40 bandpass; 562 nm dichroic mirror) and fluorescence signals were collected with GaAsP amplified PMTs (PMT2102, Thorlabs). Image series (512 × 512 pixels, ×2 digital zoom, ~340 × 340-µm FOV) were acquired at 30 fps. Approximately 80 mW of laser power was used during imaging (measured under the objective).

### Miniature two-photon microscope imaging

Two-photon imaging of PinkyCaMP in freely moving mice was performed using a Mini2P miniscope (Mini2P-L, PhenoSys) equipped with 2 kHz large-angle MEMS, a fiber-coupled 1,064 nm Laser (ALCOR

1064, Spark Lasers) and a ×3 0.45 NA objective (D0309, Domilight). PinkyCaMP emission was isolated using a bandpass filter (655/40) and detected using a GaAsP amplified PMT (PMT2102, Thorlabs). Image series (256 × 256 pixels, ×1.5 digital zoom, ~360 × 360-µm FOV) were acquired at 15 fps. Approximately 80 mW of laser power was used during imaging (measured under the objective). The microscope was fixed above the hippocampal window and animals were video monitored at 30 fps (U3-3140CP, IDS) exploring a 50 × 50-cm OF arena.

#### Miniature one-photon microscope imaging

Miniature one-photon microscope images were acquired with a dual-color miniature microscope (nVue 2.0, Inscopix) similar to the head-fixed two-photon imaging. A GRIN lens (1 × 4 mm, Inscopix) was placed above the hippocampal cranial window and the miniature microscope was placed above the GRIN lens. Ca$^{2+}$ traces were extracted with IDPS (Inscopix). In brief, Ca$^{2+}$ imaging videos were cropped around the imaging area and four times spatially down sampled (cutoff of 0.5 and 0.005). Videos were motion corrected in IDPS[57]. Regions of interest (ROIs) were selected manually and projected on the $\Delta F/F$ video and calculated as the mean of the ROI/frame in IDPS.

#### Histology

After ex vivo electrophysiology, acute brain slices were placed in 4% PFA at 4 °C overnight, rinsed in 1× PBS and mounted on Superfrost slides using Imaging Spacers (SecureSeal, Grace Bio-Labs). Coverslips were applied with DAPI-containing mounting medium. All slides were stored at 4 °C until fluorescence microscopy.

Following behavioral dual-color fiber photometry experiments, animals received a lethal i.p. injection of a ketamine/xylazine cocktail (130 mg kg$^{-1}$ ketamine and 10 mg kg$^{-1}$ xylazine) and were transcardially perfused with 1× PBS, followed by 4% PFA. Brains were extracted, stored overnight at 4 °C in 4% PFA, and then transferred to a 30% sucrose solution before sectioning. Brains were frozen and 45-µm coronal sections of target areas were cut using a cryostat. Sections were rinsed in 1× PBS, mounted on Superfrost slides (Thermo Scientific) and coverslipped with DAPI-containing mounting medium (ROTI Mount FluorCare with DAPI, Carl Roth). Viral injection sites and fiber-optic cannula placement were confirmed histologically using a confocal microscope (LSM880, Carl Zeiss) with Zen software. Images were acquired at either ×10 or ×20 magnification and processed in ImageJ.

Following optogenetic disinhibition of GCs 50-µm thick coronal brain sections were collected in 1× PBS using a vibratome (VT1200 S, Leica Biosystems). Four slices were selected in a range of 400 µm from the fiber location. Slices were blocked for 2 h at room temperature in a solution containing 5% bovine serum albumin and 0.3% Triton in PBS. Transduced vGAT neurons were detected by incubating a primary antibody against GFP, which recognizes both the eGFP and mCerulean (chicken a-GFP, 1:1,000 dilution; cat. no. GFP-1010, Aves Labs). Additionally, a primary antibody against cFOS (rat a-cFOS, 1:1,000; cat. no. 226 017, Synaptic Systems) was used. Both primary antibodies were incubated at 4 °C overnight. The sections were washed (3 × 10 min, 1× PBS) and incubated with secondary antibodies (Alexa Fluor 488 donkey anti-chicken, 1:500 dilution; cat. no. 703-545-155, The Jackson Laboratory and Alexa Fluor 647 goat anti-rat, 1:500 dilution; cat. no. 31226, Invitrogen, conjugated in house) for 3 h at room temperature. Finally, the sections were washed again (2 × 10 min, 1× PBS) and incubated with DAPI, diluted in blocking solution with factor (1:10,000, 1 × 10 min) before mounting on microscope slides using Hydromount (cat. no. HS-106, National Diagnostics).

A 2 × 2-tiled image of each DG was taken from both hemispheres in all sections at a confocal laser scanning microscope (Axio Imager LSM 800, Zeiss) using a ×25 oil-immersion objective (i LCI Plan-Neofluar ×25/0.8 IMM Korr DIC M27, Zeiss). Four z-stacks tile images at a resolution (pre-stitching) 1,024 × 1,024-pixel resolution were acquired. The Smart Setup function of the Zen microscopy software (v.6, blue

edition) was used to determine the optimal acquisition settings for the fluorophores used (DAPI, Alexa Fluor 488 (vGAT neurons), mScarlet (PinkyCaMP) and Alexa Fluor 647 (cFOS)). Each channel was acquired sequentially. For the overview image of PinkyCaMP expression, a ×10 air objective (Plan-Apochromat ×10/0.45 M27, Zeiss) was used. Only DAPI and mScarlet channels were used and no z-stack was acquired at 1,024 × 1,024 pixels per image tile.

#### Data analysis, quantification, statistic and reproducibility

Experimenters were not blinded to group allocation during data collection and analysis because group identity was required for optical stimulation parameters, and all analyses were based on objective imaging signals. No statistical methods were used to predetermine sample sizes. For each experiment, sample sizes were guided by similar studies using equivalent imaging, viral expression, and behavioral procedures, as well as our previous experience indicating that these group sizes are sufficient to observe reproducible effects. For all experiments, samples were randomly allocated to experimental groups. To determine biophysical properties of the purified protein fluorescence intensity as a function of pH was then fitted by a sigmoidal function to determine the pKa. For $K_d$ measurements, fluorescence intensities versus [Ca$^{2+}$] were plotted and fitted by a sigmoidal function to calculate the apparent Kd value for the purified protein. Calcium imaging data obtained in HEK cells were analyzed using ImageJ[56]. ROIs were drawn around individual cells in the FOV, with an additional ROI used to measure the background fluorescence. The fluorescent brightness of each cell was calculated by subtracting the mean gray value of the background ROI from that of the cell ROI. Analysis was performed in ImageJ, with ROIs drawn as described above. $\Delta F/F$ values were calculated using the formula: $\Delta F/F = (F - F_0)/F_0$, where $F_0$ represents the fluorescence intensity of the first frame, and $F$ represents the fluorescence intensity at a given time point. Statistical analysis was conducted in Python using a one-way ANOVA and Tukey's post hoc test. To assess brightness fluorescence values were background-subtracted and normalized to the peak fluorescence for each respective sensor. The half-decay time ($\tau 1_{/2}$) was determined by fitting a single-exponential decay curve to the mean fluorescence trace and extrapolating the time point at which the fluorescence decreased to half its initial value. All calculations were performed using Python.

For experiments in cultured neurons $\Delta F/F$ Ca$^{2+}$ traces were extracted from manually selected ROIs using the following equation: $(F - F_0)/F_0$, where $F_0$ is the 3 s median fluorescence before the first stimulus (field stimulation or photoexcitation of CoChR) is applied. For the recording of spontaneous spiking activity, a 2 s median at the beginning of each recording was chosen. Stimuli (type and pulse number) were performed randomly. Statistical analysis was performed with GraphPad Prism 10. Electrophysiological data were analyzed using Clampfit 11 (both Molecular Devices). Cells were excluded from the analysis if the access resistance was above 25 MΩ or if the holding current exceeded 250 pA at −70 mV holding potential. Cells were always patched randomly without any preselection by fluorescence. Calcium imaging data were analyzed using ImageJ[57]. Stimuli (type and pulse number) were performed randomly. Statistical analysis was performed with GraphPad Prism v.10. In field stimulation experiments using PinkyCaMP-only expressing neurons, characterization data analysis was performed using Phyton as follows. For each trace, the peak amplitude was defined as the maximum $\Delta F/F$ value and its corresponding time point. The half-rise time was determined as the time when the trace first reached half of the maximum amplitude measured from stimulus onset. The half-decay time was calculated as the interval between the peak and the time when the signal decayed to half of its maximum. To quantify SNR, baseline noise was estimated as the s.d. of $\Delta F/F$ within a 2-s window preceding the stimulus. The SNR was then calculated as the ratio of the maximum amplitude to this baseline noise level.

For the analysis of Ca$^{2+}$ signals in cortical slice cultures data were processed in Fiji (v.2.16.0)[57]. The images were further binned to

256 × 256 pixels and transformed to 32 bit. First, an ROI was defined by drawing a single polygon across the part of the slice that seemed focused, showed clear fluorescence above background and responses during synchronous network activity. A second region showing only membrane from the inserts was defined to obtain the background signal. The mean intensity value of the background region at $t = 0$ was subtracted from all pixels in all images across the whole video. Then a baseline fluorescence ($F_0$) image was obtained for each stack by averaging ten images at an early time point that showed fluorescence close to baseline (minimal intensity). Last, using the ImageJ image calculator, the stacks were converted to $\Delta F/F_0$ stacks. For further analysis, the mean $\Delta F/F_0$ signal of the ROI was transferred to Clampfit 11.2 (Molecular Devices). Data was collected in Excel Professional Plus 2019 (Microsoft) and statistically analyzed and plotted in OriginPro 2023 (OriginLab Corporation). Not all datasets were normally distributed (Shapiro–Wilk tests, $P > 0.05$) and Kruskal–Wallis ANOVA (two-sided; $P < 0.05$) was performed on all data followed by a Dunn's multiple comparison. Figures were assembled in CorelDraw 2018 (Corel Corporation). For each sensor data were obtained from at least four independent transductions. Videos, which showed a focus shift, movement, unusual event heterogeneity or baseline drifts were not analyzed further. We also excluded single slices with unusually high or low baseline fluorescence intensity ($F_0$) deviating >tenfold from the mean. All slices showed $\Delta F/F_0$ changes, which report on global synchronous network activity (Fig. 3a). The measured intensity changes reflect bursts of epileptiform network activity and they were identified as separate events, if the signal returned to around 50% compared to baseline. The event frequency was determined by counting the number of events in the first 300 s of each video. To avoid undersampling because of slow calcium dynamics or sensor kinetics, only slices with event frequencies between 0.05 to 0.25 Hz were taken into account (Supplementary Fig. 4e).

Quantified brightness values give the mean $F_0$ signal (background-subtracted, see above) of the ROI. The relative signal change is reported as the peak $\Delta F/F_0$ value of the first 20 events that were identified in each slice. The absolute signal strength of these events was calculated by multiplying their peak $\Delta F/F_0$ value with the mean brightness ($F_0$) of the corresponding slice. Time constants $\tau$ of the signal decays were obtained by fitting single exponentials to the decay region. For determining the photostability (bleaching) in each slice (Fig. 3f), the average baseline intensity (offset from exponential fits) after the first four events at the beginning of the recording and after four events close to 10 min recording time was determined. Then the obtained baseline intensities from the beginning were subtracted from the 10 min values and multiplied by 100 (to give %). Photoswitching (Fig. 3g) describes the difference in baseline fluorescence intensity after blue-light illumination (measured within 0.1 s after the LED was turned off) compared to before blue-light illumination (Supplementary Fig. 4g–j). Here, up to three measurements were averaged per slice, however, stimuli that were directly followed by a synchronous event were excluded.

For acute brain slices ROIs were manually drawn around visible somas and a random section of neuropil. For PinkyCaMP, the neuropil signal was scaled by a factor of 0.7 (as described previously[55]) before being subtracted from the somatic signal. To account for photobleaching and artifacts, an exponential decay was fitted to each ROI and subtracted, followed by applying a rolling median with a five-frame window. The maximum $\Delta F/F$ of each cell was recorded and averaged to compare responses at different stimulation frequencies. All data were statistically analyzed with a Shapiro–Wilk test and Mann–Whitney $U$-test in GraphPad Prism.

RCaMP3 recordings required a much higher light intensity (11.83 mW mm$^{-2}$). Both sensors were measured under these conditions during 2-Hz and 5-Hz stimulation. Owing to RCaMP3's neuropil signal magnitude exceeding that of the somas, neuropil subtraction was omitted to prevent the appearance of negative responses.

For the 5-Hz stimulation, we compared the maximum $\Delta F/F$ averaged across cells, brightness (calculated as the mean fluorescence of cells in the first frame) and SNR, defined as the maximum $\Delta F/F$ divided by the mean s.d. of the 1-s pre-stimulus baseline between PinkyCaMP and RCaMP3. Statistical comparisons of these metrics were performed using Shapiro–Wilk tests for normality and Mann–Whitney $U$-tests in GraphPad Prism.

For cFOS Image Analysis Bit-plane Imaris (Oxford Instruments, v.10.2.0) was used to process, analyze and quantify GC cFOS expression. First, the surface creation tool was used to mask the GC layer visualized by DAPI. The parameters for surface creation were a surface grain size of 2 μm and a threshold for absolute intensity of the signal >29.1 (a.u.). This surface was used to mask the cFOS channel, so as to restrict the cFOS$^+$ cells counted to the detected GC layer. Finally, the spot detection algorithm was used on the mask to automatically detect the number of cFOS$^+$ cells. The parameters used were an estimated cell diameter of 8 μm and a quality above 11.8, where quality is a measure of signal intensity. The volume of the GC layer, as well as the number of detected cFOS spots were extracted and further analyzed using R on R-studio (v.4.0.4 and 1.4.1106, respectively). To compute the number of cFOS$^+$ GCs, the GC number per image was estimated using the reported numerical density[55]. The cFOS fraction was calculated by dividing the cFOS$^+$ cells by the estimated GCs per image. Finally, cFOS was reported as a ratio between the 488 nm irradiated (ipsilateral) versus contralateral side.

For data analysis of dual-color fiber photometry, a custom Python script was developed following the guidelines previously outlined[58]. The analysis script is publicly available on GitHub (https://github.com/masseck/FibPho-PinkyCaMP.git) under MIT License.

For the two-photon data analysis, recorded image series were motion corrected using the Python toolbox for Ca$^{2+}$ data analysis CaImAn[37] applying rigid-body registration. Detection of cell bodies and source-separation was performed using the CaImAn algorithm based on constrained non-negative matrix factorization[59]. $\Delta F/F$ Ca$^{2+}$ traces were extracted from detected ROIs using the following equation ($F - F_0$)/$F_0$, where $F_0$ is the minimum eighth quantile of a rolling window of 200 frames. Ca$^{2+}$ imaging traces were processed and analyzed using Gaussian process regression (GPR)[60–62] to obtain a smooth approximation of the fluorescence signal over time. Peaks in the GPR-predicted traces were identified and characterized by fitting an exponentially modified Gaussian function[63]. Metrics such as peak amplitude were extracted for each identified peak. Baseline fluorescence was calculated as the average of the tenth percentile of fluorescence values during the baseline window (10–45 s). For all following distribution analysis, a filter was applied to only select transients with >20% $\Delta F/F$. For each cell the number of Ca$^{2+}$ transients per minute (>20% $\Delta F/F$) were calculated. To ensure reproducibility[63], the analysis pipeline was implemented in Python and details of the computational environment and dependencies are provided. Additionally, 129 representative events were selected by hand and extracted from $\Delta F/F$ Ca$^{2+}$ traces with an amplitude of at least 60% $\Delta F/F$ using Igor Pro (Wavemetrics). Rise time was calculated from these events using the 10–90% time interval of onset until peak of the respective event transient.

For PinkyCaMP two-photon emission profile recordings, mean fluorescence of the whole FOV was extracted using ImageJ Fiji and averaged over time. Signal-to-background ratio was calculated by averaging the mean fluorescence intensities of 6 cells, subtracting the mean background fluorescence, and dividing the result by the s.d. of the background fluorescence. This analysis was performed exclusively for 15 mW, as PMT saturation occurred at higher power levels.

To obtain an animal's trajectory during OF exploration, the 30-fps video was downsampled to 15 fps and scaled to 500 × 500 pixels. The animal's position was tracked from the thresholded image series with MTrackZ in ImageJ Fiji. Velocity was computed from the $x$ and $y$ position over time.

Statistical analyses and graph preparation were carried out using GraphPad Prism v.9 (GraphPad Software). To test for normal distribution of data, D'Agostino and Pearson omnibus normality test was used. Statistical significance for groups of two normally distributed datasets paired or unpaired two-tailed Student's $t$-tests were applied. One-way ANOVA with Šídák's multiple comparison test was performed on datasets larger than two, if normally distributed. If not indicated differently, data are represented as mean ± s.e.m. Chosen sample sizes were similar to commonly used ones in the community. We did not predetermine the sample size. Whenever possible automated data analysis was used.

### Reporting summary

Further information on research design is available in the Nature Portfolio Reporting Summary linked to this article.

### Data availability

DNA sequences are available in Supplementary Information and at GenBank under accession number: BankIt3060018 syntheticPZ111304. DNA plasmids used for viral production have been deposited both on the UZH Viral Vector Facility (https://vvf.ethz.ch/, v1177, v1197) and on Addgene (plasmids #232857–232861). Viral vectors can be obtained either from the UZH Viral Vector Facility or from the Masseck laboratory. Due to the large size of the raw imaging datasets, public deposition is currently not feasible. Raw data can be obtained by emailing the corresponding author. Source data are provided with this paper.

### Code availability

The Fiber Photometry analysis script is publicly available on GitHub (https://github.com/masseck/FibPho-PinkyCaMP.git), MIT License.

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

### Acknowledgements

We thank C. Schreiber, F. Piel, M. Neubauer, M. Dopatka, H. Urbschat, the Charité Viral Core Facility and the Viral Vector Facility Univeristy of Zurich for expert technical assistance. O.A.M. was funded by the Deutsche Forschungsgemeinschaft (DFG; project 408367170 and 548214625) and by the iBehave network (funded by the Ministry of Culture and Science at the State of North Rhine-Westphalia). Work at The University of Tokyo was supported by grants from the Japan Society for the Promotion of Science to T.T. (KAKENHI 21H00273, 23H02101 and 24H02267) and to R.E.C. (24H00489, 24H02267 and 19H05633) and a grant from the National Institutes of Health (RF1NS126102 to PI Shy Shoham). S.I. was supported by the MERIT-WINGS program of the University of Tokyo. Work at the Ruhr University Bochum was supported by the DFG (project 394431587 – FOR2795 to A.R.). Work at the Charité was supported by the DFG, project 184695641 – SFB 958, project 327654276 – SFB 1315, Clinical Research Unit KFO 5023 'BecauseY' / project number 504745852, project 415914819 - FOR 3004, project 431572356 and under Germany's Excellence Strategy – Exc-2049-390688087), by the European Research Council (ERC) under the European Union's Horizon 2020 research and innovation program (BrainPlay Grant agreement no. 810580), by the Federal Ministry of Education and Research (BMBF, SmartAge – project 01GQ1420B) and by the Einstein Foundation Berlin (grant ID EZ-2014-226). Work at the University of Zurich was supported by the ERC under the European Union's Horizon 2020 research and innovation program (grant agreement no. 891959 to T.P.) as well as by the Swiss National Science Foundation (project grant no. 320030E_224301, 310030L_212508 and 320030-236030 to T.P.). N.G., F.F., M.M. and M.F. were supported by the European Union ERC-CoG (MicroSynCom 865618), the DFG (SFB1089 C01, B06; SPP2395) and the iBehave network to M.F. and J.G. (funded by the Ministry of Culture and Science of the State of North Rhine-Westphalia). J.G. was supported by the DFG (SFB1089 B06 and SPP2411). We thank J. C. Paterna and the Viral Vector Facility of the University of Zurich for producing the Cre-dependent stGtACR-mCerulean AAV. pGP-CMV-NES-jRCaMP1a (Addgene, plasmid #61562) and pGP-CMV-NES-jRGECO1a (Addgene, plasmid #61563) were a gift from D. Kim & the GENIE Project, Howard Hughes Medical Institute Janelia Research Campus. H. U. Zeilhofer, University of Zurich, provided the vGAT mouse line. pAAV–CaMKIIa–mCherry was a gift from K. Deisseroth, Stanford University (Addgene, plasmid #114469).

### Author contributions

O.A.M. conceived and supervised the project. S.I. discovered the PinkyCaMP0.1a,b prototypes and performed all of the directed evolution to produce PinkyCaMP as represented in Fig. 1 and Supplementary Fig. 1, under the supervision of T.T. and R.E.C. Together, R.F. (under the supervision of O.A.M.) and S.I. (under the supervision of T.T. and R.E.C.) characterized purified proteins. Together, R.F. and M.K. characterized all variants in HEK cells, brain slices and produced custom made viruses under the supervision of O.A.M.). J.O. cloned, screened and analyzed various circularly permutation of mScarlet under the supervision of O.A.M. J.W. performed the in vitro characterization of PinkyCaMP, designed the bicistronic PinkyCaMP-CoChR with the help of A.K. and performed, analyzed and interpreted all related data (Fig. 2) under the supervision of D.S. Organotypic slice experiments (Fig. 3a–g and Supplementary Fig. 4) were performed by G.L., as well as T.Z. and R.F., and supervised by A.R. Brain slice recordings were carried out by R.F. and V.K. and supervised

by O.A.M. and S.I.H. Dual-color fiber photometry experiments were performed and analyzed by K.R. and O.A.M. P.J.L.-M. was responsible for the experimental design for in vivo photometry and optogenetics disinhibition, stereotaxic surgeries, photometry recording and analysis, paper writing and editing, under the supervision of T.P. A.C. was responsible for tissue preparation and immunofluorescence staining for photometry and optogenetic experiment, confocal imaging and image analysis under the supervision of T.P. In vivo two-photon head-fixed and Mini2P measurements were performed and analyzed by N.G., F.F. and M.M. and supervised by M.F. In vivo one-photon measurements were performed and analyzed by E.S. and F.F. and supervised by M.F. and J.G. All authors contributed equally to the data analysis and writing of the paper.

## Funding

## Competing interests

The authors declare no competing interests.

## Additional information

**Extended data** is available for this paper at https://doi.org/10.1038/s41592-026-03065-2.

**Correspondence and requests for materials** should be addressed to Olivia Andrea Masseck.

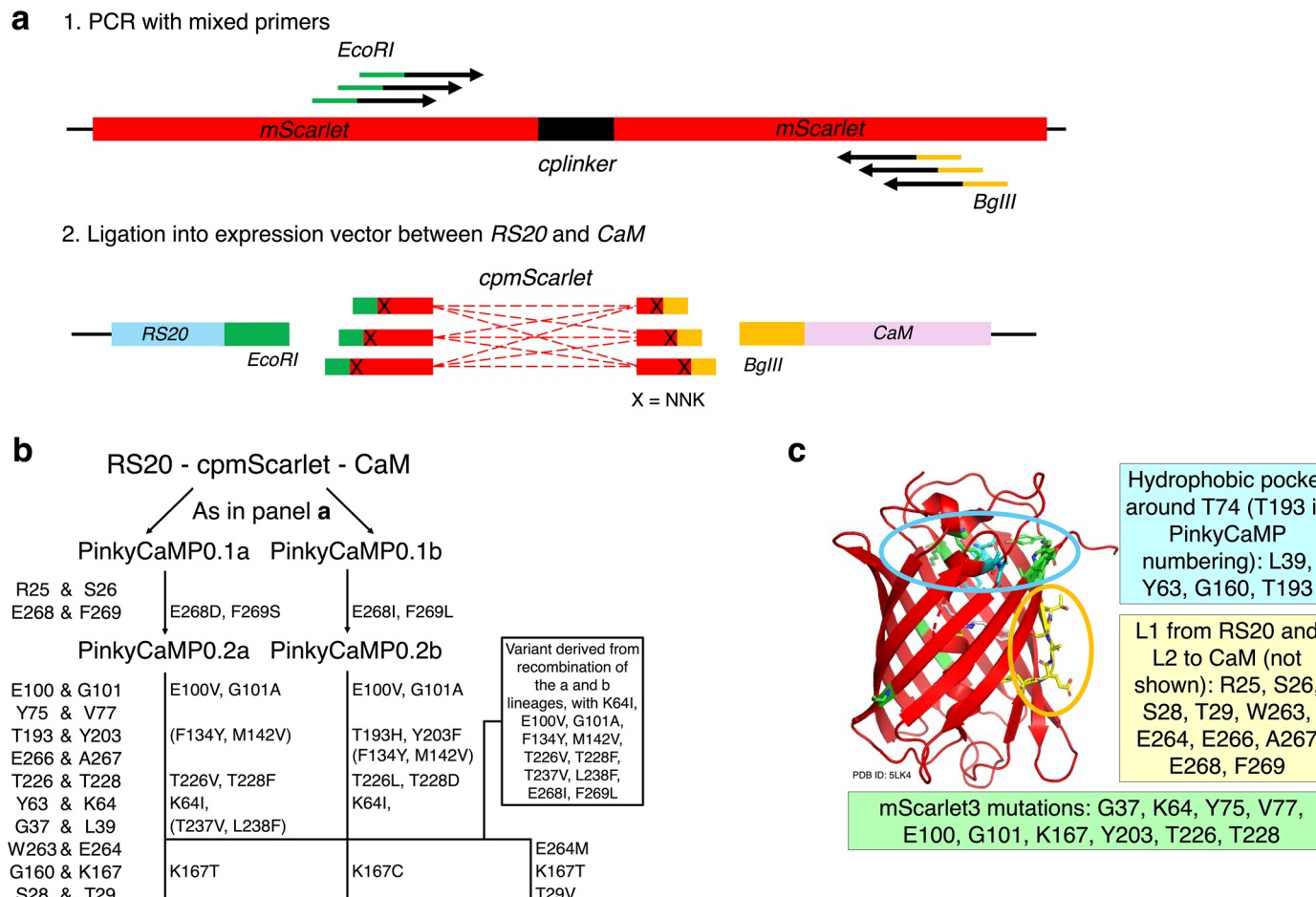

**Extended Data Fig. 1 | The development of PinkyCaMP. (a)** Schematic representation of the method used to initially identify promising prototypes. We generated a library of circularly permuted mScarlet (cpmScarlet) variants with a calmodulin (CaM)-binding peptide (RS20, a variant of CaM-binding region of the smooth muscle form of myosin light chain kinase, derived from R-GECO1)[6] fused to the N-terminus and CaM (also derived from R-GECO1) fused to the C-terminus. This library was expressed in the context of *E. coli* colonies and 400 of the brightest red-fluorescent colonies were picked out of ~10,000 colonies visually inspected. These 400 clones were cultured, the protein was extracted, and evaluated in terms of brightness and response to Ca²⁺. Two promising prototypes were identified and designated as PinkyCaMP0.1a and 0.1b **(b)** Lineage of PinkyCaMP variants. To further improve the performance, PinkyCaMP0.1a and PinkyCaMP0.1b were subjected to directed evolution. As demonstrated for other single FP-based biosensors, the linker regions (that is, RS20 to FP and FP to CaM) have a particularly large impact on biosensor function. We first optimized these residues by using site-saturation mutagenesis and then performing colony based screening and Ca²⁺-response assays on crude protein extracts. For the RS20 to FP linker, R25 and S26 of both PinkyCaMP0.1a and 0.1b were randomized using the overlap-extension method. The two resulting libraries were screened in

parallel. The best performing variants, PinkyCaMP0.2a and 0.2b, both retained the original RS sequence in the RS20 to FP linker, and had the EF of the FP to CaM linker changed to GS and IL, respectively. To further improve the performance, we next focussed our attention on optimization of the cpmScarlet domain and systematically screened libraries created by randomization of 20 additional positions in the protein, two residues at a time. The top one or two winning clones from each library were used as the template for the subsequent library generation. At the completion of this optimization process we had three high-performance variants designated as PinkyCaMP0.9a, PinkyCaMP0.9b, and PinkyCaMP0.9c. **(c)** The position of the 24 residues subjected to site-saturation mutagenesis mapped on the crystal structure of mScarlet (PDB ID: 5LK4)[25]. Ten of these 24 positions are the residues mutated during the development of mScarlet3 (G37, L64, Y75, V77, E100, G101, K167, Y203, T226, T228; colored green)[46]. Further inspired by the development of mScarlet3, which emphasized the hydrophobic pocket around T74 of mScarlet (corresponding to T193 of PinkyCaMP) we also focussed on the four hydrophobic pocket residues L39, Y63, G160, and T193 (colored cyan). Finally, to further optimize residues located at or close to the linker regions, as in the FP to CaM linker (colored yellow).

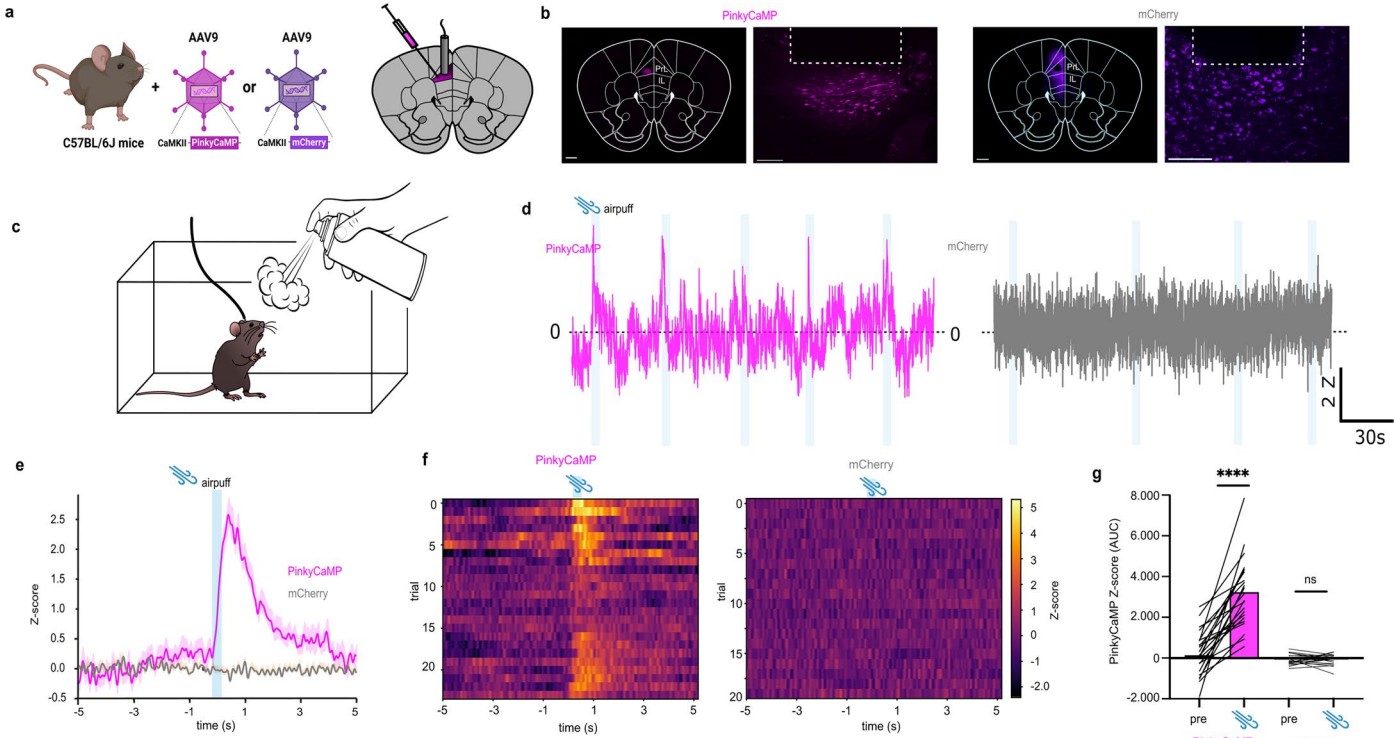

**Extended Data Fig. 2 | Monitoring *in vivo* activity dynamics with PinkyCaMP.**
(**a**) Schematic drawing of AAV injection into the prelimbic area (PrL) of the
right prefrontal cortex. (**b**) Histology example of PinkyCaMP expression and
fiber placement. Scale bar 500 μm, magnification inset scale bar 100 μm. (**c**)
Experimental setup for airpuff application. (**d**) Example traces of PinkyCaMP and
mCherry fluorescence in freely moving mice during an aversive airpuff in their

home cage. (**e**) Averaged PinkyCaMP activity aligned to an aversive airpuff *n* = 3
mice (PinkyCaMP), *n* = 4 mice (control) (mean ± s.e.m.). (**f**) Single trial heatmap
of PinkyCaMP (24 trials from 3 mice) and control animals (20 trials from 4 mice).
(**g**) Area under the curve (AUC) of PinkyCaMP and mCherry signal around the
airpuff. Ordinary one-way ANOVA, PinkyCaMP: mean pre 136 ± 238; mean during:
3.246 ± 335; mCherry: mean pre 77 ± 57; mean during: -69 ± 58, **** $P \leq 0.0001$.

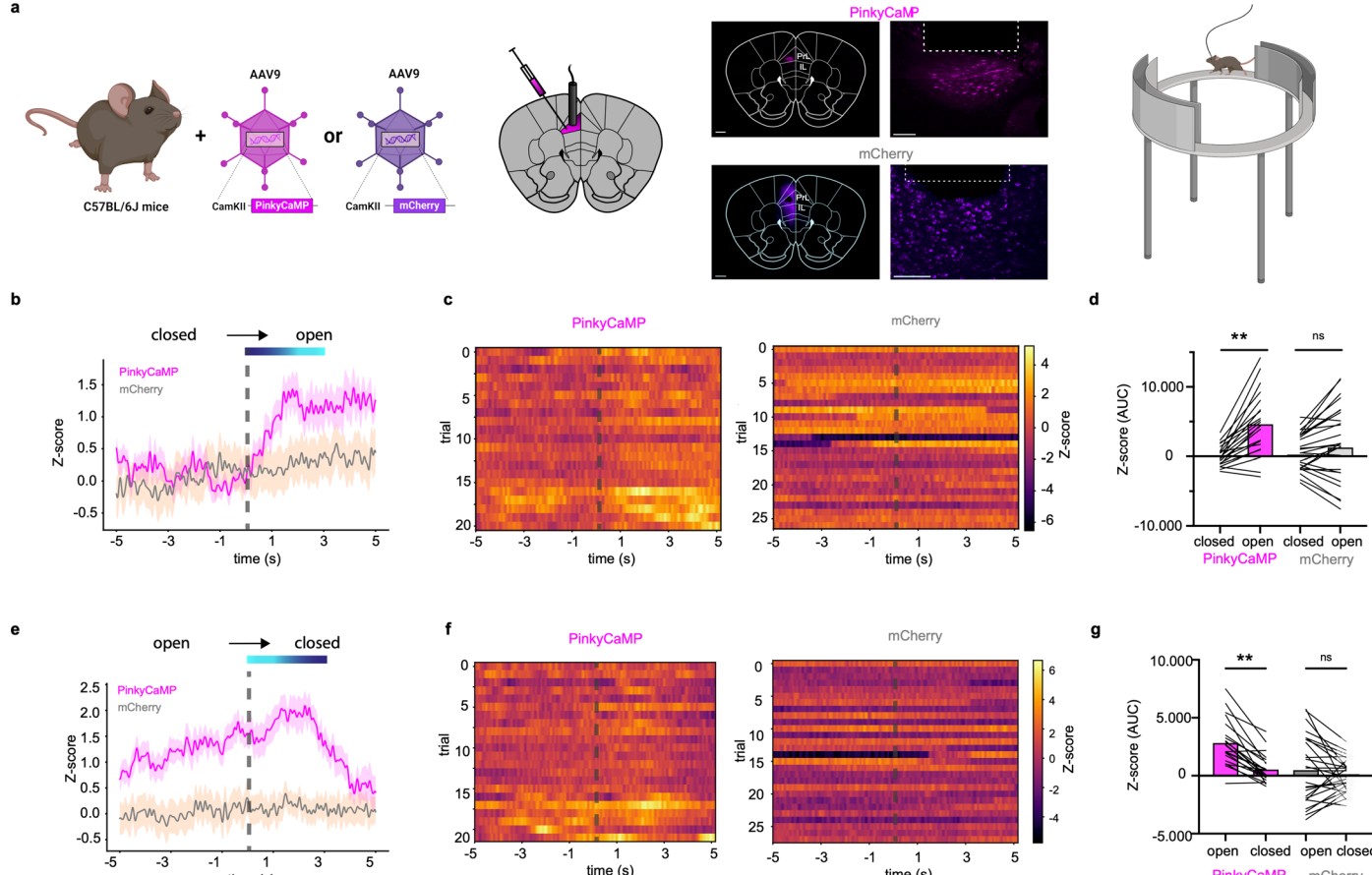

**Extended Data Fig. 3 | Monitoring *in vivo* activity dynamics with PinkyCaMP in EZM.** (**a**) Schematic drawing of AAV injection into the prelimbic area (PrL) of the right prefrontal cortex. Histology example of PinkyCaMP expression and fiber placement. Scale bar 500 μm, magnification inset scale bar 100 μm. Experimental setup of elevated zero maze (EZM) shown on the right. (**b**) Averaged PinkyCaMP activity aligned to the transition from closed to open arm *n* = 3 mice (PinkyCaMP), *n* = 4 mice (control) (mean ± s.e.m.). (**c**) Single trial heatmap of PinkyCaMP (20 trials from 3 mice) and control animals (26 trials from 4 mice) (**d**) Area under the curve (AUC) of PinkyCaMP and mCherry signal before and after

the transition. Ordinary one-way ANOVA, PinkyCaMP: mean closed arm 90 ± 324; mean open arm: 4.708 ± 3.984; mCherry: mean closed 243 ± 557; mean closed: 1.336 ± 1120, ** *P* ≤ 0.01. (**e**) Averaged PinkyCaMP activity aligned to the transition from open to closed arm *n* = 3 mice (PinkyCaMP), *n* = 4 mice (control) (mean ± s.e.m.). (**f**) Single trial heatmap of PinkyCaMP (20 trials from 3 mice) and control animals (26 trials from 4 mice). (**g**) Area under the curve (AUC) of PinkyCaMP and mCherry signal before and after the transition. Ordinary one-way ANOVA, PinkyCaMP: mean open arm 2.870 ± 411; mean closed arm: 576 ± 258; mCherry: mean open arm 530 ± 563; mean open arm: 136 ± 247, ** *P* ≤ 0.01.

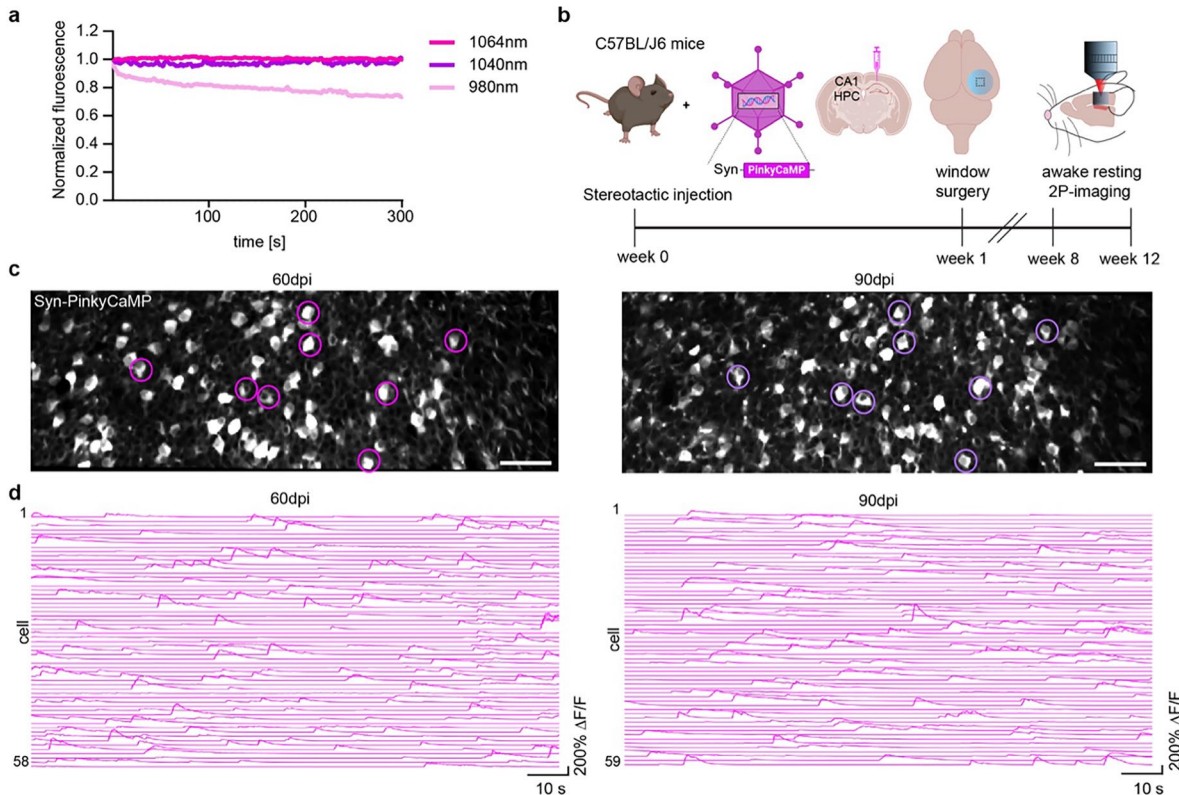

**Extended Data Fig. 4 | Stable PinkyCaMP expression and recording up to 90 days.** (**a**) Normalized fluorescence of PinkyCaMP *in vivo* recordings at 1064, 1040 and 980 nm two-photon excitation (Full field fluorescence average of $n = 3, 2, 3$ mice per wavelength respectively). (**b**) Schematic representation of the experimental timeline for in vivo 2P imaging. (**c**) The same FOV (average intensity projection) in hippocampal CA1 pyramidal layer was imaged at 60 (left) and 90 (right) days post injection (dpi). Some neurons that were visible in both recordings are marked with circles. Scale bar 50 μm. (**d**) CaImAn-detected active neurons and their corresponding Ca²⁺ traces in the same FOV at 60 and 90 dpi.

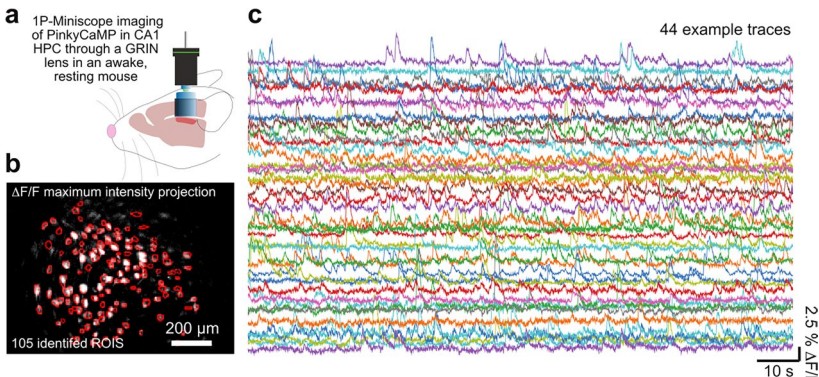

**Extended Data Fig. 5 | Miniature microscope 1-photon imaging of PinkyCaMP expression in dorsal hippocampus. (a)** Cartoon of recording setup. **(b)** Maximum intensity projection and cell maps identified with CNMFE. **(c)** PinkyCaMP $Ca^{2+}$ fluorescence of 44 example neurons recorded in a head-fixed awake mouse with a miniature microscope for red fluorescence imaging.

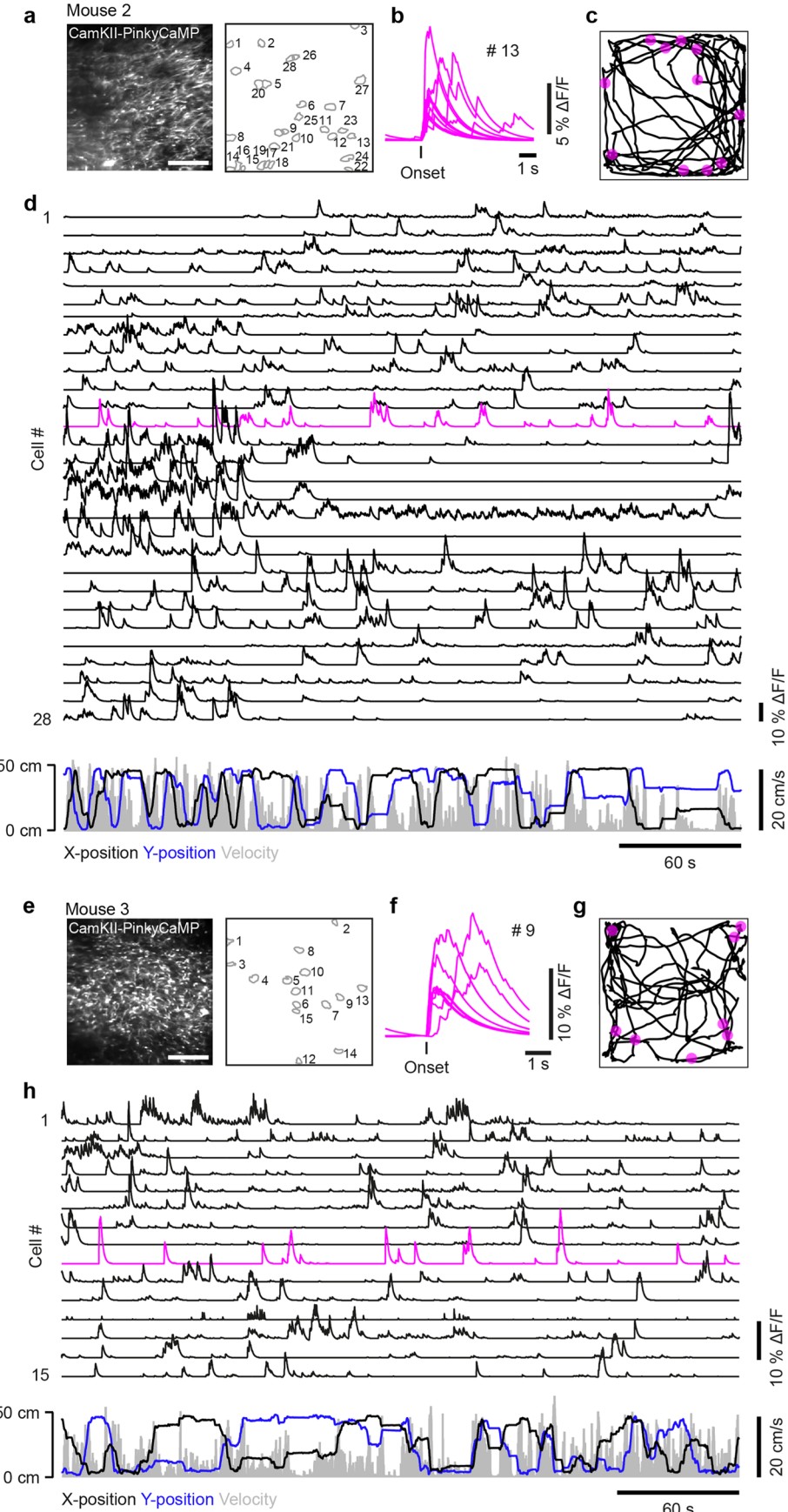

**Extended Data Fig. 6 | See next page for caption.**

**Extended Data Fig. 6 | Mini2P PinkyCaMP imaging in freely moving mice. (a)** Average intensity projection of the imaging FOV (left) and CaImAn-detected numbered ROIs (right), scale bar 100 µm. (**b**) Exemplary individual Ca²⁺ transients > 5 % ΔF/F of cell #13. (**c**) Trajectory of mouse #2 for 5 min open field exploration. Pink dots indicate Ca²⁺ transient positions of cell #13. (**d**) Ca²⁺ traces of the 29 labeled cells in (**a**) (upper panel) and corresponding x/y position and velocity during 5 min open field exploration (lower panel). (**e**) Average intensity projection of the imaging FOV (left) and CaImAn-detected numbered ROIs (right), scale bar 100 µm. (**f**) Exemplary individual Ca²⁺ transients > 5 % ΔF/F of cell # 9. (**g**) Trajectory of mouse # 3 for 5 min open field exploration. Pink dots indicate Ca²⁺ transient positions of cell #9. (**h**) Ca²⁺ traces of the 15 labeled cells in (**e**) (upper panel) and corresponding x/y position and velocity during 5 min open field exploration (lower panel).

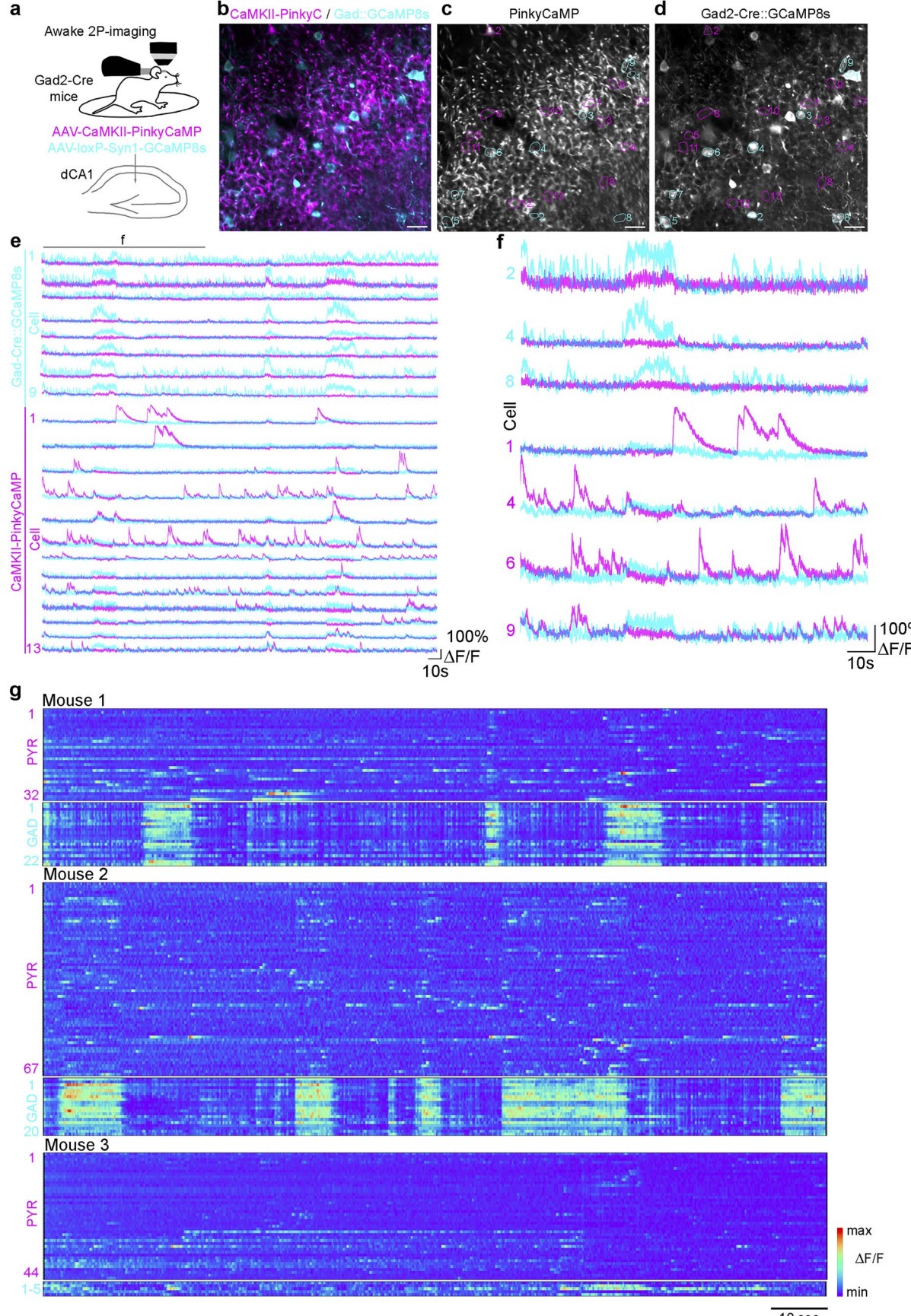

**Extended Data Fig. 7 | See next page for caption.**

**Extended Data Fig. 7 | Simultaneous recording of GCaMP8s in inhibitory and PinkyCaMP in pyramidal neurons.** (**a**) Schematic representation of awake head-fixed two-photon imaging in Gad2-Cre mice injected with *AAV2/1-CaMKII-PinkyCaMP* and *AAV2/1-loxP-Syn1-GCaMP8s* into the dorsal CA1 (dCA1) of the hippocampus. (**b-d**) Exemplary images of CA1 pyramidal neurons expressing PinkyCaMP and GAD2 neurons expressing GCaMP8s as overlayed channels (**b**) and separately (**c**, **d**). Regions of interest (ROIs) of exemplary individual neurons expressing PinkyCaMP are labeled in magenta and ROIs of individual neurons expressing GCaMP8s are labeled in cyan in both channels. (**e**) Ca²⁺-changes were recorded in each ROI for both channels (magenta: red channel, cyan: green channel) and displayed for exemplary cells that showed Ca²⁺ transients during the recording. (**f**) Zoom of a subset of cells from (**e**). (**g**) Exemplary Ca²⁺-traces of pyramidal neurons expressing PinkyCaMP (PYR) and GAD2⁺ neurons expressing GCaMP8s (GAD) in dorsal CA1 of three mice ($n = 153$ PYR, $n = 47$ GAD).

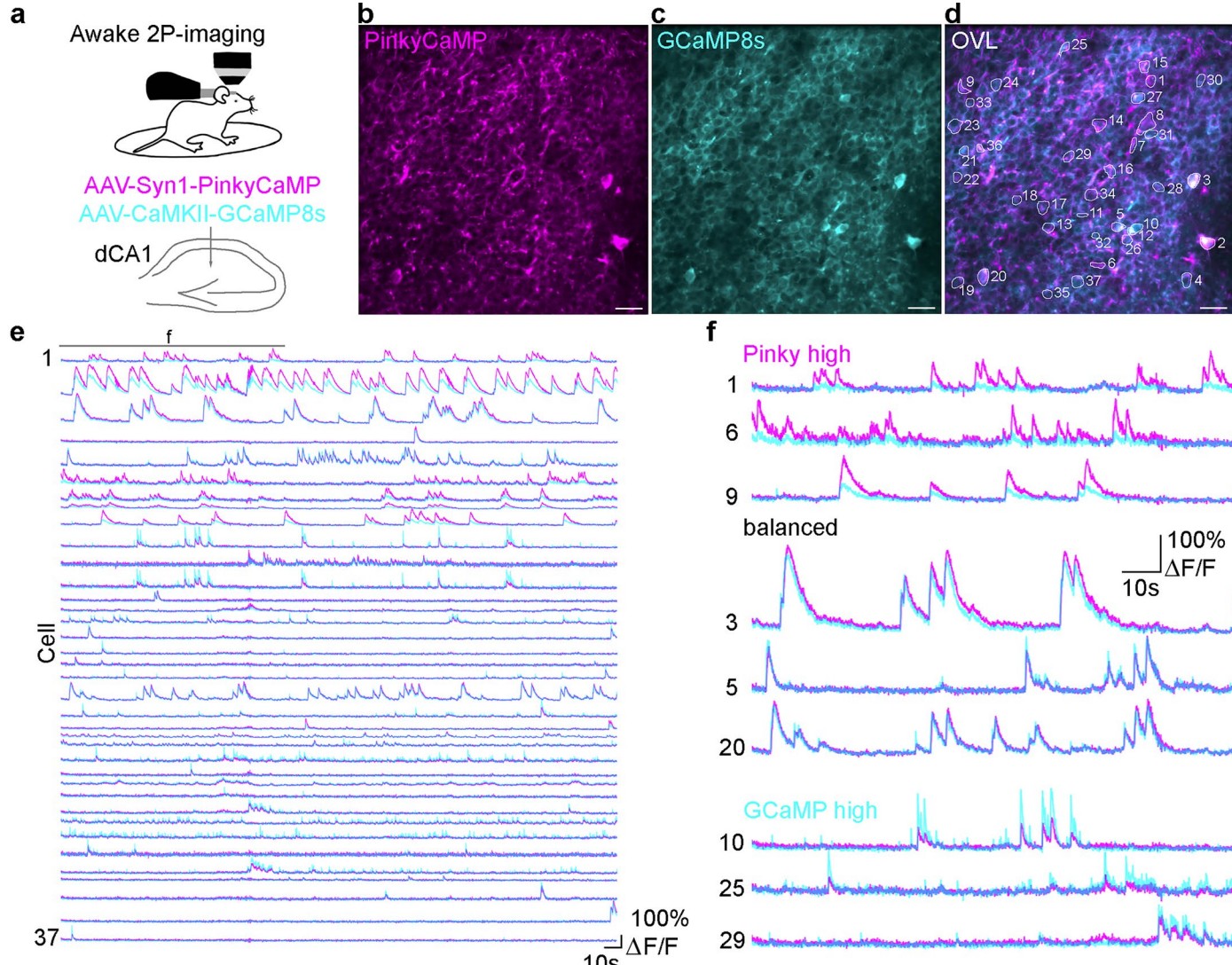

**Extended Data Fig. 8 | PinkyCaMP and GCaMP8s recording in the same hippocampal neurons.** (**a**) Schematic representation of awake head-fixed two-photon imaging in mice injected with *AAV2/1-Syn-PinkyCaMP* and *AAV2/1-CamKII-GCaMP8s* into the dorsal CA1 (dCA1) of the hippocampus. (**b-d**) Exemplary images of CA1 neurons simultaneously expressing PinkyCaMP and GCaMP8s in separate red (PinkyCaMP) and green (GCaMP8s) channels (**b**, **d**) and overlayed (**d**). Regions of interest (ROIs) of exemplary individual neurons expressing PinkyCaMP and GCaMP8s are labeled in (**d**). (**e**) Ca²⁺-changes were recorded in each ROI for both channels (magenta: red channel, cyan: green channel) and displayed for all ROIs labeled in (**d**). (**f**) Zoom of a subset of cells from (**e**). (*n* = 37 neurons from one mouse). The relative transient amplitudes of PinkyCaMP and GCaMP8s observed across neurons likely reflect differences in individual expression levels of the two indicators, which can vary between cells despite co-transfection, rather than systematic differences in their calcium responses.

**Extended Data Table 1 | Biophysical properties of purified proteins**

| Sensor | Condition | Abs$_{max}$ (nm) | p$K_a$ | $K_d$ (nM) | Hill coefficient | $\varepsilon^b$ (M$^{-1}$ cm$^{-1}$) | $\varphi$ | Brightness c | $\Delta F/F^d$ |
|---|---|---|---|---|---|---|---|---|---|
| PinkyCaMP | -Ca$^{2+}$ | 565.5 | 4.24 | 54 | 2.18 | 71,000 | 0.03 | 2.1 | 12.8 |
| | +Ca$^{2+}$ | 565.5 | 6.83 | | | 60,000 | 0.48 | 28.9 | |
| PinkyCaMP0.9a | -Ca$^{2+}$ | 569.5 | 5.66 | 149 | 1.30 | 89,000 | 0.16 | 14.3 | 2.0 |
| | +Ca$^{2+}$ | 568.5 | 6.31 | | | 98,000 | 0.44 | 43.2 | |
| PinkyCaMP0.9b | -Ca$^{2+}$ | 583 | 6.97 | 202 | 1.44 | 85,000 | 0.02 | 1.6 | 14.3 |
| | +Ca$^{2+}$ | 570 | 6.64 | | | 57,000 | 0.43 | 24.4 | |
| RCaMP3 | -Ca$^{2+}$ | 571.5 | 9.21 | 130 | 1.81 | 3,300 | 0.20 | 0.7 | 25.5 |
| | +Ca$^{2+}$ | 558 | 5.79 | | | 56,400 | 0.31 | 17.5 | |
| jRGECO1a$^a$ | -Ca$^{2+}$ | - | 8.6 | 148 | 1.90 | 6,180 | 0.18 | 1.1 | 9.2 |
| | +Ca$^{2+}$ | - | 6.3 | | | 53,300 | 0.21 | 11.2 | |
| jRCaMP1a$^a$ | -Ca$^{2+}$ | - | 5.6 | 214 | 0.86 | 33,800 | 0.29 | 9.8 | 2.0 |
| | +Ca$^{2+}$ | - | 6.4 | | | 54,100 | 0.54 | 29.2 | |

[a]Values are from Dana et al.[7] except for quantum yields which are from Molina et al[24]

[b]Extinction coefficient measured at each absorption maxima

[c]Brightness = $\varepsilon \times \varphi$

[d]Values calculated based on the brightnesses as listed in this table.

**Extended Data Table 2 | Sensor Characteristics in Neurons**

| Sensor | ΔF/F (1 stim) | ΔF/F (10 stim) | Half Rise Time (1 stim; ms) | Half Decay Time (1 stim; ms) |
|---|---|---|---|---|
| PinkyCaMP | ≤ 0.18 ± 0.0 1▲* | ≤ 0.26 ± 0.01▲ * | ≤ 670 ± 18 ▲* | ≤ 5572 ± 145▲* |
| RCaMP3 | ~0.25[a] ● | - | ~25[a] ● | ~180[a] ● |
| XCaMP-R | ~0.25[b] ● | - | - | ~190[b] ● |
| jRGECO1a | ~0.25[c] or ~0.03[a]● or ~0.1[b] ● | ~1.7[c] ▲ | ~25[c] ▲or ~60[a] ● | ~125[c] ▲or ~175[a] ●or ~310[b]● |
| jRCaMP1a | ~0.2[c] ▲ | ~0.2[c] ▲ | ~40[c] ▲ | ~1100[c] ▲ |
| OCaMP | 0.49 ± 0.03[d]△ | ~3.5[d] ▲ | 47.36 ± 0.97[d] ▲ | 291.53 ± 9.71[d] ▲ |
| GCaMP6f | 0.19 ± 0.028[e]△ | 1.692 ± 0.036[f] ▲ | 26 ± 2[f] ▲ | 140 ± 20[f] ▲ |
| GCaMP6m | 0.13 ± 0.009[e] ▲ | - | 80 ± 7[e]△ | 270 ± 23[e]△ |
| GCaMP6s | 0.23 ± 0.032[e] ▲ | 2.136 ± 0.037[f] ▲ | 56 ± 3[f] ▲ | 455 ± 40[f] ▲ |
| jGCaMP7f | 0.316 ± 0.024[f] ▲ | 2.99 ± 1.04[g] ▲ | 27 ± 2[f] ▲ | 265 ± 20[f] ▲ |
| jGCaMP7s | 0.657 ± 0.048[f] ▲ | 3.733 ± 0.115[f] ▲ | ~25[g] ▲ | ~300[g] ▲ |
| jGCaMP8f | 0.37 ± 0.1[g] ▲ | 2.45 ± 0.7[g] ▲ | 6.6 ± 0.1[g] ▲ | 87.5 ± 21.9[g] ▲ |
| jGCaMP8m | 0.75 ± 0.23[g] ▲ | 3.64 ± 0.8[g] ▲ | 7.4 ± 0.6[g] ▲ | 134.0 ± 13.6[g] ▲ |
| jGCaMP8s | 1.10 ± 0.21[g] ▲ | 3.88 ± 0.5[g] ▲ | 10.2 ± 0.9[g] ▲ | 330.7 ± 32.0[g] ▲ |

▲ cell attached recording *in vitro*; △ cell attached recording *in vivo* ; ● whole cell patch clamp recording *in vitro*

* Values for PinkyCaMP were obtained using field stimulation (as performed in Chen et al.[e] with an 1 ms, 40 V, 83 Hz field stimulus) and may reflect responses to one or more action potentials. Thus $\Delta F/F_0$ values represent upper bounds for single-AP responses. Comparisons across different studies from different labs is difficult due to methodological differences in imaging configuration, such as objectives, illuminators, filters, detectors, gain, frame rate and sample preparation. As imaging conditions differ across studies, our $\Delta F/F_0$ values should be interpreted as upper bounds rather than directly comparable measurements.

[a] Approximate values of whole-cell patch-clamp recordings in L2/3 pyramidal neurons of barrel cortex in mouse acute brain slices at 35°C from Yokoyama et al. 2024[10]
[b] Approximate values of whole-cell patch-clamp recordings in L2/3 pyramidal neurons of barrel cortex in mouse acute brain slices at 30°C from Inoue et al 2019 [9]
[c] Approximate values of field stimulation of rat primary hippocampal neurons at 30°C from Dana et al. [7]
[d] Approximate and exact values of field stimulation of rat primary hippocampal neurons at RT. Mean ± SEM. from Aggarwal et al. 2025[25]
[e] *In vivo* cell attached V1 neuron measurements in mice. Mean ± SD. from Chen et al. 2013[26]
[f] Recordings of field stimulation of rat primary hippocampal neurons at 30°C. Mean ± SEM. From Dana et al. 2019[27]
[g] Approximate and exact values of field stimulation of rat primary cortical neurons at RT. Mean ± SD. from Zhang et al. 2023[8]

# Reporting Summary

## Statistics

For all statistical analyses, confirm that the following items are present in the figure legend, table legend, main text, or Methods section.

| n/a | Confirmed | |
|---|---|---|
| ☐ | ☒ | The exact sample size (*n*) for each experimental group/condition, given as a discrete number and unit of measurement |
| ☐ | ☒ | A statement on whether measurements were taken from distinct samples or whether the same sample was measured repeatedly |
| ☐ | ☒ | The statistical test(s) used AND whether they are one- or two-sided *Only common tests should be described solely by name; describe more complex techniques in the Methods section.* |
| ☐ | ☒ | A description of all covariates tested |
| ☐ | ☒ | A description of any assumptions or corrections, such as tests of normality and adjustment for multiple comparisons |
| ☐ | ☒ | A full description of the statistical parameters including central tendency (e.g. means) or other basic estimates (e.g. regression coefficient) AND variation (e.g. standard deviation) or associated estimates of uncertainty (e.g. confidence intervals) |
| ☐ | ☒ | For null hypothesis testing, the test statistic (e.g. *F*, *t*, *r*) with confidence intervals, effect sizes, degrees of freedom and *P* value noted *Give P values as exact values whenever suitable.* |
| ☒ | ☐ | For Bayesian analysis, information on the choice of priors and Markov chain Monte Carlo settings |
| ☒ | ☐ | For hierarchical and complex designs, identification of the appropriate level for tests and full reporting of outcomes |
| ☒ | ☐ | Estimates of effect sizes (e.g. Cohen's *d*, Pearson's *r*), indicating how they were calculated |

*Our web collection on statistics for biologists contains articles on many of the points above.*

## Software and code

Policy information about availability of computer code

Data collection: For imaging in HEK cells and brain slices: LNscope from Luigs&Neumann equipped with a CMOS camera (Hamamatsu), Primary disscociated hippocampal neuronal culture: experiments were performed on an Olympus BX51 upright microscope equipped with a LUMPlanFL/IR ×40/0.80 W objective. A Multiclamp 700B (Molecular Devices) and Digidata 1550B digitizer (both Molecular Devices) were used to control and acquire electrophysiological recordings as well as light engine LEDs, bipolar field stimulation and camera exposure timing. Field stimuli were applied thorugh a Warner Instrument SIU-102 Stimulus isolator. For action spectra recordings, light from the Lumencor SpectraX23 light engine was filtered with narrow bandpass filters mounted on a FW212C filter wheel (Thorlabs) and delivered to the sample plane using a FM03R cold mirror (Thorlabs) in the epifluorescence beam path. The following filters were used (center wavelength ± 10 nm, Edmund Optics catalog no.): 372 nm (12147), 400 nm (65071), 422 nm (34496), 450 nm (65079), 480 nm (65084), 505 nm (34505), 535 nm (65095), 568 nm (65099), 600 nm (65102), 632 nm (65105) and 660 nm (86086). Imaging was performed with a Hamamatsu ORCA-Fire digital CMOS camera (C16240-20UP).

Organotypic slices cultures: Ca2+ Imaging experiments were in a custom recording chamber (1.5 ml) and superfused with Ringer's solution at 1 ml/min at 24 °C using a peristaltic pump (Minipuls 3, Gilson) and an in-line heater, respectively. The chamber was placed under an upright microscope (Axioscope, Zeiss) fitted with a 10x/0.3 water immersion objective (W N-Achroplan, Zeiss). Epifluorescence excitation for all red fluorescent indicators was provided by a collimated 554 nm LED (MINTL5, Thorlabs) using a 560/40 nm excitation filter and a 585 nm dichroic mirror, while fluorescence was collected with a 630/75 nm emission filter (ET-TxRed filter set, Chroma).

: DMi8 Leica with EMCCD camera (Evolve 512 delta, Photometrics), Confocal Microscopy LSM880 from Zeiss, 2P Imaging in vivo:For acquisition a custom-made Thorlaps two-photon microscope connected with a titanium sapphire 80 MHz Cameleon Ultra II two-photon lase (Coherent, Inc.) and equipped with an 8 kHz galvo-resonant scanner (LSK.GR08/M, Thorlabs), a GaAsP PMT (Thorlabs) and a 16x water immersion objective (Nikon) was used.In vivo fiber photometry Ca2+ signals (PinkyCaMP) and serotonin dynamics (sDarken) were recorded using an RX10x LUX-I/O Processor and Synapse software (TDT). An integrated LED driver controlled three LEDs for excitation: 560 nm (Lx560, TDT) for PinkyCaMP and mCherry, 465 nm (Lx465, TDT) for sDarken and 0Mut-sDarken, and 405 nm (LX405, TDT) for the isosbestic control

March 2021

signal. Each wavelength was set to a light intensity of 25-30 µW and modulated at unique frequencies—530 Hz for 560 nm, 330 Hz for 465 nm, and 210 Hz for 405 nm. The LEDs were connected to a 6-port Fluorescence Mini Cube (Doric Lenses), with output delivered via a fiber-optic patch cord (NA: 0.48, 600 µm; Thorlabs) through a rotary joint (RJ1 1x1, Thorlabs) to a subject cable secured to the implanted ceramic fiber-optic cannula using an interconnect (ADAL2, Thorlabs).
Fiber photometry: Fiber photometry recordings in the dorsal DG were performed using an iFMC6_IE(400-410)_E1(460-490)_F1(500-540)_E2(555-570)_F2(580-680)_S photometry system (Doric Lenses) controlled by the Doric Neuroscience Studio v6.1.2.0 software. A low-autofluorescence patch cord (400 µm, 0.57 N.A., Doric Lenses) was attached to the metallic ferrule on mouse's head

| Data analysis | ImageJ (Schneider et al.2012), IgorPro (WaveMetrics), ProfFit 7.0 (QuantumSoft),GraphPadPrism 9.3.1.Selfwritten Data analysis for dual color fiberphotometry is available on Github, other selfwritten codes will be made available upon request.The Fiber Photometire analysis script is publicly available on GitHub (https://github.com/masseck/FibPho-PinkyCaMP.git). |

For manuscripts utilizing custom algorithms or software that are central to the research but not yet described in published literature, software must be made available to editors and reviewers. We strongly encourage code deposition in a community repository (e.g. GitHub). See the Nature Portfolio guidelines for submitting code & software for further information.

## Data

Policy information about availability of data

All manuscripts must include a data availability statement. This statement should provide the following information, where applicable:
- Accession codes, unique identifiers, or web links for publicly available datasets
- A description of any restrictions on data availability
- For clinical datasets or third party data, please ensure that the statement adheres to our policy

DNA sequences are available in the Supplementary Information. DNA plasmids used for viral production have been deposited both on the UZH Viral Vector Facility (https://vvf.ethz.ch/) and on AddGene (plasmid #: 232857-232861). Viral vectors can be obtained either from the UZH Viral Vector Facility or from the Masseck lab. Raw data can be obtained by emailing the corresponding author. Source data are provided with this paper.

## Human research participants

Policy information about studies involving human research participants and Sex and Gender in Research.

| Reporting on sex and gender | *Use the terms sex (biological attribute) and gender (shaped by social and cultural circumstances) carefully in order to avoid confusing both terms. Indicate if findings apply to only one sex or gender; describe whether sex and gender were considered in study design whether sex and/or gender was determined based on self-reporting or assigned and methods used. Provide in the source data disaggregated sex and gender data where this information has been collected, and consent has been obtained for sharing of individual-level data; provide overall numbers in this Reporting Summary. Please state if this information has not been collected. Report sex- and gender-based analyses where performed, justify reasons for lack of sex- and gender-based analysis.* |
| Population characteristics | *Describe the covariate-relevant population characteristics of the human research participants (e.g. age, genotypic information, past and current diagnosis and treatment categories). If you filled out the behavioural & social sciences study design questions and have nothing to add here, write "See above."* |
| Recruitment | *Describe how participants were recruited. Outline any potential self-selection bias or other biases that may be present and how these are likely to impact results.* |
| Ethics oversight | *Identify the organization(s) that approved the study protocol.* |

Note that full information on the approval of the study protocol must also be provided in the manuscript.

# Field-specific reporting

Please select the one below that is the best fit for your research. If you are not sure, read the appropriate sections before making your selection.

☒ Life sciences      ☐ Behavioural & social sciences      ☐ Ecological, evolutionary & environmental sciences

For a reference copy of the document with all sections, see nature.com/documents/nr-reporting-summary-flat.pdf

# Life sciences study design

All studies must disclose on these points even when the disclosure is negative.

| Sample size | Sample size was not predetermined. Sampel size is given for each experiment |
| Data exclusions | No data have been excluded from the analysis. |

| Replication | Each of the described experiments are replicated in several dishes, cultures, slices or animals. |
| Randomization | Mice were randomly assigned in experimental groups. Mice of both sexes were used. |
| Blinding | Investigators were not blinded to the data or experimental animals. |

# Reporting for specific materials, systems and methods

We require information from authors about some types of materials, experimental systems and methods used in many studies. Here, indicate whether each material, system or method listed is relevant to your study. If you are not sure if a list item applies to your research, read the appropriate section before selecting a response.

## Materials & experimental systems

| n/a | Involved in the study |
|-----|-----------------------|
| ☐ | ☒ Antibodies |
| ☐ | ☒ Eukaryotic cell lines |
| ☒ | ☐ Palaeontology and archaeology |
| ☐ | ☒ Animals and other organisms |
| ☒ | ☐ Clinical data |
| ☒ | ☐ Dual use research of concern |

## Methods

| n/a | Involved in the study |
|-----|-----------------------|
| ☒ | ☐ ChIP-seq |
| ☒ | ☐ Flow cytometry |
| ☒ | ☐ MRI-based neuroimaging |

## Antibodies

| Antibodies used | Primary antibody against GFP and mCerulean (chicken a-GFP, 1:1000; catalog no. GFP-1010, Aves Labs) andC-Fos rat a-cFOS, 1:1000; catalog no. 226 017, Synaptic Systems).<br>Secondary antibodies: Alexa Fluor 488 donkey anti-chicken, 1:500; catalog no. 703-545-155, Jackson laboratories and Alexa Fluor 647 goat anti-rat, 1:500; catalog no. 31226, Invitrogen, conjugated in-house) |
| Validation | Validation done by the manufactorer. |

## Eukaryotic cell lines

Policy information about cell lines and Sex and Gender in Research

| Cell line source(s) | HEK293T (DSMZ ACC-635), HEK293T  (Sigma Aldrich 12022001) |
| Authentication | none of the cell lines were authentificated |
| Mycoplasma contamination | Cells were are tested for Mycoplasma contamination on a regular basis. |
| Commonly misidentified lines<br>(See ICLAC register) | n.a. |

## Animals and other research organisms

Policy information about studies involving animals; ARRIVE guidelines recommended for reporting animal research, and Sex and Gender in Research

| Laboratory animals | C57BL/6J mice of both sexes and vGAT mice (B6J.129S6(FVB)-Slc32a1tm2(cre)Lowl/MwarJ |
| Wild animals | n.a. |
| Reporting on sex | Sex was not considered in the study design. |
| Field-collected samples | n.a. |
| Ethics oversight | Animal protocols were approved by local authorities. |

Note that full information on the approval of the study protocol must also be provided in the manuscript.

