## [Peer Review File · Nature Methods]

PinkyCaMP a mScarlet-based calcium sensor with exceptional brightness, photostability, and multiplexing capabilities

Corresponding Author: Professor Olivia Masseck

Version 0:

Decision Letter:

21st May 2025

Dear Professor Masseck,

Thank you for your patience. I am sorry that the review of your manuscript took much longer than you and me would have liked. Your Article, "PinkyCaMP a mScarlet-based calcium sensor with exceptional brightness, photostability, and multiplexing capabilities", has now been seen by three reviewers. As you will see from their comments below, although the reviewers find your work of considerable potential interest, they have raised a number of concerns. We are interested in the possibility of publishing your paper in Nature Methods, but would like to consider your response to these concerns before we reach a final decision on publication.

We therefore invite you to revise your manuscript to address these concerns. As you can see below, the reviewers request additional characterizations of your sensor. They also ask that responses to single action potentials are shown. Please also address their other concerns.

Link Redacted

We hope to receive your revised paper within 2-3 months. If you cannot send it within this time, please let us know. In this event, we will still be happy to reconsider your paper at a later date so long as nothing similar has been accepted for publication at Nature Methods or published elsewhere.

OPEN SCIENCE REQUIREMENTS

REPORTING SUMMARY AND EDITORIAL POLICY CHECKLISTS

EXTENDED DATA FIGURES

DATA AVAILABILITY

All novel DNA and RNA sequencing data, protein sequences, genetic polymorphisms, linked genotype and phenotype data, gene expression data, macromolecular structures, and proteomics data must be deposited in a publicly accessible database, and accession codes and associated hyperlinks must be provided in the "Data Availability" section.

MATERIALS AVAILABILITY

ORCID

Nature Methods is committed to improving transparency in authorship. As part of our efforts in this direction, we are now requesting that all authors identified as 'corresponding author' on published papers create and link their Open Researcher and Contributor Identifier (ORCID) with their account on the Manuscript Tracking System (MTS), prior to acceptance. This applies to primary research papers only. ORCID helps the scientific community achieve unambiguous attribution of all scholarly contributions. You can create and link your ORCID from the home page of the MTS by clicking on 'Modify my Springer Nature account'. For more information please visit <http://www.springernature.com/orcid>.

Best regards,
Nina

Nina Vogt, PhD
Senior Editor
Nature Methods

Reviewers' Comments:

Reviewer #1 (Remarks to the Author):

This manuscript characterizes a novel long-wavelength GECI that represents a step change in improvement over the existing pre-eminent red GECIs (jRGECO and rCaMP). The new variant, PinkyCaMP, is reported to show notable improvements in overall brightness, Ca²⁺-evoked signal-to-noise ratio, lack of photoswitching, resistance to photobleaching and lack of internalization. The authors demonstrate these improvements convincingly using standard functional measures in vitro and in vivo, and also demonstrate the utility of the sensor for all-optical activation and monitoring of neural activity as well as dual-color imaging. The reporting of statistics and summary results is appropriate. There are some features - and limitations - that could be addressed more clearly, but in general this is a nice report and the sensor should be a welcome addition to the optical toolbox.

Specific comments

Major:

1. The rise-time of the pinkyCaMP (e.g., in response to a single field stimulation in cultured neurons) is not reported. This would be helpful in comparing to other GECIs including the GCaMP variants, as characterized by the Looger group.

2. Likewise, dynamic range (in terms of spike rate or stimulation rate, for example) is also not addressed here. It seems important to do so.

2. Responses to a single field stimulation are reported in Figure 2j and appear quite small - possibly 50 times smaller (or more) than that for GCaMP8m, for example (comparing with Figure 1c of Zhang et al. 2021). What about responses to a single action potential - or, alternatively the number of action potentials necessary to evoke a detectable response in a neuron? In general, a simultaneous cell-attached recording and pinkyCaMP imaging from a neuron exhibiting 'normal' bouts of spontaneous activity would be very helpful to assess the general utility of the sensor for following neural activity and the single-cell level.

3. The dual-color fiber photometry experiments use the serotonin sensor sDarken, which exhibits dimming on binding 5-HT. This is an odd choice of green sensor to use to demonstrate dual-color imaging. Dual-color imaging using pinkyCaMP and GCaMPx expressed in distinct neural populations would be more compelling. A technical concern with such an approach is

the possibility that photon emission from GCaMP could excite pinkyCaMP fluorescence, thus leading to cross-talk in the channels? Given the high dF exhibited by the latest GCaMP variants this actually seems like a reasonable possibility worth addressing.

4. The Ca^{2+} affinity of the pinkyCaMP0.9c (54 nM) is extremely high - matched only by that of GCaMP8s. Relatedly, the decay time is very long. It is worth mentioning these features more prominently and addressing their implications/limitations for monitoring neural activity and the potential for disruption of neural signaling processes more explicitly in the Discussion.

Minor:

1. Fig 1 legend does not match the Figure panels. The data referred to as Fig 1G in the legend appears to be missing, and subsequent panels are mislabeled.

2. Also in Figure 1, the frequency of stimulation in panel H appears to be faster than 0.1 Hz based on the time-scales in the Figure. Please confirm.

Reviewer #2 (Remarks to the Author):

As mScarlet is indeed the brightest fluorescent protein (FP) in the red wavelength region and various variants of mScarlet are being developed actively, it is natural for many researchers to expect the development of mScarlet-based bright biosensors. Just as expected, Masseck, Campbell and their colleagues successfully developed red intensimetric sensors for calcium with exceptional brightness, photostability, and multiplexing capabilities. Overall, I think that the manuscript presents sufficient information that demonstrates the usefulness of the product, PinkyCaMP.

Most FPs, including the classic variants of Aequorea GFP, have Tyr at the second position of the chromophore-forming tripeptide. As a result, they each show a bimodal absorption spectrum characterized by two peak maxima, which correspond to the protonated (neutral) and deprotonated (ionized) states of the chromophore. And this equilibrium is determined by a hydrogen bond network around the chromophore. In single FP-based biosensors, furthermore, the equilibrium can be modulated by the environment that their sensing domains generate. Therefore, GECIs, such as GCaMPs, exhibit a big change in their absorption spectra with and without calcium. In a recent Nature paper by Looger and colleagues (Zhang et al., Nature 615, 884-891 (2023)), for example, three jGCaMP8s exhibited >20-fold increases in absorbance but only slight changes in fluorescence quantum yield upon binding to calcium (Zhang et al., Extended Data Table 3). So, I was very surprised to see very large differences in fluorescence quantum yield presented in Table 1 of this manuscript (PinkyCaMP, 16 fold; PinkyCaMP0.9b, 21.5 fold). It is concluded that in PinkyCaMPs calcium binding enhances the emission of light but not the absorption of light, for some reason. I think this is a remarkable departure from the conventional GECI concept. For example, I am certain that the lifetime of PinkyCaMP changes with calcium and would suggest its combination with FLIM technology. I also investigated some other GECIs developed by Campbell group recently. In one paper (Dalangin et al., Nature Comm. 16:3318 (2025)), FR-GECO1c showed a large change in absorbance with calcium and a moderate change (4.5 fold) in fluorescence quantum yield.

In relation to this point, I have some comments as follows.

#1 Photostability (Figure 1i)

Baseline brightness in HEK cells expressing PinkyCaMP, RCaMP3, or jRGECO1a was monitored with time. I understand that all these GECIs were depleted of calcium. So I strongly suggest that these data be normalized by absorption cross section or extinction coefficient. According to Table 1, the extinction coefficients of PinkyCaMP, RCaMP3, and jRGECO1a are 71,000, 3,300, and 6,180, respectively. In this situation, PinkyCaMP is highly disadvantaged. Alternatively, I would suggest that these experiments be performed with persistent calcium mobilization. Due to increases in absorbance, RCaMP3 and jRGECO1a would photobleach faster. Likewise, please examine Figure 3f and 3g.

#2 Minor comments

The legend for Figure 1g-l is a mess.

Please specify the wavelengths for extinction coefficients.

Reviewer #2 (Remarks on code availability):

I was able to open the Python script for analysis but had no time to review it carefully.

Reviewer #3 (Remarks to the Author):

Fink et al describe the engineering and testing of PinkyCaMP, a new red fluorescent genetically encoded calcium indicator (GECI) that features high brightness, large responses, and lack of photoactivation by blue light. Of these features, the last one is arguably the most unique and useful, as orthogonality with blue-excitable opsins is one of the major reasons to develop red indicators, and previous red GECIs all showed photoactivation.

PinkyCaMP demonstrates impressive metrics in the parameters mentioned above, as a result of heroic engineering efforts

involving 12 rounds of random mutagenesis. The in cellulo, ex vivo, and in vivo characterization are well done and relevant for the most part. The experiment with GtAChR in vivo is especially rigorous and elegant.

However there are a few issues that influence the rigor of the conclusions and the utility of PinkyCaMP that should be rectified prior to publication of this manuscript. The major ones are:

1. The numbers in Table 1 do not support PinkyCaMP0.9c (selected as “the” PinkyCaMP) being superior to PinkyCaMP0.9b. Authors state “PinkyCaMP0.9c was selected as the best variant for its balance of brightness and responsiveness and renamed as PinkyCaMP”. However Table 1 shows the overall dF/F (from calcium-free to -saturated states) to be 14.3 for PinkyCaMP0.9b vs 12.8 for PinkyCaMP(0.9c). Meanwhile the calcium free/saturated brightnesses are 1.6/24.4 for PinkyCaMP0.9b vs 2.1/28.9 for PinkyCaMP(0.9c). These differences thus favoring 0.9b in one metric and 0.9c in the other, but both by small degrees. Alone it does not seem sufficient to explain the choice, so “the authors should either present more convincing data to rule out 0.9b, or continue to characterize 0.9b more”. Specifically, it is not clear if the high affinity of PinkyCaMP0.9c at 54nM is an asset in this case. The past history of GECIs is marked by a realization that selection for high affinity without coselection for fast activation kinetics leads to sensors with very slow off-rates. That appears to have happened here too. The >5-second off-rates in Fig. 2l when CoChR is tuned on for only 1ms is rather striking, and “the authors should mention this result explicitly in the text”. jGCaMP engineering on the other hand specifically sought to identify sensors that improved affinity via increasing activation kinetics, which allows separation of closely spaced APs. Selection for fast on-kinetics does not appear to have happened here, and PinkyCaMP0.9c has very slow off-kinetics. Meanwhile 0.9b has 4-fold lower Kd. It’s likely it has faster off-kinetics, maybe even 4-fold faster. This may be more useful to researchers and would be willing to sacrifice the slightly higher brightness of 0.9c to gain the faster kinetics of 0.9b. Indeed the authors appear aware of this issue as they mention the desire to accelerate kinetics in the discussion. But maybe they already have a faster indicator in 0.9b! “To settle this question, single AP measurements are necessary”. Indeed the GECI field has long coalesced on using single-AP responses to provide figures of merit, which have been reported for every GCaMP since GCaMP3 and was seen in the jRGECO1a and RCaMP3 papers as well. In single-AP responses, it is possible PinkyCaMP0.9c will outperform PinkCaMP0.9b in dF/F given its lower Kd, but importantly the authors will be able to ascertain times of half-rise and half-decay. The RCaMP3 and jRGECO1a papers reported responses to single APs in neurons by electrical stimulation of neuronal cultures or by patch-clamping, respectively. Either method could be performed here. I’d suggest this should be done for both 0.9b and 0.9c, and if there are advantages of 0.9b, it should be presented as an useful alternative. For example if 0.9b decay is faster, it could be PinkyCaMP1f and 0.9c could be PinkyCaMP1s. Notably such an elevation of status for 0.9b would not necessitate repeating the in vivo experiments with 0.9b, as the existing experiments with 0.9c are sufficient to prove what the authors want to demonstrate, e.g. ability to be used with blue-excitable opsins.

2. The Fig. 3 slice experiment is extremely important as the only experiment in which PinkyCaMP is compared to a green GECI (GCaMP6) and the other best red GECIs, jRGECO1a and RCaMP3. The only activity that is imaged is cyclical spontaneous bursting, but only one sample trace is shown for each indicator and these are split between main Fig. 3 and Fig S5. “Representative traces for all indicators should be shown in Fig. 3 so that some primary data is presented for each indicator”. Care should be taken that the displayed traces are near the mean of their group for frequency and dF/F.

3. The differences in bursting behavior in organotypic slices expressing the different GECIs in Fig. 3 are concerning. The periods of these bursts is significantly lower for PinkyCaMP than jRGECO1a. This may indicate that PinkyCaMP is slowing synaptic transmission, perhaps due to calcium buffering at synaptic terminals or longer-term suppression of neuronal maturation via calcium quenching. This may be another situation where perhaps PinkyCaMP0.9b would be better. “It would be useful to add an experiment comparing PinkyCaMP0.9c at two different gene expression levels (unchanged and 10x lower, for example) to PinkyCaMP0.9b and jRGECO1a at the same two levels”. This will inform the reader whether overexpression may have effects on neuronal physiology, and whether this is more likely to happen for PinkyCaMP0.9c, PinkyCaMP0.9b, or jRGECO1a.

4. The only photostability experiments are done in 1-photon excitation in Fig. 2b. “Photostability should also be characterized in 2-photon excitation”. As the RCaMP3 paper already compared RCaMP3, jRGECO1a, and XCaMP-R for photostability under 1040nm excitation (Extended Data Fig. 4), the authors here would only need to compare PinkyCaMP0.9c, PinkyCaMP0.9b, and RCaMP3 under the same 1040nm excitation and the same power, and the other comparisons can be made transitively.

5. In the paragraph describing Figure 6, imaging at 10 Hz cannot fairly be described as picking up unitary events. Unitary somatic events would be understood to be single APs. In addition to the slow imaging, the slow kinetics of PinkyCaMP would make that difficult anyway. Likewise there is no basis to claim that “These data show reliable measurement of Ca²⁺ transients with PinkyCaMP.” In other GECI papers, similar claims derive only from simultaneous ephys (loose-cell is enough) and GECI imaging in acute slice or in vivo. Here, all that can be claimed is that PinkyCaMP can detect some activity. “To make this experiment informative, the authors should coimage a GCaMP6, 7, or 8 in the same cell”. It’s surprising this is not done anywhere in the paper since that is an easy way to benchmark to well studied indicators in vivo.

Minor points:

1. The statement “other monomeric red fluorescent proteins (RFPs) used in GECIs, such as mApple or mRuby have yields far below 50%” appears not well supported. Perhaps all that can be said is that they have quantum yields lower than mScarlet. Reference 22 shows mApple to have QY of 47%, which is not “far below 50%”. Shaner et al also measured QY of 49% for mApple in the existing reference 15 which should probably be cited again here. In addition, Ref 22 doesn’t include

any work on mRuby so another reference is needed for that. As the brightness of isolated mRuby is not as important as the brightness of the GECIs derived from it, I think Dana et al eLife (existing ref 11) should be cited; here RCaMP1h, jRCaMP1a, jRCaMP1b, and mRuby itself all have quantum yield > 0.5 (<https://elifesciences.org/articles/12727/figures#fig2s2>). Indeed the authors in the present manuscript also measure the QY of the mRuby-based jRCaMP1a to be 54%.

2. mScarlet having minimal/negligible photoswitching behavior is repeated twice in the introduction. The second time is "Additionally, mScarlet demonstrates negligible photoswitching behavior22."

3. Fig 1 caption section g refers to a plot that has been moved to supplementals, but section g was not deleted and the subsequent sections are now mislettered.

4. In Fig 1 panel h (h in the image, not h in the caption), the two colors in the color legend are too close to be differentiated. I suggest perhaps using an orange color for RCaMP3.

5. The sentence "We anticipated that this small signal increase could be either caused by the small photocurrent generated by stCoChR (Figure 2g) and/or by the activation of voltage gated calcium channels and its inherent high Ca^{2+} -permeability paired with the high sensitivity of PinkyCaMP ($K_d = 54 \text{ nM}$)" is not completely logical. The stCoChR is definitely opening since there is a difference in calcium signal between light and no light. So the first part of the sentence doesn't belong after an "either". Rather the sentence should be "We anticipated that this small signal increase could be caused by the small photocurrent generated by stCoChR (Figure 2g) with or without activation of voltage gated calcium channels paired with the high sensitivity of PinkyCaMP ($K_d = 54 \text{ nM}$)". Also it appears the authors meant to add "its inherent high Ca^{2+} -permeability" after stCoChR but added it after VGCCs. Certainly VGCCs have inherently high Ca^{2+} permeability... they are after all calcium channels.

6. In the sentence, "However, the application of orange light used for imaging caused a small inward current of $9 \pm 3 \text{ pA}$, much lower as currents measured for action spectra determination at 480 nm ($288 \pm 65 \text{ pA}$, Figure 2h)", the authors probably mean "much lower than current measured at low powers of 480-nm light used during action spectra determination." But in addition, powers used during action spectra determination is meaningless to the reader. The authors should just write "the application of orange light used for imaging at ** mW/mm^2 caused an inward current of $9 \pm 3 \text{ pA}$, which was much smaller than currents measured with 480 nm excitation at the much lower power of ** mW/mm^2 ".

7. Figure 2h graphic is awkward. If the authors mean that there are 3 conditions of field stimulation, 438nm, or nothing, then they should put the images of electrodes and diodes in one column or one row. An image of an opsin rather than a diode would be more informative anyway.

8. "PinkyCaMP maintained strong baseline fluorescence (Figure 3l)" should refer to Figure 3m

9. Figures 3i and 3j are not cited. Figure 3i should really be combined with Figure 3l anyway.

10. The paragraph describing Figure S7 and Figure 3k-3o uses a high and low light power. The actual power in mW/mm^2 should be specified so that readers know what range of powers they might try.

11. The authors should explain why they chose to perform all ex vivo comparisons to GCaMP6f instead of a jGCaMP8.

12. "PinkyCaMP exhibited robust Ca^{2+} transients" should be "PinkyCaMP reported..."

13. Fig. 4j in the text should be Fig. 4i.

14. "As expected, PinkyCaMP signals increased during the light-ON epochs in the stGtACR2 group and had no effect on the EGFP group" should be "PinkyCaMP signals were not affected in the EGFP group"

15. In the paragraph describing Figure 6, the 3 sentences starting from "First applications of GCaMP-imaging" do not really explain the utility of a red GECI and are not relevant anyway to the rest of the paragraph. The real utility is to image the red GECI together with another indicator for something else (e.g. neurotransmitter) in the same cell or closely apposed cells.

16. The term "dlox" in dlox-GtACR2-ST-mCerulean is not defined; and google search reveals no relevant results. Did the authors mean DIO?

Version 1:

Decision Letter:

Our ref: NMETH-A59124A

4th Dec 2025

Dear Olivia,

Thank you for submitting your revised manuscript "PinkyCaMP a mScarlet-based calcium sensor with exceptional

brightness, photostability, and multiplexing capabilities" (N METH-A59124A). As I have already communicated, the manuscript has been seen by the original referees and their comments are below. The reviewers find that the paper has improved in revision, and therefore we'll be happy in principle to publish it in Nature Methods, pending minor revisions to satisfy the referees' final requests as outlined in your previous message and to comply with our editorial and formatting guidelines.

TRANSPARENT PEER REVIEW

Nature Methods offers a transparent peer review option for new original research manuscripts. We encourage increased transparency in peer review by publishing the reviewer comments, author rebuttal letters and editorial decision letters if the authors agree. Such peer review material is made available as a supplementary peer review file. **Please state in the cover letter 'I wish to participate in transparent peer review' if you want to opt in, or 'I do not wish to participate in transparent peer review' if you don't.** Failure to state your preference will result in delays in accepting your manuscript for publication.

ORCID

Author names using non-Roman characters

Nature Portfolio journals can support presentation of author names using non-Roman characters in the HTML version of the article. If you wish to, please include author names in parentheses after the Roman-character spelling; [see example online here](https://www.nature.com/articles/s44222-024-00258-2). Currently supported scripts are: Arabic, Chinese, Cyrillic, Devanagari, Greek, Hebrew, Hangul, Japanese and Persian. You will be asked to verify the rendering is correct at proof stage.

Best regards,
Nina

Nina Vogt, PhD
Senior Editor
Nature Methods

Reviewer #1 (Remarks to the Author):

The authors have done a nice job of responding to the comments in the initial round of reviews. The addition of new data including simultaneous imaging of PinkyCaMP / GCaMP8s, relating signals to action potential firing number, and acknowledging the slower kinetics of the sensor are much appreciated. I am satisfied with these revisions and think that this paper is an important contribution that is highly appropriate for Nature .

Reviewer #2 (Remarks to the Author):

After reviewing the authors' responses to the reviewers' comments and the revised manuscript, I noticed a substantial improvement. While the slow kinetics of PinkyCaMP, which is likely due to its high calcium affinity, may still be a concern for researchers who are particular about temporal resolution, it is evident that PinkyCaMP outperforms other red GECs in brightness, photostability, signal-to-noise ratio, and compatibility with general optogenetics and neurotransmitter imaging.

I have the following minor comments:

Methods section, page 4, line 7: "A 4x air" should be "A 40x water."
Clearly state how irradiance was measured in this study.

Figure S12: I suspect that the transient ratio of PinkyCaMP to GCaMP reflects their expression levels. This could be discussed in the legend.

Reviewer #3 (Remarks to the Author):

The authors are to be commended for this thorough revision in which they performed experiments to address each of the reviewers' concerns. The revision included high-quality cell-attached patch data, field stimulus data, and jGCaMP8s co-imaging data (the jGCaMP8s co-imaging is especially nice).

However I have a major concern regarding the interpretation of the field stimulus data in relationship to published work. It cannot be assumed that a single field stimulus creates a single AP. It's possible that it creates a range of AP numbers, e.g. 1-3 with a mean of ~2. It is also not possible to compare one lab's field stimulus results to those obtained in another lab; there are numerous potential differences. Thus the sentence "Across all samples, a single AP evoked an average response of $18 \pm 1\% \Delta F/F_0$ (Figure 2g), comparable to responses of other red calcium indicators and GCaMP6 variants (Table 2)" is unsupported in two critical words: "AP" and "comparable". The groups in Janelia Farm using field stimulus made a point of validating that the field stimulus they described as generating one AP really did do so (Chen et al Nature 499:295, "Voltage imaging using the ArchWT-GFP archaerhodopsin-based voltage sensor confirmed that individual pulses (1 ms, 40 V, 83 Hz) reliably triggered single APs"), but I do not see that done here. Indeed on the patch-clamped neuron the one AP that was recorded only produced $\sim 2.5\% \Delta F/F_0$. Thus the authors have several choices:

1. Characterize the distribution of AP numbers created by their single field stimulus pulses as in Chen et al, and adjust their field stimulus if necessary to evoke truly only 1 AP, and repeat the measurement if necessary (might be okay already, or might be tedious)
2. Skip the calibration, but characterize a GCaMP as a comparator in their field stimulation system, and remove entries in Table 2 of other GECIs characterized in other labs in with field stimulation because those numbers may not be comparable, and replace the word "AP" in the sentence above with "field stimulus", and modify the phrase starting with "comparable" to state the actual result with the GCaMP in the same system.
3. Do neither of the above, but consider the dF/F to be an upper bound for a single AP (as it may reflect 1 or more APs), and state clearly in the text that the field stimulus elicits a calcium response but the number of underlying APs is unknown, and describe a single-AP response as " $\leq 18 \pm 1\%$ " in the text, and place an asterisk in Table 2 next to " $< 18 \pm 1\%$ " to explain that the conditions tested may not have elicited only a single AP, and likewise also add " \leq " before each other number in Table 2 for the PinkyCaMP row.

Related to this, Table 2 should not mix values from cultured unpatched neurons and patched neurons in vivo; these are expected to have completely different dF/F (because imaging methods will have different backgrounds and background subtractions, as well as the presence of a electrode in the way) and SNR (obviously, because they will be using different objectives, illuminators, filters, collectors, imaging gain, and time). Indeed SNR should be dropped completely as no two labs will have exactly the same combination of objectives, illuminators, filters, collectors, and imaging gain and time. About the only thing that maybe can be compared across papers is cultured neuron single field pulses, and only if a common GECI functions similarly across the papers (maybe GcaMP6f).

My desire here is simply that any comparative claims put into print are correct, or at least as likely to be correct as reasonable. Here without an in-experiment side-by-side comparison under the same conditions, no reliable comparison of performance can be made. If these issues can be corrected, then the manuscript would be a strong candidate for publication in this journal.

We would like to thank the editor and reviewers for their positive feedback and insightful comments. By revising the manuscript and adding new experimental data and analyses, we have addressed the detailed points raised by the reviewers and discussed them accordingly. Overall, we believe these revisions have substantially improved the manuscript. In particular, we focused on concerns regarding comparability to existing sensors: We now include responses to single action potentials and provide a more detailed characterization of several properties. In addition, we performed further *in vivo* experiments to demonstrate the capability of PinkyCaMP for *in vivo* imaging, i.e. dual-color 2P and Mini-2P. In the following section we will comment (in *blue italic*) on each of the reviewer suggestions and explain how we tried to integrate their suggestion and comments in the revised manuscript.

Reviewer #1 (Remarks to the Author):

This manuscript characterizes a novel long-wavelength GECI that represents a step change in improvement over the existing pre-eminent red GECIs (jRGECO and rCaMP). The new variant, PinkyCaMP, is reported to show notable improvements in overall brightness, Ca²⁺-evoked signal-to-noise ratio, lack of photoswitching, resistance to photobleaching and lack of internalization. The authors demonstrate these improvements convincingly using standard functional measures *in vitro* and *in vivo*, and also demonstrate the utility of the sensor for all-optical activation and monitoring of neural activity as well as dual-color imaging. The reporting of statistics and summary results is appropriate. There are some features - and limitations - that could be addressed more clearly, but in general this is a nice report and the sensor should be a welcome addition to the optical toolbox.

Specific comments

Major:

1. The rise-time of the pinkyCaMP (e.g., in response to a single field stimulation in cultured neurons) is not reported. This would be helpful in comparing to other GECIs including the GCaMP variants, as characterized by the Looger group.

*We have now included rise-time and other response data following single field stimulation in cultured mouse neurons expressing PinkyCaMP. These experiments were performed under comparable conditions, and the results are presented in **Figure 2**. In addition, we have expanded the comparison of PinkyCaMP kinetics with those of other GECIs in neurons, which is now summarized in **Table 2**.*

2. Likewise, dynamic range (in terms of spike rate or stimulation rate, for example) is also not addressed here. It seems important to do so.

*We now characterized the response of PinkyCaMP to spontaneous spikes in mouse cultures (**Figure 2a-c**) and to different field stimuli (**Figure 2d-j**).*

2. Responses to a single field stimulation are reported in Figure 2j and appear quite small - possibly 50 times smaller (or more) than that for GCaMP8m, for example (comparing with Figure 1c of Zhang et al. 2021). What about responses to a single action potential - or, alternatively the number of action potentials necessary to evoke a detectable response in a neuron? In general, a simultaneous cell-attached recording and pinkyCaMP imaging from a neuron exhibiting 'normal' bouts of spontaneous activity would be very helpful to assess the general utility of the sensor for following neural activity and the single-cell level.

*Indeed, responses to single field stimulation in the previous Figure 2j (now **Figure 2p**) are quite small. This can be explained by the imaging conditions used that were optimized to reduce crosstalk between PinkyCaMP and stCoChR (**Figure 2m**). As mentioned above, we now provide PinkyCaMP data in response to spontaneous activity of cultured neurons and in response to field stimulation evoked activity. These data were obtained under optimal imaging conditions and show a much higher response (e.g. 18% dF/F_0 per single field stimulus, **Figure 2e-g** and **Table 2**).*

3. The dual-color fiber photometry experiments use the serotonin sensor sDarken, which exhibits dimming on binding 5-HT. This is an odd choice of green sensor to use to demonstrate dual-color imaging. Dual-color imaging using pinkyCaMP and GCaMPx expressed in distinct neural populations would be more compelling. A technical concern with such an approach is the possibility that photon emission from GCaMP could excite pinkyCaMP fluorescence, thus leading to cross-talk in the channels? Given the high dF exhibited by the latest GCaMP variants this actually seems like a reasonable possibility worth addressing.

*In both the first and second fiber photometry experiments, we employed GFP-based sensors or fluorophores, including sDarken, the Nullmutant, and eGFP. Particularly, eGFP has a comparable molecular brightness to newer versions of GCaMP. All of these require blue light excitation and, particularly in the case of the Nullmutant and eGFP, which both emit green fluorescence. If PinkyCaMP would be substantially excited by green photon emission from these fluorophores, we would have expected to observe signal contamination under these conditions. More critically, any unintended excitation of PinkyCaMP due to light exposure would be more likely to occur during blue-light illumination—either from imaging itself or, even more so, during optogenetic stimulation at higher light intensities. We would expect to observe light-evoked transients during these conditions, particularly in the multiplexing experiments involving stGtACR2 (see corresponding **Figure 5 e-f**). However, such transients are not observed, arguing against significant photoactivation or excitation of PinkyCaMP under these conditions.*

*However, we agree that dual color imaging in combination with sDarken was probably not a smart choice. Thus we followed the recommendation of the reviewer and performed dual color imaging in separate neuronal populations: GAD2-positive interneurons and CA1 neurons in the hippocampus (**Figure 7**). In addition, we recorded PinkyCaMP and GCaMP8s in the same*

CA1 neurons (**Figure S 12**) as suggested by reviewer3. Recording PinkyCaMP in CA1 neurons and GCaMP8s in GAD2 interneurons in stratum pyramidale of the hippocampus showed that there was hardly any crosstalk from the red channel to the green channel (which would also argue against physical properties). Since the ROIs were drawn by hand, which always bears some inaccuracy, some GCaMP signal was visible in the green channel of ROIs drawn around PinkyCaMP positive neurons. In addition, the often very bright signal of highly active GABAergic neurons was only rarely leaking into the red channel. Thus, the combination of brightly fluorescent GCaMP8s with PinkyCaMP in separate subsets of neurons is possible and opens the possibility of two-color Ca^{2+} -imaging. We are convinced that this is a real advantage of PinkyCaMP. The recordings of PinkyCaMP and GCaMP in the same neurons were requested by reviewer 3. We were able to simultaneously record fluorescent changes in the red (PinkyCaMP) and green (GCaMP) channels. However, as the reviewer already points out, these results are difficult to interpret, since it is unclear how the two indicators compete for Ca^{2+} or influence each other by FRET.

4. The Ca^{2+} affinity of the pinkyCaMP0.9c (54 nM) is extremely high - matched only by that of GCaMP8s. Relatedly, the decay time is very long. It is worth mentioning these features more prominently and addressing their implications/limitations for monitoring neural activity and the potential for disruption of neural signaling processes more explicitly in the Discussion.

Thank you for this very important note and we agree that we have to be more explicit about the limitations of PinkyCaMPs slow kinetics. We extended the characterization of PinkyCaMP in neuronal culture (**Figure 2a-j**). Although PinkyCaMP exhibits relatively slow kinetics in cultured neurons at room temperature, its kinetic profile in organotypic slices at 24 °C is comparable to jRCaMP1a (**Figure 3**), and in vivo its performance appears similar to GCaMP8s (**Figure S12**). In addition we have rephrased and complemented the discussion as follows:

“ Due to its high calcium affinity, PinkyCaMP exhibits relatively slow kinetics, limiting its ability to resolve fast spiking activity and reducing spike timing precision. This may constrain its use in experiments which require high temporal resolution, such as encoding and decoding analyses. Importantly, the relatively slow decay kinetics may lead to temporal integration of signals, potentially masking high-frequency spiking activity. This could result in an underestimation of spike rates or misinterpretation of population dynamics, especially in circuits where precise spike timing carries critical information. However, its high sensitivity makes it well-suited for imaging sparsely active neuronal populations, where reliable detection of individual events is essential. Our results demonstrate that, despite its slower kinetics, PinkyCaMP enables robust detection of activity in pyramidal neurons and dentate gyrus granule cells across various in vivo settings — such as fiber photometry, awake head-fixed two-photon microscopy and freely moving 1P, 2P and miniscope -is feasible. The slow kinetics are likely a direct consequence of its high Ca^{2+} affinity, which also increases the risk of calcium buffering when overexpressed. This can likely be mitigated by careful titration of viral

load and control of expression levels. However, even after long-term expression of 90 days in the hippocampus, PinkyCaMP recording was possible. Future engineering efforts may focus on accelerating its kinetics to broaden its applicability to fast-firing neurons. As this is the first generation of an mScarlet-based calcium sensor, there is still room for improvement—primarily in its kinetics—before it reaches the temporal resolution of state-of-the-art sensors like GCaMP8. However, in terms of brightness, photostability, and signal-to-noise ratio, PinkyCaMP already reaches the performance level of the latest GCaMP variants.”

Minor:

1. Fig 1 legend does not match the Figure panels. The data referred to as Fig 1G in the legend appears to be missing, and subsequent panels are mislabeled.

The mislabeling has been corrected, and all panels are now properly labeled and referenced.

2. Also in Figure 1, the frequency of stimulation in panel H appears to be faster than 0.1 Hz based on the time-scales in the Figure. Please confirm.

*The time-scale of the full traces in **Figure 1h** was mislabeled, and thus has been corrected. Similarly, the legend has been reworded to clarify pulses of blue light occurring at 10 second intervals.*

Reviewer #2 (Remarks to the Author):

As mScarlet is indeed the brightest fluorescent protein (FP) in the red wavelength region and various variants of mScarlet are being developed actively, it is natural for many researchers to expect the development of mScarlet-based bright biosensors. Just as expected, Masseck, Campbell and their colleagues successfully developed red intensimetric sensors for calcium with exceptional brightness, photostability, and multiplexing capabilities. Overall, I think that the manuscript presents sufficient information that demonstrates the usefulness of the product, PinkyCaMP.

Most FPs, including the classic variants of Aequorea GFP, have Tyr at the second position of the chromophore-forming tripeptide. As a result, they each show a bimodal absorption spectrum characterized by two peak maxima, which correspond to the protonated (neutral) and deprotonated (ionized) states of the chromophore. And this equilibrium is determined by a hydrogen bond network around the chromophore. In single FP-based biosensors, furthermore, the equilibrium can be modulated by the environment that their sensing domains generate. Therefore, GECIs, such as GCaMPs, exhibit a big change in their absorption spectra with and without calcium. In a recent Nature paper by Looger and colleagues (Zhang et al., Nature 615, 884-891 (2023)), for example, three jGCaMP8s exhibited >20-fold increases

in absorbance but only slight changes in fluorescence quantum yield upon binding to calcium (Zhang et al., Extended Data Table 3). So, I was very surprised to see very large differences in fluorescence quantum yield presented in Table 1 of this manuscript (PinkyCaMP, 16 fold; PinkyCaMP0.9b, 21.5 fold). It is concluded that in PinkyCaMPs calcium binding enhances the emission of light but not the absorption of light, for some reason. I think this is a remarkable departure from the conventional GECI concept. For example, I am certain that the lifetime of PinkyCaMP changes with calcium and would suggest its combination with FLIM technology. I also investigated some other GECIs developed by Campbell group recently. In one paper (Dalangin et al., Nature Comm. 16:3318 (2025)), FR-GECO1c showed a large change in absorbance with calcium and a moderate change (4.5 fold) in fluorescence quantum yield.

Thank you for this insightful comment. We agree that this is a remarkable feature of PinkyCaMP, and future work will certainly further investigate lifetime changes. However, this was not the main focus of the present study.

In relation to this point, I have some comments as follows.

#1 Photostability (Figure 1i)
Baseline brightness in HEK cells expressing PinkyCaMP, RCaMP3, or jRGECO1a was monitored with time. I understand that all these GECIs were depleted of calcium. So I strongly suggest that these data be normalized by absorption cross section or extinction coefficient. According to Table 1, the extinction coefficients of PinkyCaMP, RCaMP3, and jRGECO1a are 71,000, 3,300, and 6,180, respectively. In this situation, PinkyCaMP is highly disadvantaged. Alternatively, I would suggest that these experiments be performed with persistent calcium mobilization. Due to increases in absorbance, RCaMP3 and jRGECO1a would photobleach faster. Likewise, please examine Figure 3f and 3g.

Many thanks for pointing this out! PinkyCaMP has indeed a much higher baseline extinction coefficient than RCaMP3 and jRGECO1a, and PinkyCaMP might thus be expected to show more photobleaching in the absence of calcium than the other sensors. However, PinkyCaMP shows less bleaching (Figure 1i, Figure 3f). Following the reviewer's suggestion, we corrected for this effect, which even more highlights the superior photostability of PinkyCaMP. Yes, the HEK cells were analyzed without any induction or inhibition of Ca²⁺. As such, there are only the naïve cytosolic calcium levels which kept the sensors in an equilibrium of partially to fully bound and unbound states.

As requested we normalized the recorded HEK cell data by the extinction coefficients (please see new Figure 1i inset). We have likewise included photobleaching experiments with purified protein and persistent calcium, and additionally normalized to the extinction coefficients (Figure S3).

However, we refrained from performing these corrections in Figure 3, as we cannot be sure about the basal calcium concentration in neurons, which would be required for the proposed correction. In any case, also this data shows the increased fluorescence intensity (Figure 3d),

increased photostability (Figure 3f), and absence of photoactivation (Figure 3g) of PinkyCaMP even without correcting for its higher extinction coefficient. Experiments at persistently high calcium concentrations are not possible, at least not for experiments involving neurons.

#2 Minor comments
The legend for Figure 1g-l is a mess.
Please specify the wavelengths for extinction coefficients.

We correct the mislabeled legend accordingly. We have also amended Table 1 to clarify the wavelengths used to measure extinction coefficients.

Reviewer #2 (Remarks on code availability):

I was able to open the Python script for analysis but had no time to review it carefully.

Reviewer #3 (Remarks to the Author):

Fink et al describe the engineering and testing of PinkyCaMP, a new red fluorescent genetically encoded calcium indicator (GECI) that features high brightness, large responses, and lack of photoactivation by blue light. Of these features, the last one is arguably the most unique and useful, as orthogonality with blue-excitable opsins is one of the major reasons to develop red indicators, and previous red GECIs all showed photoactivation.

PinkyCaMP demonstrates impressive metrics in the parameters mentioned above, as a result of heroic engineering efforts involving 12 rounds of random mutagenesis. The in cellulo, ex vivo, and in vivo characterization are well done and relevant for the most part. The experiment with GtAChR in vivo is especially rigorous and elegant.

Thank you for your kind words, we really appreciate them.

However there are a few issues that influence the rigor of the conclusions and the utility of PinkyCaMP that should be rectified prior to publication of this manuscript. The major ones are:

1. The numbers in Table 1 do not support PinkyCaMP0.9c (selected as “the” PinkyCaMP) being superior to PinkyCaMP0.9b. Authors state “PinkyCaMP0.9c was selected as the best variant for its balance of brightness and responsiveness and renamed as PinkyCaMP”. However Table 1 shows the overall dF/F (from calcium-free to -saturated states) to be 14.3 for PinkyCaMP0.9b vs 12.8 for PinkyCaMP(0.9c). Meanwhile the calcium free/saturated brightnesses are 1.6/24.4 for PinkyCaMP0.9b vs 2.1/28.9 for PinkyCaMP(0.9c). These

differences thus favoring 0.9b in one metric and 0.9c in the other, but both by small degrees. Alone it does not seem sufficient to explain the choice, so *the authors should either present more convincing data to rule out 0.9b, or continue to characterize 0.9b more*. Specifically, it is not clear if the high affinity of PinkyCaMP0.9c at 54nM is an asset in this case. The past history of GECIs is marked by a realization that selection for high affinity without coselection for fast activation kinetics leads to sensors with very slow off-rates. That appears to have happened here too. The >5-second off-rates in Fig. 2I when CoChR is tuned on for only 1ms is rather striking, and *the authors should mention this result explicitly in the text*. jGCaMP engineering on the other hand specifically sought to identify sensors that improved affinity via increasing activation kinetics, which allows separation of closely spaced APs. Selection for fast on-kinetics does not appear to have happened here, and PinkyCaMP0.9c has very slow off-kinetics. Meanwhile 0.9b has 4-fold lower K_d . It's likely it has faster off-kinetics, maybe even 4-fold faster. This may be more useful to researchers and would be willing to sacrifice the slightly higher brightness of 0.9c to gain the faster kinetics of 0.9b. Indeed the authors appear aware of this issue as they mention the desire to accelerate kinetics in the discussion. But maybe they already have a faster indicator in 0.9b! *To settle this question, single AP measurements are necessary*. Indeed the GECI field has long coalesced on using single-AP responses to provide figures of merit, which have been reported for every GCaMP since GCaMP3 and was seen in the jRGECO1a and RCaMP3 papers as well. In single-AP responses, it is possible PinkyCaMP0.9c will outperform PinkCaMP0.9b in dF/F given its lower K_d , but importantly the authors will be able to ascertain times of half-rise and half-decay. The RCaMP3 and jRGECO1a papers reported responses to single APs in neurons by electrical stimulation of neuronal cultures or by patch-clamping, respectively. Either method could be performed here. I'd suggest this should be done for both 0.9b and 0.9c, and if there are advantages of 0.9b, it should be presented as an useful alternative. For example if 0.9b decay is faster, it could be PinkyCaMP1f and 0.9c could be PinkyCaMP1s. Notably such an elevation of status for 0.9b would not necessitate repeating the in vivo experiments with 0.9b, as the existing experiments with 0.9c are sufficient to prove what the authors want to demonstrate, e.g. ability to be used with blue-excitable opsins.

*Thank you for this highly relevant comment. We included an additional supplementary figure which explains why we originally went on with the PinkyCaMP 0.9c and not with 0.9b.: the brightness of 0.9c was almost twice as high in HEK cells as that of 0.9b. Please see additional **Figure S3g**. We also included additional in depth characterization of PinkyCaMP to spontaneous spikes in mouse cultures (**Figure 2a-c**) and to different field stimuli (**Figure 2d-j**)*

*We totally agree with the reviewer that 0.9b might be a variant with superior kinetics however, in this very first proof of principle to design an mScarlet based variant we have decided to first focus on other properties. We are sure that changing the calcium affinity in follow up variants can be easily adjusted to less sensitivity. We also followed the advice of the reviewer to co-image GCaMP8 and PinkyCaMP in the same cells (**Figure S12**) and in two*

separate neuronal populations (Figure 7). Indeed, PinkyCaMP might be useful for dual color imaging approaches.

We have now included a more in depth discussion of the slow kinetic drawback in the discussion (please see discussion).

We also included the requested new data on single AP to make the comparison to other GECIs for the reader more easily. Please see expanded Figure 2(a-j) and Table 2.

2. The Fig. 3 slice experiment is extremely important as the only experiment in which PinkyCaMP is compared to a green GECI (GCaMP6) and the other best red GECIs, jRGECO1a and RCaMP3. The only activity that is imaged is cyclical spontaneous bursting, but only one sample trace is shown for each indicator and these are split between main Fig. 3 and Fig S5. *Representative traces for all indicators should be shown in Fig. 3 so that some primary data is presented for each indicator*. Care should be taken that the displayed traces are near the mean of their group for frequency and dF/F.

We followed this suggestion and now included representative traces in the main Figure 3.

3. The differences in bursting behavior in organotypic slices expressing the different GECIs in Fig. 3 are concerning. The periods of these bursts is significantly lower for PinkyCaMP than jRGECO1a. This may indicate that PinkyCaMP is slowing synaptic transmission, perhaps due to calcium buffering at synaptic terminals or longer-term suppression of neuronal maturation via calcium quenching. This may be another situation where perhaps PinkyCaMP0.9b would be better. *It would be useful to add an experiment comparing PinkyCaMP0.9c at two different gene expression levels (unchanged and 10x lower, for example) to PinkyCaMP0.9b and jRGECO1a at the same two levels*. This will inform the reader whether overexpression may have effects on neuronal physiology, and whether this is more likely to happen for PinkyCaMP0.9c, PinkyCaMP0.9b, or jRGECO1a.

The different calcium sensors do give different signals in the organotypic slice experiments, in particular in terms of their decay times, which are slow for jRCaMP1a and PinkyCaMP (Figure 3e), and the observed event frequencies (Figure S5e). However, the prolonged decay times are foremost a result of the slow kinetics and high affinity of these sensors, rather than indicating synaptic changes. Similarly, the resolvable frequencies, which for jRCaMP1a and PinkyCaMP level at 0.1 Hz (Figure S5e), are set by the slow decay kinetics of these two sensors ($\tau = 1100$ ms; $\tau = 5572$ ms, respectively). In summary, the data does not give any indication for different firing properties of the respective slices, but represents what is expected for the different sensor kinetics.

We are also not aware of any published data that would indicate that jRCaMP1a, which shows the same slow behaviour, has major effects on synaptic physiology. Furthermore, while in

general detrimental effects of Ca²⁺ indicator expression cannot be ruled out (especially in developmental contexts, see e.g. Gasterstädt et al. 2020, PMC7606991), we want to note that PinkyCaMP was well tolerated by our slice cultures (expression for up to 3 weeks or more) without signs of neuronal damage. In addition PinkyCaMP transients were recorded in vivo repetitively at 60 and 90 days (!) post-injection underscoring the stability of expression and possibility of long-term measurements (Figure S9b-d). Across all recordings and animals (n=11 mice), no Ca²⁺ micro-waves were observed, which have previously been reported for GCaMP overexpression (Masala et al. 2024), pointing again to high tolerability of PinkyCaMP.

The suggested experiments with lowered expression cannot readily be achieved with AAV transductions and the epileptiform activity in slice cultures measured here cannot provide a sensitive readout for synaptic changes or circuit maturation. However, if calcium buffering is a concern, the large signal obtained with PinkyCaMP would allow for working with lower indicator expression while maintaining high signal-to-noise ratios.

4. The only photostability experiments are done in 1-photon excitation in Fig. 2b. *Photostability should also be characterized in 2-photon excitation*. As the RCaMP3 paper already compared RCaMP3, jRGECO1a, and XcaMP-R for photostability under 1040nm excitation (Extended Data Fig. 4), the authors here would only need to compare PinkyCaMP0.9c, PinkyCaMP0.9b, and RCaMP3 under the same 1040nm excitation and the same power, and the other comparisons can be made transitively.

We thank the reviewer for this helpful comment. We carried out photostability measurements with two-photon excitation at 1040 nm (Figure S9a). PinkyCaMP is highly photostable at 1040 nm excitation wavelength.

5. In the paragraph describing Figure 6, imaging at 10 Hz cannot fairly be described as picking up unitary events. Unitary somatic events would be understood to be single APs. In addition to the slow imaging, the slow kinetics of PinkyCaMP would make that difficult anyway. Likewise there is no basis to claim that “These data show reliable measurement of Ca²⁺ transients with PinkyCaMP.” In other GECl papers, similar claims derive only from simultaneous ephys (loose-cell is enough) and GECl imaging in acute slice or in vivo. Here, all that can be claimed is that PinkyCaMP can detect some activity. *To make this experiment informative, the authors should coimage a GCaMP6, 7, or 8 in the same cell*. It’s surprising this is not done anywhere in the paper since that is an easy way to benchmark to well studied indicators in vivo.

We thank the reviewer for this helpful comment. We absolutely agree that 10 Hz imaging cannot detect unitary events. We removed “unitary events” throughout the paragraph. Furthermore, we adjusted the wording concerning “reliable Ca²⁺ measurements”.

We also followed the excellent suggestion of the reviewer to co-image GCaMP8 and PinkyCaMP in the same cells (Figure S12), which provides a powerful direct comparison, or in

two separate neuronal populations (Figure 7). Indeed, PinkyCaMP might be useful for dual color imaging approaches.

Minor

points:

1. The statement “other monomeric red fluorescent proteins (RFPs) used in GECIs, such as mApple or mRuby have yields far below 50%” appears not well supported. Perhaps all that can be said is that they have quantum yields lower than mScarlet. Reference 22 shows mApple to have QY of 47%, which is not “far below 50%”. Shaner et al also measured QY of 49% for mApple in the existing reference 15 which should probably be cited again here. In addition, Ref 22 doesn’t include any work on mRuby so another reference is needed for that. As the brightness of isolated mRuby is not as important as the brightness of the GECIs derived from it, I think Dana et al eLife (existing ref 11) should be cited; here RCaMP1h, jRCaMP1a, jRCaMP1b, and mRuby itself all have quantum yield > 0.5 (<https://elifesciences.org/articles/12727/figures#fig2s2>). Indeed the authors in the present manuscript also measure the QY of the mRuby-based jRCaMP1a to be 54%.

Thank you for this insightful comment. We have changed the sentence accordingly and also added your suggested citations.:

“mScarlet has a quantum yield of about 70%, while other monomeric red fluorescent proteins (RFPs) used in GECIs, such as mApple or mRuby have quantum yields lower than mScarlet^{11,15,22}.”

2. mScarlet having minimal/negligible photoswitching behavior is repeated twice in the introduction. The second time is “Additionally, mScarlet demonstrates negligible photoswitching behavior²².”

Thank you for this helpful comment. We have deleted the repetition. The paragraph is changed to:

These attributes make mScarlet an excellent candidate for use in optogenetic tools and biosensors. However, possibly due to its limited structural similarity to other RFPs, no GECIs utilizing mScarlet have been developed yet.

3. Fig 1 caption section g refers to a plot that has been moved to supplementals, but section g was not deleted and the subsequent sections are now mislabeled.

Please excuse our mislabeling. We have now corrected the figure legend.

4. In Fig 1 panel h (h in the image, not h in the caption), the two colors in the color legend are too close to be differentiated. I suggest perhaps using an orange color for RCaMP3.

*We changed the color to a more distinct color as suggested (please see **Figure 1**)*

5. The sentence “We anticipated that this small signal increase could be either caused by the small photocurrent generated by stCoChR (Figure 2g) and/or by the activation of voltage gated calcium channels and its inherent high Ca²⁺-permeability paired with the high sensitivity of PinkyCaMP (K_d = 54 nM)” is not completely logical. The stCoChR is definitely opening since there is a difference in calcium signal between light and no light. So the first part of the sentence doesn’t belong after an “either”. Rather the sentence should be “We anticipated that this small signal increase could be caused by the small photocurrent generated by stCoChR (Figure 2g) with or without activation of voltage gated calcium channels paired with the high sensitivity of PinkyCaMP (K_d = 54 nM)”. Also it appears the authors meant to add “its inherent high Ca²⁺-permeability” after stCoChR but added it after VGCCs. Certainly VGCCs have inherently high Ca²⁺ permeability... they are after all calcium channels.

Thank you for this insightful comment, we changed the sentence to:

*“We anticipated that this small signal increase could be caused by the small photocurrent generated by stCoChR (**Figure 2g**) with or without activation of voltage gated calcium channels paired with the high sensitivity of PinkyCaMP (K_d = 54 nM).”*

6. In the sentence, “However, the application of orange light used for imaging caused a small inward current of 9 ± 3 pA, much lower as currents measured for action spectra determination at 480 nm (288 ± 65 pA, Figure 2h)”, the authors probably mean “much lower than current measured at low powers of 480-nm light used during action spectra determination.” But in addition, powers used during action spectra determination is meaningless to the reader. The authors should just write “the application of orange light used for imaging at ** mW/mm² caused an inward current of 9 ± 3 pA, which was much smaller than currents measured with 480 nm excitation at the much lower power of ** mW/mm²”.

Thank you for this insightful comment, we changed the sentence to:

*“The application of orange light used for imaging at 0.443 mW/mm² caused an inward current of 9 ± 3 pA, which was much smaller than currents measured with 480 nm excitation at the much lower power of 0.346 mW/mm² (288 ± 65 pA, **Figure 2o**).”*

7. Figure 2h graphic is awkward. If the authors mean that there are 3 conditions of field stimulation, 438nm, or nothing, then they should put the images of electrodes and diodes in one column or one row. An image of an opsin rather than a diode would be more informative anyway.

*We now changed the figure as requested which can be found as a new panel **Figure 2p**.*

8. “PinkyCaMP maintained strong baseline fluorescence (Figure 3l)” should refer to Figure 3m
Corrected to: Under the higher light conditions, PinkyCaMP maintained strong baseline fluorescence (Figure 3m).

9. Figures 3i and 3j are not cited. Figure 3i should really be combined with Figure 3l anyway.

We now refer to both subpanels in the text:

PinkyCaMP showed strong cytosolic and dendritic expression (Figure 3i) resulting in low-light recording capability (Figure 3i-k), with $\Delta F/F$ increasing significantly by field stimulation (Figure S6b,c). While neuropil fluorescence affected $\Delta F/F$ ratios, PinkyCaMP’s absolute fluorescent signal, photostability, and signal-to-noise ratio (SNR) remained high across trials. In contrast, RCaMP3 exhibited severe organellar, presumably lysosomal accumulation (Figure 3l).

10. The paragraph describing Figure S7 and Figure 3k-3o uses a high and low light power. The actual power in mW/mm² should be specified so that readers know what range of powers they might try.

The light power is given in the legend of Figure S6. Low light power: 0.23 mW/mm² high light power 11 mW/mm²

11. The authors should explain why they chose to perform all ex vivo comparisons to GCaMP6f instead of a jGCaMP8.

We performed the organotypic experiments using GCaMP6 as a reference because we considered it more comparable to PinkyCaMP. The engineering efforts behind GCaMP8 are significantly more advanced, and we never expected to match its brightness or kinetics. It is absolutely clear that GCaMP8 outperforms PinkyCaMP. However, by comparing PinkyCaMP to GCaMP6, we still aim to give the reader a sense of how much PinkyCaMP has improved relative to other red GECIs. We are also aware that a first-generation probe based on mScarlet is not yet expected to reach the performance level of GCaMP8. However we have included new in vivo data, where we expressed PinkyCaMP alongside with GCaMP8s, please see new Figure 7, Figure S12.

12. “PinkyCaMP exhibited robust Ca²⁺ transients” should be “PinkyCaMP reported...”

Corrected to “PinkyCaMP reported robust Ca²⁺ transients in response to the air puff”

13. Fig. 4j in the text should be Fig. 4i.

Corrected.

14. "As expected, PinkyCaMP signals increased during the light-ON epochs in the stGtACR2 group and had no effect on the EGFP group" should be "PinkyCaMP signals were not affected in the EGFP group"

Corrected to: "PinkyCaMP signals were not affected in the EGFP group..."

15. In the paragraph describing Figure 6, the 3 sentences starting from "First applications of GCaMP-imaging" do not really explain the utility of a red GECI and are not relevant anyway to the rest of the paragraph. The real utility is to image the red GECI together with another indicator for something else (e.g. neurotransmitter) in the same cell or closely apposed cells.

We thank the reviewer for the comment and changed the introduction of the paragraph. We just focus on red GECI in combination with other indicators now. Indeed, we also provide two color Ca²⁺-imaging now (Figure 7, Figure S12).

16. The term "dlox" in dlox-GtACR2-ST-mCerulean is not defined; and google search reveals no relevant results. Did the authors mean DIO?

"DIO" indicates double-floxed inverse orientation. "DIO" is a "concept" that implies:

two different lox sequences (= a pair of lox sequences) that are positioned twice upstream and downstream of an ORF

the ORF is inverted relative to the transcriptional direction of a promoter fragment

"dlox" that we use describes two different lox sequences.

the full name of the expression cassette is (mentioned in the methods)

ssAAV-{capsid}/2-hSyn1-chl-dlox-stGtACR2_mCerulean(rev)-dlox-WPRE-bGHp(A)

The "(rev)" at the end of the ORF (please note that individual elements in a single ORF are joined by underlines) indicates its reverse orientation relative to the transcriptional direction of a promoter fragment.

Therefore, no, "dlox" is not DIO. However, "dlox" plus "(rev)" are DIO.

We have added that information in the main text:

“PinkyCaMP, together with the soma-targeted chloride-conducting opsin GtACR2 (rAAV/DJ.dlox-GtACR2-ST-mCerulean³⁷, where “dlox” indicates a DIO configuration using two different lox sites and an ORF in reverse orientation to the promoter) or ...

Dear Dr. Vogt,

we would like to thank the editor and the reviewers for their positive feedback and insightful comments, as well as for the opportunity to address the remaining concerns raised by reviewer 3. All requests are addressed thoroughly and point-by-point in the following response.

Reviewer #1:

Remarks to the Author:

The authors have done a nice job of responding to the comments in the initial round of reviews. The addition of new data including simultaneous imaging of PinkyCaMP / GCaMP8s, relating signals to action potential firing number, and acknowledging the slower kinetics of the sensor are much appreciated. I am satisfied with these revisions and think that this paper is an important contribution that is highly appropriate for Nature .

We sincerely appreciate your positive assessment and recommendation.

Reviewer #2:

Remarks to the Author:

After reviewing the authors' responses to the reviewers' comments and the revised manuscript, I noticed a substantial improvement. While the slow kinetics of PinkyCaMP, which is likely due to its high calcium affinity, may still be a concern for researchers who are particular about temporal resolution, it is evident that PinkyCaMP outperforms other red GECIs in brightness, photostability, signal-to-noise ratio, and compatibility with general optogenetics and neurotransmitter imaging.

We sincerely appreciate your positive assessment and recommendation.

I have the following minor comments:

Methods section, page 4, line 7: "A 4x air" should be "A 40x water."
Clearly state how irradiance was measured in this study.

We changed the methods section accordingly and included a brief statement for the irradiance measurements:

" with a 40x water objective (LUMPLFLN40xW, Olympus), a CMOS camera (ORCA-spark, Hamamatsu), and 560 nm illumination at 57 mW/mm². Irradiance was measured with a calibrated S170C slide power sensor (Thorlabs)."

Figure S12: I suspect that the transient ratio of PinkyCaMP to GCaMP reflects their expression levels. This could be discussed in the legend.

We have added your discussion point to the legend of Figure S12 (now Extended Data Figure 8): “The relative transient amplitudes of PinkyCaMP and GCaMP observed across neurons likely reflect differences in individual expression levels of the two indicators, which can vary between cells despite co-transfection, rather than systematic differences in their calcium responses.”

Reviewer #3:

Remarks to the Author:

The authors are to be commended for this thorough revision in which they performed experiments to address each of the reviewers' concerns. The revision included high-quality cell-attached patch data, field stimulus data, and jGCaMP8s co-imaging data (the jGCaMP8s co-imaging is especially nice).

We sincerely appreciate your positive assessment and recommendation.

However I have a major concern regarding the interpretation of the field stimulus data in relationship to published work. It cannot be assumed that a single field stimulus creates a single AP. It's possible that it creates a range of AP numbers, e.g. 1-3 with a mean of ~2. It is also not possible to compare one lab's field stimulus results to those obtained in another lab; there are numerous potential differences. Thus the sentence “Across all samples, a single AP evoked an average response of $18 \pm 1\% \Delta F/F_0$ (Figure 2g), comparable to responses of other red calcium indicators and GCaMP6 variants (Table 2)” is unsupported in two critical words: “AP” and “comparable”. The groups in Janelia Farm using field stimulus made a point of validating that the field stimulus they described as generating one AP really did do so (Chen et al Nature 499:295, “Voltage imaging using the ArchWT-GFP archaerhodopsin-based voltage sensor confirmed that individual pulses (1 ms, 40 V, 83 Hz) reliably triggered single APs”), but I do not see that done here. Indeed on the patch-clamped neuron the one AP that was recorded only produced ~2.5% $\Delta F/F_0$. Thus the authors have several choices:

1. Characterize the distribution of AP numbers created by their single field stimulus pulses as in Chen et al, and adjust their field stimulus if necessary to evoke truly only 1 AP, and repeat the measurement if necessary (might be okay already, or might be tedious)
2. Skip the calibration, but characterize a GCaMP as a comparator in their field stimulation

system, and remove entries in Table 2 of other GECIs characterized in other labs in with field stimulation because those numbers may not be comparable, and replace the word “AP” in the sentence above with “field stimulus”, and modify the phrase starting with “comparable” to state the actual result with the GCaMP in the same system.

3. Do neither of the above, but consider the dF/F to be an upper bound for a single AP (as it may reflect 1 or more APs), and state clearly in the text that the field stimulus elicits a calcium response but the number of underlying APs is unknown, and describe a single-AP response as “ $\leq 18 \pm 1\%$ ” in the text, and place an asterisk in Table 2 next to “ $< 18 \pm 1\%$ ” to explain that the conditions tested may not have elicited only a single AP, and likewise also add “ \leq ” before each other number in Table 2 for the PinkyCaMP row.

Related to this, Table 2 should not mix values from cultured unpatched neurons and patched neurons in vivo; these are expected to have completely different dF/F (because imaging methods will have different backgrounds and background subtractions, as well as the presence of a electrode in the way) and SNR (obviously, because they will be using different objectives, illuminators, filters, collectors, imaging gain, and time). Indeed SNR should be dropped completely as no two labs will have exactly the same combination of objectives, illuminators, filters, collectors, and imaging gain and time. About the only thing that maybe can be compared across papers is cultured neuron single field pulses, and only if a common GECI functions similarly across the papers (maybe GcaMP6f).

My desire here is simply that any comparative claims put into print are correct, or at least as likely to be correct as reasonable. Here without an in-experiment side-by-side comparison under the same conditions, no reliable comparison of performance can be made. If these issues can be corrected, then the manuscript would be a strong candidate for publication in this journal.

We thank the reviewer for this detailed and insightful comment, and we fully agree that the interpretation of field stimulation data as well as the comparison of data obtained by different laboratories requires caution. We appreciate the reviewer’s suggestions and have revised the manuscript accordingly. We fully agree that the exact number of action potentials evoked by a single field stimulus cannot be assumed.

However, implementing option 1: systematically characterizing the distribution of action potentials induced by each stimulus, as done by Chen et al.—would require extensive recordings in multiple preparations under the same optical conditions used for imaging. Given that field-stimulation efficacy varies with culture health, electrode placement, and expression level, such a calibration would ideally need to be repeated for each preparation and indicator, making it technically demanding and time-intensive while not altering the principal conclusions of our study.

Option 2, introducing an internal GCaMP comparator, would also be informative but would necessitate additional viral constructs and imaging campaigns dedicated solely to cross-sensor calibration. As the main goal of this study is to characterize PinkyCaMP performance, this would considerably broaden the experimental scope and delay completion without providing fundamentally new insight into PinkyCaMP's properties. However, for clarity, we have replaced every mention of "AP" with "stimuli".

Option 3 therefore offers the most appropriate solution: First we now explicitly state that our field-stimulation protocol elicits calcium responses whose underlying number of action potentials is unknown and that the reported $\Delta F/F_0$ values represent an upper bound for a single-AP response. We now explicitly state that the observed $\Delta F/F_0$ values represent an upper bound for a single action potential, since a single field stimulus may evoke a variable number of spikes (e.g., 1–3 APs) depending on cell excitability. To address this, the relevant sentences in the Results section have been revised to read:

"To achieve tighter control over neuronal activity, we next monitored PinkyCaMP Ca^{2+} transients during field stimulation (**Fig. 2d**). The Ca^{2+} transient amplitudes scaled with the number of stimuli (**Fig. 2d–e**), faithfully reporting neuronal activity at the single-cell level (**Fig. 2e**). Across all samples, a single field stimulus evoked an average calcium response of $18 \pm 1\% \Delta F/F_0$ (**Figure 2g**, representing an upper bound for the response to a single AP as the number of underlying APs has not been verified) falling within the range reported for other red calcium indicators and GCaMP6 variants (**Extended Data Table 2**). Importantly, PinkyCaMP exhibited a superior signal-to-noise ratio, reaching 129 ± 7 for a single field stimulus and 172 ± 7 for 10 field stimuli (**Fig. 2h**; **Extended Data Table 2**). PinkyCaMP showed relatively slow kinetics, with a half-rise time of 670 ± 18 ms and a half-decay time of 5.6 ± 0.1 s (**Fig. 2i–j**)."

We have also added a legend in **Extended Data Table 2** indicating that responses were measured under field stimulation and may reflect responses to one or more action potentials. Thus $\Delta F/F_0$ values represent upper bounds for single-AP responses. Comparisons across different studies from different labs is difficult due to methodological differences in imaging configuration, such as objectives, illuminators, filters, detectors, gain, frame rate and sample preparation. As imaging conditions differ across studies, our $\Delta F/F_0$ values should be interpreted as upper bounds rather than directly comparable measurements.

Second, we agree with the reviewer that direct quantitative comparisons across laboratories and experimental systems are not appropriate and quite difficult to achieve. As the reviewer correctly noted, differences in imaging configurations (objectives, illuminators, filters, detectors, gain, frame rate, etc.) and sample preparations (cultured neurons vs. in vivo patch-clamped neurons) can profoundly affect both $\Delta F/F_0$ amplitude and signal-to-noise ratio (SNR). We have therefore removed SNR values from **Extended Data Table 2** and in addition added color coding that differentiate between patched or unpatched neurons and highlights if

recordings are done in vivo. However, we wanted to provide the reader with a comparison to previously published data.

Legend to Extended Data Table2:

“Values for PinkyCaMP were obtained using field stimulation (as performed in Chen et al.^e with an 1 ms, 40 V, 83 Hz field stimulus) and may reflect responses to one or more action potentials. Thus $\Delta F/F_0$ values represent upper bounds for single-AP responses. Comparisons across different studies from different labs is difficult due to methodological differences in imaging configuration, such as objectives, illuminators, filters, detectors, gain, frame rate and sample preparation. As imaging conditions differ across studies, our $\Delta F/F_0$ values should be interpreted as upper bounds rather than directly comparable measurements.”

We believe these revisions address the reviewer’s concerns by eliminating unsupported comparisons and clearly stating the methodological limitations.